# In silico discovery of representational relationships across visual cortex

**Alessandro T. Gifford** [1,2,3] ✉, **Maya A. Jastrzębowska**[1], **Johannes J. D. Singer**[1] & **Radoslaw M. Cichy** [1,2,3,4]

Human vision is mediated by a complex interconnected network of cortical brain areas that jointly represent visual information. Although these areas are increasingly understood in isolation, their representational relationships remain unclear. Here we developed relational neural control and used it to investigate the representational relationships for univariate and multivariate functional magnetic resonance imaging (fMRI) responses of areas across the visual cortex. Through relational neural control, we generated and explored in silico fMRI responses for large numbers of images, discovering controlling images that align or disentangle responses across areas, thus indicating their shared or unique representational content. This revealed a typical network-level configuration of representational relationships in which shared or unique representational content varied on the basis of cortical distance, categorical selectivity and position within the visual hierarchy. Closing the empirical cycle, we validated the in silico discoveries on in vivo fMRI responses from independent participants. Together, this reveals how visual areas jointly represent the world as an interconnected network.

Human vision is mediated by a complex interconnected network of cortical areas that jointly represent visual information[1–9]. The network consists of hierarchies and loops, with each area distinctly responding to visual properties of incoming visual stimuli, resulting in idiosyncratic representations of visual phenomena[10–14].

Over the past half century, taking an atomistic approach, neuroscientists have studied visual representations by characterizing each area in isolation of other areas in a hypothesis-driven fashion using small, limited sets of stimuli carefully chosen by the experimenter. Seminal work in this spirit built the foundations of modern vision neuroscience, from characterizing the role of primary visual cortex for processing of oriented edges[15] to elucidating the role of higher-level visual cortex for processing of complex visual categories such as faces, places and objects[16].

However, assessing areas one by one does not capture the visual system as an interconnected network; it does not assess representational relationships between areas and thus remains silent about what

representational content is shared between areas or unique to a specific area. While anatomical[2] and functional[17] connectivity research assess the visual system at the network level, they miss what representational content the network encodes. Compounding the situation, theories of visual representations are based on sparse neural data for small sets of experimenter-picked stimuli, risking to reproduce experimenter biases while missing important neural signals that would be available from broad sampling.

Here, we addressed these challenges by developing relational neural control (RNC) and used it to reveal representational relationships between early-, mid- and high-level visual areas across the human visual cortical network (that is, visual areas one to four (V1, V2, V3, V4), extrastriate body area (EBA), fusiform face area (FFA), parahippocampal place area (PPA) and retrosplenial cortex (RSC)). First, through deep-neural-network-based encoding models[18–20], RNC generated in silico functional magnetic resonance imaging (fMRI) responses of these areas for a larger set of naturalistic images than are available in vivo.

[1]Institute of Psychology, Freie Universität Berlin, Berlin, Germany. [2]Einstein Center for Neurosciences Berlin, Charité – Universitätsmedizin Berlin, Berlin, Germany. [3]Bernstein Center for Computational Neuroscience, Humboldt-Universität zu Berlin, Berlin, Germany. [4]Berlin School of Mind and Brain, Humboldt-Universität zu Berlin, Berlin, Germany. ✉e-mail: alessandro.gifford@gmail.com

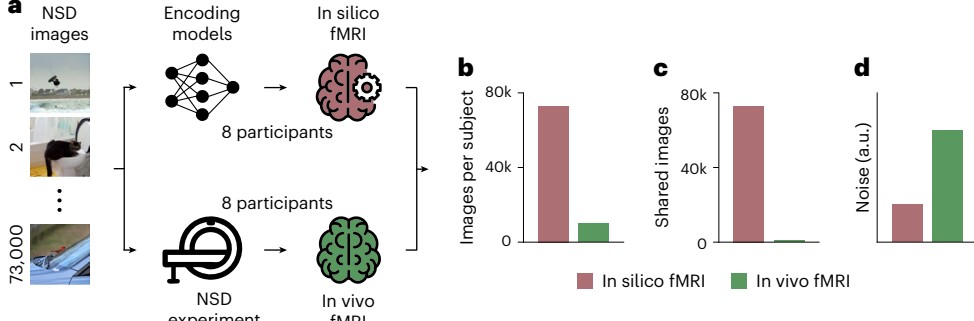

**Fig. 1 | RNC provides accurate and denoised in silico fMRI responses for thousands of images. a**, Through encoding models, we generated in silico fMRI responses to each of the 73,000 images and for each of the 8 participants in the NSD. We then compared these in silico responses with the in vivo fMRI responses from the NSD. Photos from the COCO image dataset/Flickr[85]. **b**, Comparison of the number of image conditions presented to each participant, for the in silico and in vivo fMRI responses. **c**, Comparison of the number of image conditions shared across participants, for the in silico and in vivo fMRI responses. **d**, Comparison of in silico and in vivo fMRI response noise, in arbitrary units.

This in turn enabled the evaluation of a larger, more diverse and, thus, less biased hypothesis space. Next, to uncover representational relationships, RNC selected controlling images aligning or disentangling the in silico fMRI responses of the areas at both their univariate (that is, voxel average)[16,21–23] and multivariate (that is, voxel population pattern)[21,22,24–26] response level, under the assumption that alignment or disentanglement are indicative of shared or unique representational content, respectively. Finally, we validated our in silico findings in vivo through new experiments on independent participants, thus closing the empirical cycle and validating RNC as a powerful exploratory neural control method to investigate representational relationships.

## Results

### RNC provides accurate and denoised in silico fMRI responses for thousands of images

Using RCN, we determined representational relationships across visual cortex, starting with human early- and mid-level visual cortical areas (V1, V2, V3 and V4).

The first step was creating high-quality in silico brain responses for a large set of visual stimuli (Fig. 1a). For this, we used the Natural Scenes Dataset (NSD)[27], a large-scale dataset of 7 T fMRI responses from 8 participants who each viewed ca. 10,000 natural scenes, for a total of 73,000 images across participants, with 1,000 images shared across participants. We trained participant-specific encoding models for areas V1 to V4, mapping image activations from a visual artificial deep neural network onto voxel-wise fMRI responses (Extended Data Figs. 1 and 2a). The trained encoding models accurately predicted fMRI responses not used for training, resulting in a participant-average noise-ceiling-normalized explained variance score of 78.14% for V1, 72.54% for V2, 65.07% for V3 and 53.29% for V4 (Extended Data Fig. 2b; single-participant results shown in Extended Data Fig. 3a). We further tested the robustness of the encoding models on NSD-synthetic[28], the NSD out-of-distribution companion dataset of fMRI responses to artificial images, obtaining a participant-average noise-ceiling-normalized explained variance score of 60.72% for V1, 52.22% for V2, 46.75% for V3, and 38.89% for V4 (Extended Data Fig. 3a). These results indicate that the trained encoding models generate reliable in silico fMRI responses, including for images very different from the ones on which the models were trained, therefore providing a solid foundation for in silico experiments. Using the trained encoding models, we generated in silico fMRI responses to all 73,000 NSD images for each of the 8 participants, thus increasing the number of image-specific brain responses per participant by a factor of ~7 (Fig. 1b).

This had three advantages. First, the large number of responses allowed a wider exploration than possible with in vivo data, thus reducing experimental biases inherent in small datasets. Second, as the in-silico-generated fMRI responses for the whole 73,000 images were present for all participants, this allowed more robust cross-participant validation than would be possible using the in vivo responses to only 1,000 shared images from the NSD (Fig. 1c), thus reducing overfitting. Finally, because neural noise is not predictable from the stimulus images, encoding models modelled the signal- and not noise-related variability of the neural response[19,29], thus resulting in silico fMRI responses less affected by noise compared with the NSD responses (Fig. 1d; for the noise comparison, see Extended Data Fig. 2c,d).

Together, this provided the basis for revealing representational relationships.

### RNC controls in silico univariate fMRI responses across early- and mid-level visual cortical areas

We began by investigating representational relationships for in silico univariate fMRI responses (that is, the average activity over all voxels within an area), thus capturing visual information encoded in the strongest activation trends common across voxels[16,21–23].

For each pairwise comparison of areas (V1 versus V2, V1 versus V3, V1 versus V4, V2 versus V3, V2 versus V4, and V3 versus V4), we used univariate RNC to search, across all 73,000 NSD images, for images that would either align or disentangle (that is, control) the in silico univariate fMRI responses of the two areas being compared, thus indicating shared or unique representational content, respectively. Alignment consisted in two neural control conditions where the univariate responses of both areas were either driven or suppressed. Disentanglement consisted in two neural control conditions where the univariate response of one area was driven while the response of the other area suppressed, or vice versa (Fig. 2a; the univariate RNC algorithm is visualized in Extended Data Fig. 4). To assess the success of the neural control conditions, we compared them against a baseline of univariate responses for a set of images selected without optimization. We used cross-participant validation, thus ensuring generalization of results.

Through univariate RNC, we found images that significantly drove and suppressed univariate responses of most pairwise comparisons of areas (within-participant permutation test, one-sided, $P < 0.05$, Benjamini–Hochberg corrected over eight tests for each pairwise comparison of areas; population prevalence test, one-sided, $P < 0.01$, indicating within-participant significance in at least 3 out of 8 participants) (Fig. 2b, upper triangle of the results matrix). Thus, we successfully aligned or disentangled different areas at the univariate response level. For each pairwise comparison of areas, we then visualized the in silico fMRI response manifolds for all 73,000 images in univariate activity space and found their activation profiles to be highly correlated, suggesting that a large portion of representational content is shared across areas (Fig. 2b, lower triangle of the results matrix).

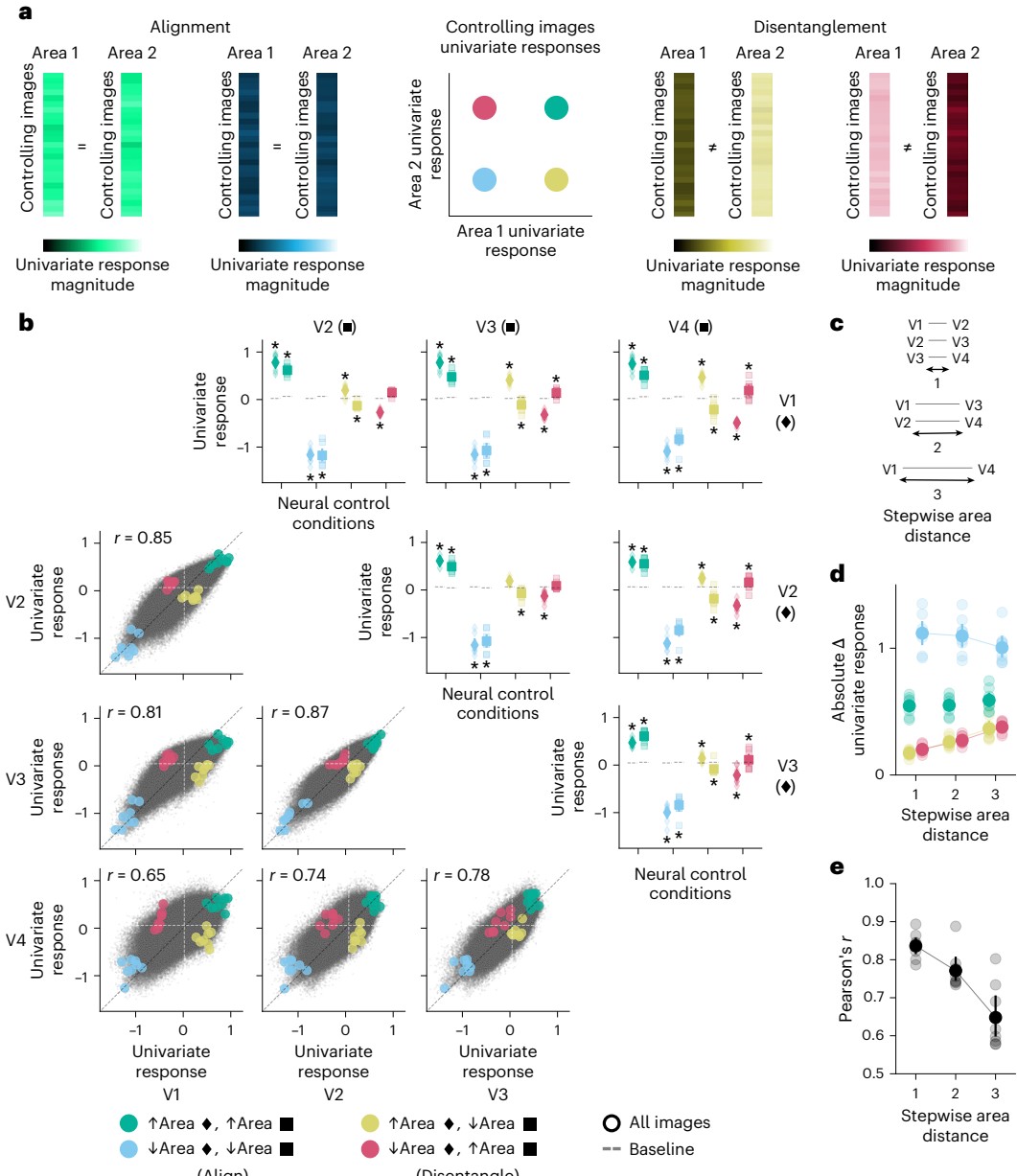

**Fig. 2 | RNC controls in silico univariate fMRI responses across early- and mid-level visual cortical areas. a**, Univariate RNC neural control conditions. **b**, Univariate RNC results for each pairwise comparison of areas, embedded in a four-by-four matrix. The upper triangle of the results matrix shows the univariate responses for the controlling images against the baseline. Diamonds and squares indicate the univariate responses of the areas indexed by the rows and columns of the results matrix, respectively. Asterisks indicate neural control conditions for which the in silico univariate fMRI responses for the controlling images are significantly different from baseline (within-participant permutation test, one-sided, $P < 0.05$, Benjamini–Hochberg corrected over eight tests for each pairwise comparison of areas; population prevalence test, one-sided, $P < 0.01$, indicating within-participant significance in at least 3 out of 8 participants). The lower triangle of the results matrix shows the univariate response image manifolds. Coloured dots indicate in silico univariate fMRI responses averaged across the controlling images of each neural control condition, and small black points indicate in silico univariate fMRI responses of all participants for all 73,000 NSD images. Vertical and horizontal dashed lines indicate the participant-average univariate response baseline for each area. **c**, Stepwise distances between areas. **d**, Absolute difference between controlling and baseline image univariate responses, averaged across all pairwise comparisons of areas with same stepwise distance. Connectors between area distances indicate a significant increasing or decreasing trend (within-participant permutation test, one-sided, $P < 0.05$, Benjamini–Hochberg corrected over four tests; population prevalence test, one-sided, $P < 0.001$, indicating within-participant significance in at least 4 out of 8 participants). **e**, Correlation between univariate responses in two areas, averaged across pairwise comparisons of areas with same stepwise distance. Connectors between area distances indicate a significant decreasing trend (within-participant permutation test, one-sided, $P < 0.05$; population prevalence test, one-sided, $P < 1 × 10^{-10}$, indicating within-participant significance in all 8 participants). In **b**–**e**, opaque and transparent diamonds, squares and dots represent participant-average and single participant results, respectively. Error bars reflect 95% confidence intervals.

Visual areas V1 to V4 form a processing hierarchy in terms of anatomical connectivity[5], response latency[30] and the complexity of stimulus properties maximally driving neural responses[1]. This suggests that disentanglement should increase and that alignment should decrease with increasing node distance across this hierarchy (Fig. 2c). We confirmed this prediction. As the stepwise distance between two areas increased, the absolute difference between the in silico univariate fMRI responses in the disentangling control condition and the

baseline increased (within-participant permutation test, one-sided, $P < 0.05$, Benjamini–Hochberg corrected over four tests; population prevalence test, one-sided, $P < 0.001$, indicating within-participant significance in at least 4 out of 8 participants) (Fig. 2d, yellow and red dots). Furthermore, the absolute distance between the univariate responses in the alignment control condition suppressing both areas (but not driving them) and the baseline decreased (within-participant permutation test, one-sided, $P < 0.05$, Benjamini–Hochberg corrected over four tests; population prevalence test, one-sided, $P < 1 \times 10^{-9}$, indicating within-participant significance in 7 out of 8 participants) (Fig. 2d, blue dots). This indicates that the univariate responses of areas further away from each other were less aligned and more strongly disentangled. Strengthening this finding, as the stepwise distance between two areas increased, the correlation between their univariate responses decreased (within-participant permutation test, one-sided, $P < 0.05$; population prevalence test, one-sided, $P < 1 \times 10^{-10}$, indicating within-participant significance in all 8 participants) (Fig. 2e).

To ascertain that the demonstrated representational relationships reflect properties of visual processing, rather than biases of specific datasets, we performed two tests. First, to ensure that RNC's solutions are not biased by the visual distribution from which the controlling images are selected, we applied univariate RNC on the in silico fMRI responses for the 50,000 images from the ImageNet 2012 challenge validation split[31], and for the 26,107 images from the THINGS database[32] (that is, single objects presented centrally on natural backgrounds, as opposed to the NSD's complex natural scenes consisting of several or no objects appearing at different locations). Second, to ensure that RNC's solutions are not biased by the visual distribution of the encoding models' training data, we applied univariate RNC on the in silico fMRI responses generated from encoding models trained on the Visual Illusion Reconstruction (VIR) dataset[33] (that is, fMRI responses for images of diverse objects, natural scenes and materials). Both tests replicated our previous findings (Supplementary Figs. 1–3), demonstrating RNC's robustness and generalizability and indicating that its solutions truly reflect properties of the brain.

Together, through univariate RNC we discovered controlling images that align or disentangle the in silico univariate fMRI responses of multiple areas, revealing that a large portion of univariate responses representational content is shared between areas and that unique representational content increases as a function of cortical distance.

## Spatial frequency and object-like shapes determine unique representational content for V1 and V4 in silico univariate fMRI responses

To determine the visual features leading to aligned or disentangled responses of different areas, we visualized the controlling images that aligned or disentangled their univariate responses. Here, we exemplarily focus on the V1 versus V4 comparison (Fig. 3a).

The controlling images driving V1 while suppressing V4 responses contained high-spatial-frequency backgrounds (for example, green vegetation), whereas the controlling images driving V4 while suppressing V1 responses contained one or multiple objects on a low-spatial-frequency background (for example, a plane on a sky background). Controlling images driving or suppressing both areas simultaneously were the logical combination thereof: high spatial frequency and objects were present in controlling images driving the response of both areas (for example, warm-colour cluttered food items), whereas they were lacking in controlling images suppressing the response of both areas (for example, empty skies). As expected, we discerned no consistent visual patterns in the baseline images. When using alternative distributions of images and of encoding model training data, the resulting controlling images also consisted of combinations of high spatial frequencies and objects (Supplementary Figs. 1–3). However, they did not always contain green vegetation, planes on a sky background or warm-colour cluttered food items (as was the case

with the NSD images in Fig. 3a), suggesting that these visual categories correlate with, but are not, the visual features controlling univariate responses. This showed, through large-scale exploratory analysis using naturalistic images from diverse image sets, that V1 is uniquely tuned to high-spatial-frequency content[34,35], whereas V4 is uniquely tuned to object-like shapes[36].

Naturalistic images are complex combinations of multiple visual features, making it challenging to isolate, by mere visual inspection, the features leading to aligned or disentangled responses across areas. To further isolate the relevant visual features, we generated de novo controlling images that controlled univariate responses, while being as simple as possible. To this end, we combined RNC with an image generator[37] and genetic optimization[38,39] to iteratively generate images following two serial objectives. The first objective, active throughout the entire optimization procedure, was to generate images controlling (that is, driving or suppressing) in silico univariate fMRI responses of V1 and V4 up to a threshold. Once this threshold was reached, the second objective became activated, which was to lower image complexity as measured by the images' portable network graphics (PNG) compression file size[40,41], while at the same time keeping the univariate responses above threshold. This promoted the generation of controlling images (first objective) containing only the visual features strictly necessary to align or disentangle in silico univariate fMRI responses (second objective) (Fig. 3b; the generative univariate RNC algorithm is visualized in Extended Data Fig. 5; for a fine-grained progression of images across generations, see Supplementary Fig. 4). For each neural control condition, we ran ten independent evolutions, resulting in ten genetically optimized images for each condition.

Inspection of the genetically optimized images converged with the insights previously gained by naturalistic images. The genetically optimized images driving V1 while suppressing V4 consisted of a uniform high-spatial-frequency pattern, whereas the images driving V4 while suppressing V1 consisted of multiple small object-like shapes on a uniform background. The images driving or suppressing both areas were again logical combinations of the previous cases: the images driving both areas consisted of many small object-like shapes clustered together, and the images suppressing both areas consisted of a uniform white background. The fact that all ten generated images within each neural control condition were strikingly similar to each other indicates that these controlling visual features are the ones optimally aligning or disentangling V1 and V4. Further supporting this, generating images without the image complexity constraint led to images that, albeit visually more complex, still contained the same controlling visual features (Supplementary Fig. 5).

Together, this shows that high spatial frequencies and object-like shapes are the visual features leading to unique representational content for V1 and V4 in silico univariate fMRI responses.

## RNC controls in silico multivariate fMRI responses across early- and mid-level visual cortical areas

We next used RNC to reveal the representational relationships for visual information encoded in in silico multivariate fMRI responses (that is, the population response patterns over all voxels within an area, rather than averaged voxel responses)[21,22,24–26].

To control in silico multivariate fMRI responses across areas, their response patterns must be directly comparable to each other. We thus transformed response patterns into representational similarity matrices (RSMs), capturing the representational geometry of each area in a common format[25]. For each pairwise comparison of areas, we used multivariate RNC and genetic optimization[38,39,42,43] to search, across all 73,000 NSD images, for controlling image batches that would either align or disentangle the RSMs of the two areas being compared. Alignment consisted in an image batch leading to a high representational similarity analysis (RSA)[25] correlation score (that is, Pearson's $r$) for the RSMs of the two areas. Disentanglement consisted in an image

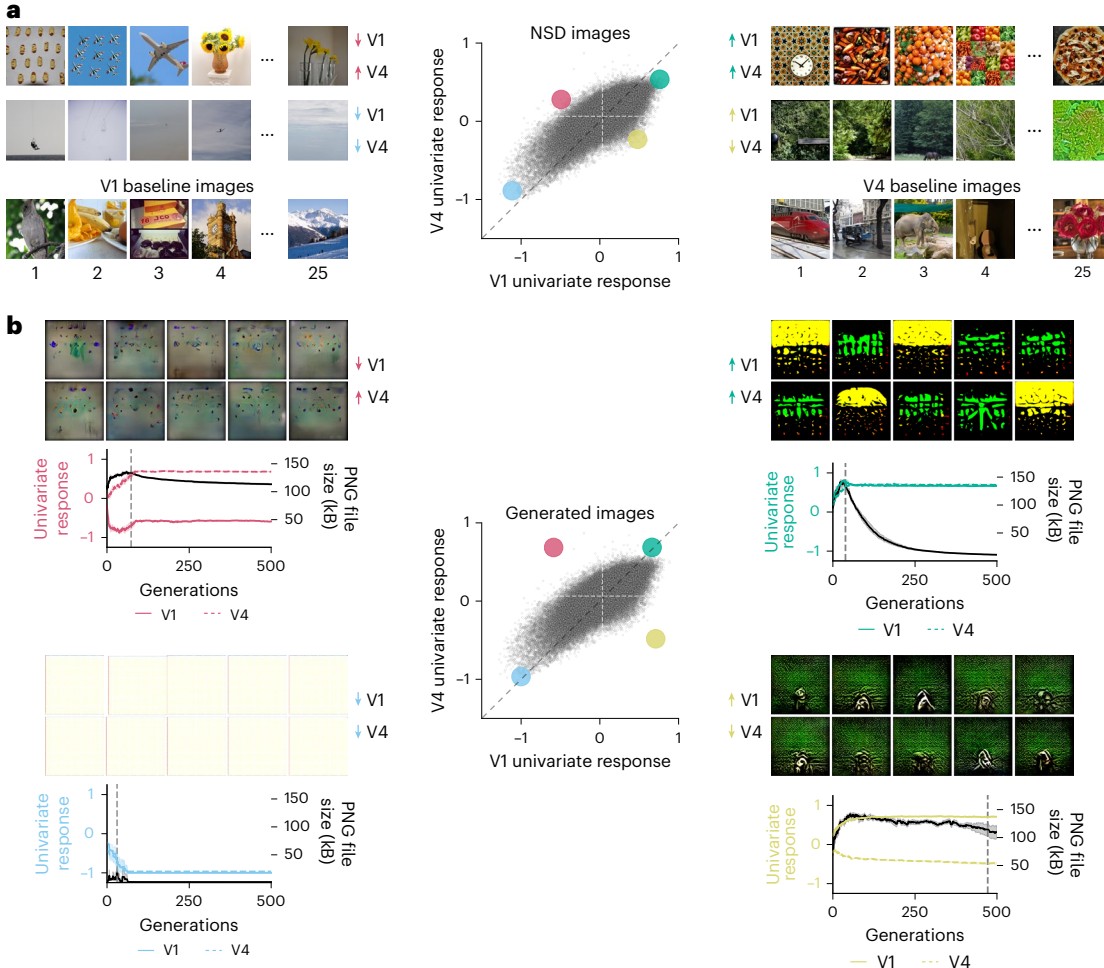

**Fig. 3 | Spatial frequency and object-like shapes determine unique representational content for V1 and V4 in silico univariate fMRI responses.** **a**, V1 versus V4 neural control scores and controlling or baseline images obtained by applying univariate RNC jointly on the in silico fMRI responses of all 8 participants for the 73,000 NSD images. Photos from the COCO image dataset/Flickr[85]. **b**, Results of ten independent generative univariate RNC evolutions using in silico fMRI responses averaged across all eight participants. For each neural control condition, the plots show the in silico univariate fMRI responses (represented by coloured lines) and the PNG compression file size (represented by black lines) for the best generated image of each genetic algorithm generation, averaged across the ten evolutions. The vertical dashed lines indicate the generation where the univariate response threshold is reached (also averaged across evolutions), after which the PNG compression file size starts decreasing. On top of each plot are the optimized images from the ten evolutions.

batch leading to a low absolute RSA correlation score for the RSMs of the two areas (Fig. 4a; the multivariate RNC algorithm is visualized in Extended Data Fig. 6). The results were cross-participant validated and compared with a baseline RSM defined on an image batch selected without optimization.

Through multivariate RNC, we found controlling image batches that significantly aligned or disentangled the RSMs of all pairwise comparisons of areas (within-participant permutation test, one-sided, $P < 0.05$, Benjamini–Hochberg corrected over two tests for each pairwise comparison of areas; population prevalence test, one-sided, $P < 1 \times 10^{-9}$, indicating within-participant significance in at least 7 out of 8 participants) (Fig. 4b; for the genetic optimization curves, see Supplementary Fig. 6). Thus, we successfully aligned or disentangled different areas at the multivariate response level.

Here, too, we tested whether alignment of multivariate responses decreases, and disentanglement increases, with increasing node distance across the visual processing hierarchy (Fig. 4c). The RSA scores for the disentangling and baseline images decreased as the stepwise distance between two areas increased (within-participant permutation test, one-sided, $P < 0.05$, Benjamini–Hochberg corrected over three tests; population prevalence test, one-sided, $P < 10^{-5}$, indicating

within-participant significance in at least 5 out of 8 participants), but the RSA scores for the aligning images did not decrease, probably due to a ceiling effect (Fig. 4d). Thus, the multivariate responses of areas further away from each other were more strongly disentangled.

We verified the generalizability of these representational relationships, observing quantitatively similar results when using alternative distributions of images and of encoding model training data (Supplementary Figs. 7–9).

Together, through multivariate RNC we discovered controlling images that align or disentangle the in silico multivariate fMRI responses of multiple areas, revealing that, while a large portion of representational content is shared between multivariate responses across visual areas, unique representational content increases as a function of cortical distance.

### Shared representational content for V1 and V4 in silico multivariate fMRI responses stems from similar retinotopic properties

Which visual features underlie the representational relationships captured in multivariate responses? Here, we focus on the V1 versus V4 comparison (Fig. 5a).

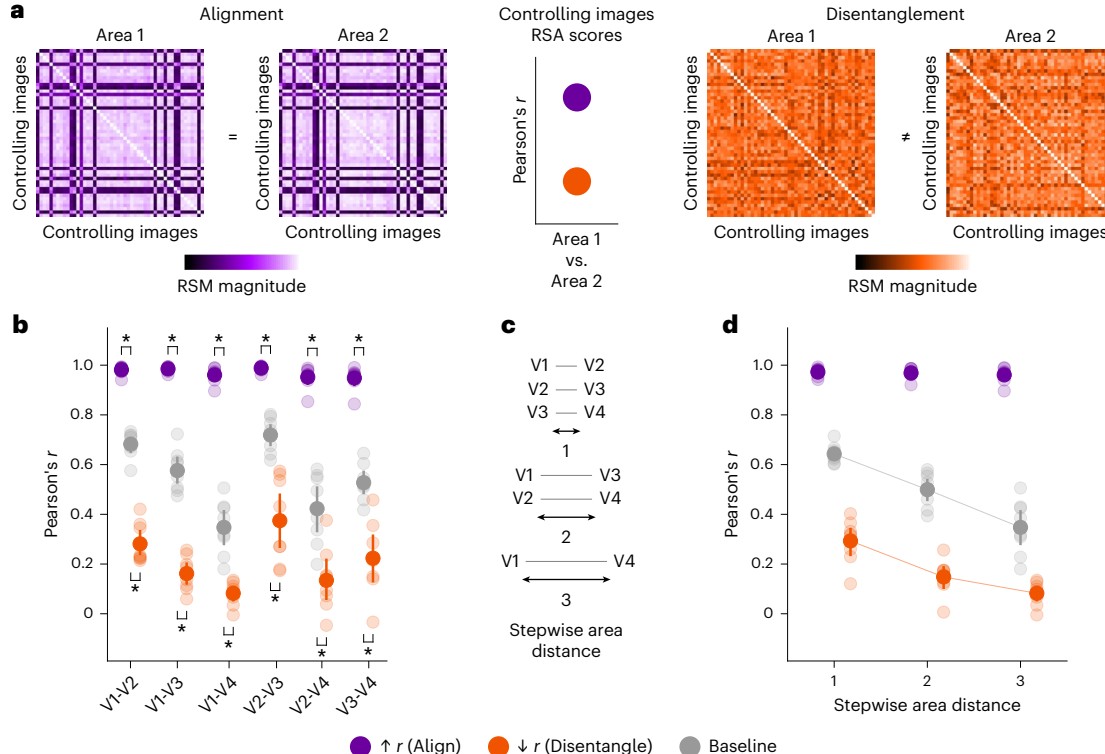

**Fig. 4 | RNC controls in silico multivariate fMRI responses across early- and mid-level visual cortical areas. a**, Multivariate RNC neural control conditions. **b**, Multivariate RNC results, consisting of RSA scores (Pearson's *r*) for each pairwise comparison of areas. Asterisks indicate neural control conditions for which the RSA scores from the controlling images are significantly higher (alignment) or lower (disentanglement) than baseline (within-participant permutation test, one-sided, $P < 0.05$, Benjamini–Hochberg corrected over two tests for each pairwise comparison of areas; population prevalence test, one-sided, $P < 1 \times 10^{-9}$, indicating within-participant significance in at least 7 out of 8 participants). **c**, Stepwise distances between areas. **d**, Multivariate RNC RSA scores, averaged across pairwise comparisons of areas with same stepwise distance. Connectors between area distances indicate a significant decreasing trend (within-participant permutation test, one-sided, $P < 0.05$, Benjamini–Hochberg corrected over three tests; population prevalence test, one-sided, $P < 1 \times 10^{-5}$, indicating within-participant significance in at least 5 out of 8 participants). In **b–d**, opaque and transparent dots represent participant-average and single-participant results, respectively. Error bars reflect 95% confidence intervals.

The aligning images often contained uniform portions (that is, the sky on their upper half), whereas the disentangling images did not, and the baseline images did but to a lesser extent (Fig. 5a). This was also the case when using alternative distributions of images and of encoding model training data (Supplementary Figs. 7–9).

To understand the effect of image properties on the multivariate RNC scores, we visually inspected the V1 and V4 RSMs in conjunction with the controlling images (Fig. 5b). For both areas, RSM entries comparing different images including the sky in their upper half indicated highly positive correlations, while RSM entries comparing images with and without the sky in the upper half indicated highly negative correlations (Fig. 5c and Extended Data Fig. 7a). This similar combination of highly positive and negative correlation RSM entries led to a high RSA correlation score for V1 and V4 and, thus, to alignment. Meanwhile, upon visual inspection, the V1 and V4 RSMs for the disentangling images contained correlation scores of lower absolute magnitude and did not reveal common visual patterns (Fig. 5b).

Combining the insights gained from inspecting controlling images and RSMs, we stipulated that retinotopic organization determines neural alignment[44]: uniform regions on a spatially constrained portion of the image will lead to suppressed responses for V1 and V4 voxels tuned to the corresponding portion of the visual field, in turn leading to aligned RSMs for the two areas.

We tested this hypothesis by comparing the V1 and V4 univariate responses of voxels tuned to the upper and lower portion of the visual field, for aligning images including uniform regions (that is, the sky) in their upper half. As predicted, for both areas, the univariate response of voxels was lower for the upper than for the lower visual field (within-participant permutation test, one-sided, $P < 0.05$, Benjamini–Hochberg corrected over two tests for each area; population prevalence test, one-sided, $P < 1 \times 10^{-5}$, indicating within-participant significance in at least 5 out of 8 participants) (Fig. 5d), explaining why RSM entries comparing different images including the sky in their upper half resulted in highly positive correlations (Extended Data Fig. 7b). We observed the opposite pattern when comparing voxel responses for aligning images not including the sky in their upper half (within-participant permutation test, one-sided, $P < 0.05$, Benjamini–Hochberg corrected over two tests for each area; population prevalence test, one-sided, $P < 1 \times 10^{-9}$, indicating within-participant significance in at least 7 out of 8 participants) (Fig. 5d), explaining why RSM entries comparing images with and without the sky in the upper half resulted in highly negative correlations (Extended Data Fig. 7b).

Together, these results point to common retinotopic properties as a source of shared representational content in V1 and V4 in silico multivariate fMRI responses.

## RNC controls in silico univariate and multivariate fMRI responses across high-level visual cortical areas

Next, we extended RNC from early- and mid- to high-level visual areas. Using NSD, we trained encoding models of high-level visual areas that play a key role in the representation of bodies (EBA[45]), faces (FFA[23]) and scenes (PPA[46]) and in visual navigation (RSC[47]) (encoding accuracies are shown in Extended Data Fig. 3a). Through RNC, we found controlling images that successfully aligned or disentangled both

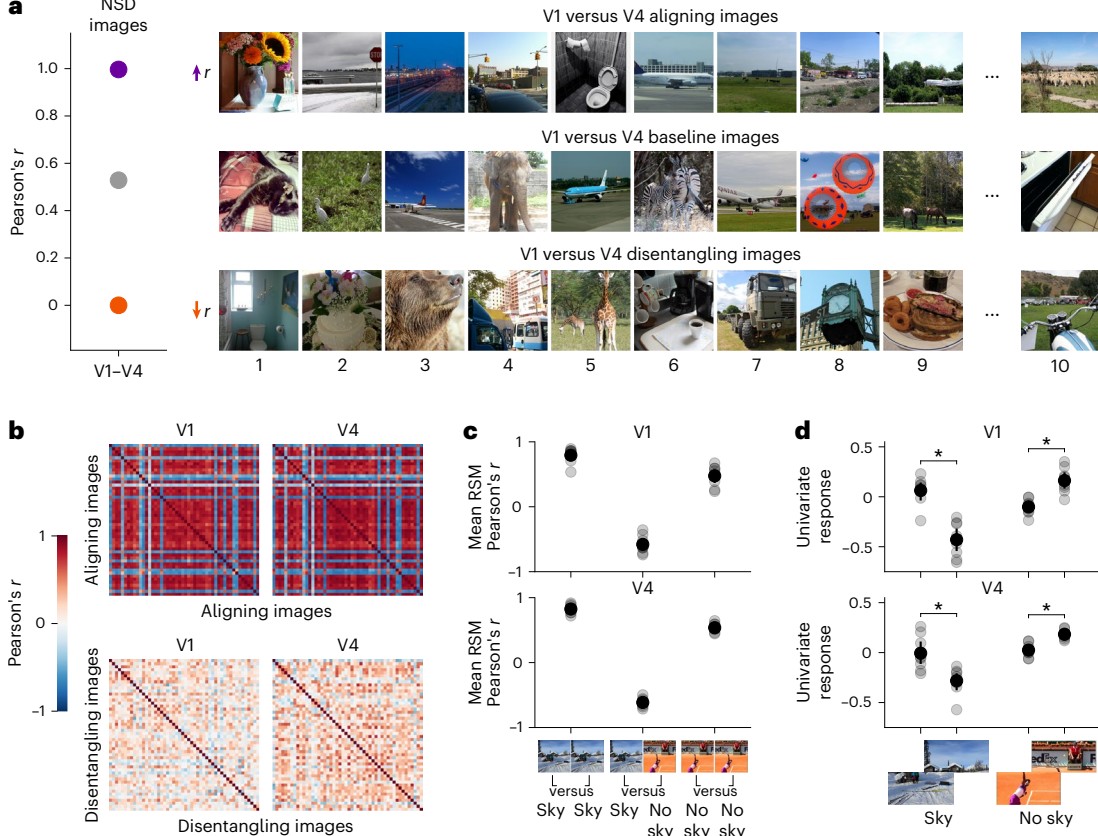

**Fig. 5 | Shared representational content for V1 and V4 in silico multivariate fMRI responses stems from similar retinotopic properties. a**, V1 versus V4 neural control scores and controlling or baseline images obtained by applying multivariate RNC jointly on the in silico fMRI responses of all 8 participants for the 73,000 NSD images. **b**, V1 and V4 participant-average RSMs for the multivariate RNC aligning and disentangling images. **c**, V1 and V4 aligning image RSM mean Pearson's *r* scores across all comparisons of two sky images, two no sky images, or sky and no sky images, for the 8 participants. **d**, V1 and V4 mean univariate response for aligning images that either contain or do not contain the sky in their upper half, divided into voxels tuned to the lower and upper part of the visual field. Asterisks indicate a significant difference between the univariate responses of voxels tuned to the lower and upper part of the visual field (within-participant permutation test, one-sided, $P < 0.05$, Benjamini–Hochberg corrected over two tests for each area; population prevalence test, one-sided, $P < 1 \times 10^{-5}$, indicating within-participant significance in at least 5 out of 8 participants). In **c** and **d**, opaque and transparent dots represent participant-average and single-participant results, respectively. Error bars reflect 95% confidence intervals. Photos from the COCO image dataset/Flickr[85].

univariate (Fig. 6a) and multivariate (Fig. 6b) in silico fMRI responses for the 73,000 NSD images generated through these encoding models (within-participant permutation test, one-sided, $P < 0.05$, Benjamini–Hochberg corrected over eight or two tests for each univariate or multivariate RNC pairwise comparison of areas, respectively; population prevalence test, one-sided, $P < 0.01$, indicating within-participant significance in at least 3 out of 8 participants) (RNC results for interactions between early-, mid- and high-level visual areas are shown in Supplementary Figs. 10–12). Thus, we successfully aligned or disentangled different high-level visual cortical areas at the univariate and multivariate response levels.

EBA, FFA, PPA and RSC fall within two broader groups of categorical selectivity: animate objects (EBA and FFA) and scenes (PPA and RSC). This suggests that alignment should be higher and disentanglement lower for within-group areas than for between-group areas (Fig. 6c). We confirmed this prediction. For univariate RNC, the absolute difference between the in silico univariate fMRI responses in the control conditions and the baseline was larger for within-group areas in the case of alignment (Fig. 6d, green and blue dots) and larger for between-group areas in the case of disentanglement (Fig. 6d, yellow and red dots) (within-participant permutation test, one-sided, $P < 0.05$, Benjamini–Hochberg corrected over four tests; population prevalence test, one-sided, $P < 0.001$, indicating within-participant significance in at least 4 out of 8 participants). This indicates that

the responses of within-group areas were more aligned and less disentangled, compared with between-group areas. Strengthening this finding, the univariate responses of within-group areas were strongly correlated, whereas the responses of between-group areas were anticorrelated (Fig. 6a, lower triangle of the results matrix). Similarly, for multivariate RNC, the RSA scores for the aligning, disentangling and baseline images were higher for within-group areas (Fig. 6e) (within-participant permutation test, one-sided, $P < 0.05$, Benjamini–Hochberg corrected over three tests; population prevalence test, one-sided, $P < 0.01$, indicating within-participant significance in at least 3 out of 8 participants). We observed quantitatively similar results when using alternative distributions of images (Supplementary Figs. 13 and 14).

Together, through RNC we discovered controlling images that align or disentangle the in silico univariate and multivariate fMRI responses of high-level visual areas. This demonstrated RNC's applicability across the visual cortical network and revealed that shared representational content is higher and unique representational content is lower for high-level visual areas with similar categorical selectivity.

## Representational relationships between visual areas adaptively vary around a typical network-level configuration

Vision is enabled by a complex interconnected network of cortical areas that jointly represent visual information. Thus, we next moved

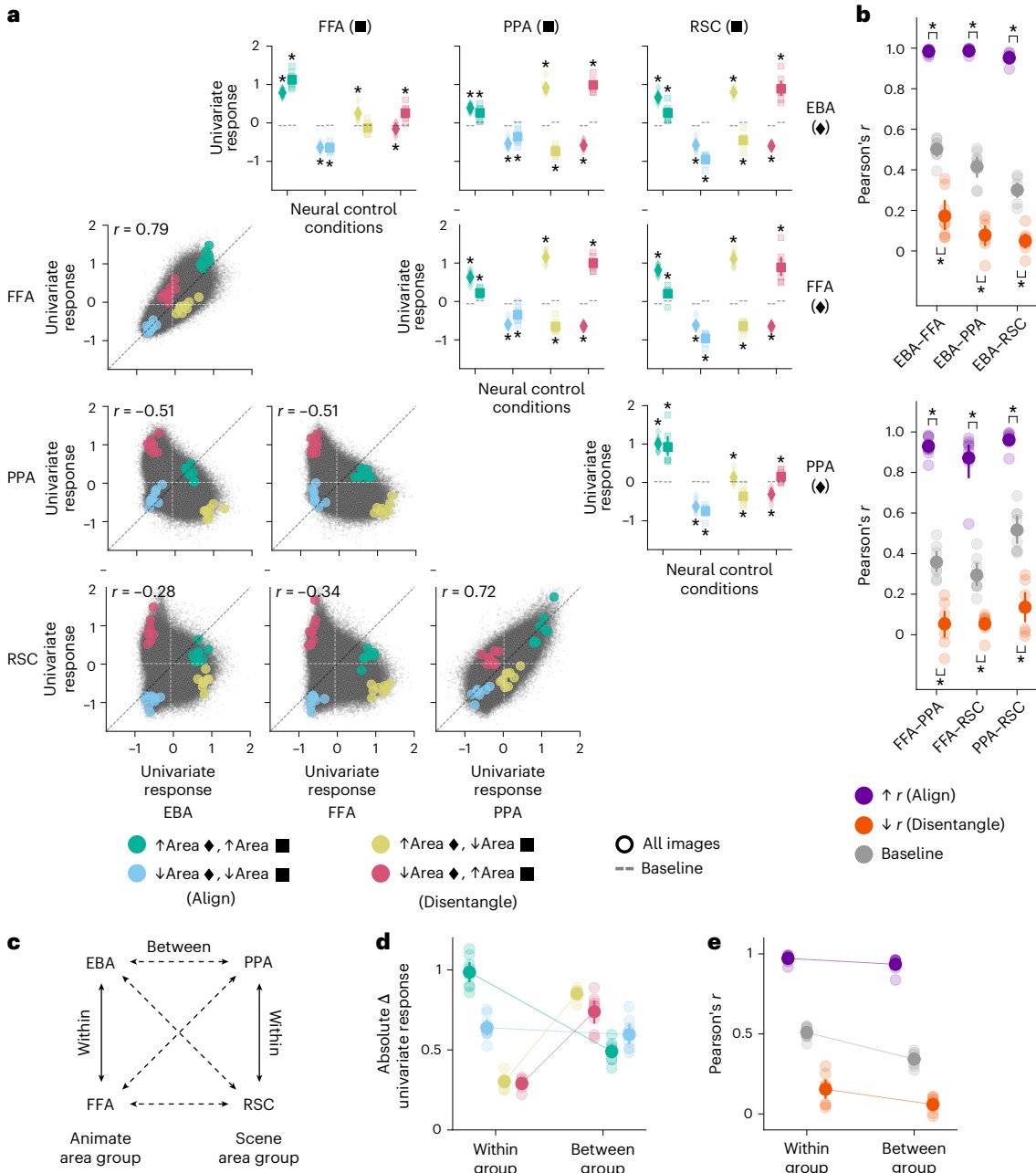

**Fig. 6 | RNC controls in silico univariate and multivariate fMRI responses across high-level visual cortical areas. a**, Univariate RNC results for each pairwise comparison of areas, embedded in a four-by-four matrix. The upper triangle of the results matrix shows the univariate responses for the controlling images against the baseline. Diamonds and squares indicate the univariate responses of the areas indexed by the rows and columns of the results matrix, respectively. Asterisks indicate neural control conditions for which the in silico univariate fMRI responses for the controlling images are significantly different from baseline (within-participant permutation test, one-sided, $P < 0.05$, Benjamini–Hochberg corrected over eight tests for each pairwise comparison of areas; population prevalence test, one-sided, $P < 0.01$, indicating within-participant significance in at least 3 out of 8 participants). The lower triangle of the results matrix shows the univariate response image manifolds. Coloured dots indicate in silico univariate fMRI responses averaged across the controlling images of each neural control condition, and small black points indicate in silico univariate fMRI responses of all participants for all 73,000 NSD images. Vertical and horizontal dashed lines indicate the participant-average univariate response baseline for each area. **b**, Multivariate RNC results, consisting of RSA scores (Pearson's $r$) for each pairwise comparison of areas. Asterisks indicate neural control conditions for which the RSA scores from the controlling images

are significantly higher (alignment) or lower (disentanglement) than baseline (within-participant permutation test, one-sided, $P < 0.05$, Benjamini–Hochberg corrected over two tests for each pairwise comparison of areas; population prevalence test, one-sided, $P < 1 × 10^{-9}$, indicating within-participant significance in at least 7 out of 8 participants). **c**, Categorical selectivity groups. Solid and dashed lines represent within- and between-group area comparisons, respectively. **d**, Absolute difference between controlling and baseline image univariate responses, averaged across within- or between-group area comparisons. Connectors indicate significant differences (within-participant permutation test, one-sided, $P < 0.05$, Benjamini–Hochberg corrected over four tests; population prevalence test, one-sided, $P < 0.001$, indicating within-participant significance in at least 4 out of 8 participants). **e**, Multivariate RNC RSA scores, averaged across within- or between-group area comparisons. Connectors indicate significant differences (within-participant permutation test, one-sided, $P < 0.05$, Benjamini–Hochberg corrected over three tests; population prevalence test, one-sided, $P < 0.01$, indicating within-participant significance in at least 3 out of 8 participants). In **a**–**e**, opaque and transparent diamonds, squares and dots represent participant-average and single-participant results, respectively. Error bars reflect 95% confidence intervals.

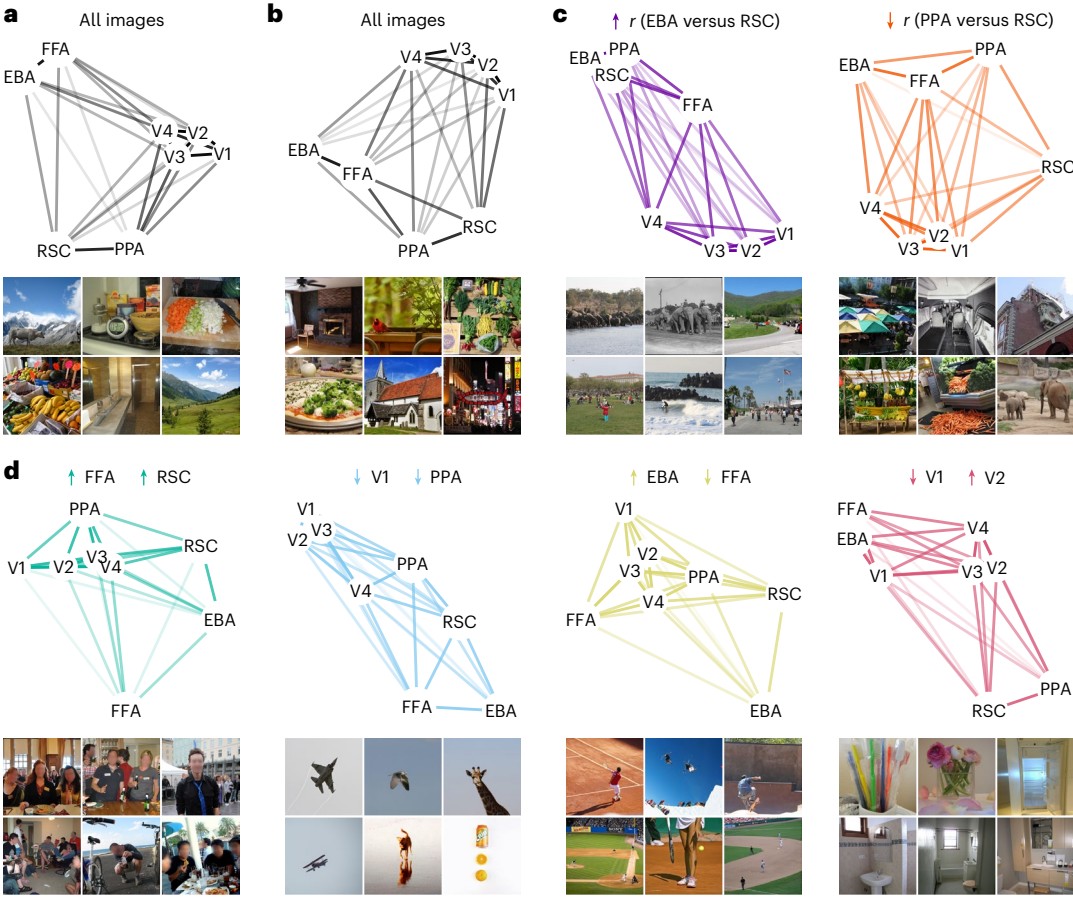

**Fig. 7 | Representational relationships between visual areas adaptively vary around a typical network-level configuration. a**, MDS embeddings of the in silico univariate fMRI responses of early-, mid- and high-level visual areas for all 73,000 NSD images, indicating the typical representational relationship configuration of the visual cortical network. **b**, MDS embeddings of the in silico multivariate fMRI responses for all 73,000 NSD images, indicating the typical representational relationship configuration of the visual cortical network. **c**, MDS embeddings of the in silico multivariate fMRI responses for the controlling images from two multivariate RNC control conditions: images aligning EBA and RSC (purple condition) or disentangling PPA and RSC (orange condition). **d**, MDS embeddings of the in silico univariate fMRI responses for

the controlling images from four univariate RNC control conditions: images driving both FFA and RSC (green condition), suppressing both V1 and PPA (blue condition), driving EBA while suppressing FFA (yellow condition) or suppressing V1 while driving V2 (red condition). In **a**–**d**, the opaqueness of the lines connecting each area reflects the proximity of these areas in two-dimensional embedding space (more opaque lines indicate higher proximity; we computed the opaqueness independently for each MDS plot, by normalizing the corresponding pairwise area distances between 0.1 (almost transparent) and 1 (fully opaque)). A higher proximity between two areas indicates a stronger resemblance of their representational content. Six representative images are shown for each MDS analysis. Photos from the COCO image dataset/Flickr[85].

to the network-level visualization of the representational relationships discovered for individual pairwise comparisons of areas.

We first asked what the typical representational relationship configuration of areas within the visual cortical network is. Using multidimensional scaling (MDS)[48], we reduced the dimensionality of the participant-average in silico univariate or multivariate fMRI responses for all 73,000 NSD images of early-, mid- and high-level visual areas (V1, V2, V3, V4, EBA, FFA, PPA and RSC). This resulted in two-dimensional embeddings where a higher proximity between two areas reflects a stronger resemblance of their representational content. For both univariate (Fig. 7a) and multivariate (Fig. 7b) in silico fMRI responses, these embeddings revealed three network-level properties that together defined a common, typical network-level configuration: (1) that early- and mid-level visual areas' proximity in embedding space mirrors their cortical distance, further supporting that unique representational content increases as a function of cortical distance; (2) that high-level visual areas cluster on the basis of categorical selectivity for animate objects (EBA and FFA) and scenes (PPA and RSC), further supporting that shared representational content is higher and unique representational content lower for areas with similar categorical selectivity; and (3) that

early- and mid-level visual areas are closer to each other than they are to high-level visual areas, indicating an analogous relationship for their representational content.

Visual stimulation continuously alters the representational content of visual areas, leading to reconfigurations of these areas' representational relationships. Are these reconfigurations rigidly preserving the typical representational relationship configuration properties revealed above, or is the visual cortical network flexibly spanning any configuration? To assess this, we applied MDS on the in silico fMRI responses for the aligning or disentangling images selected through RNC. The controlling images led to representational relationship configurations that negated one and two, but not all three, properties, indicating that representational relationships adaptively vary around their typical configuration (Fig. 7c,d). As an illustrative example, the controlling images suppressing V1 while driving V2's univariate responses moved V1 closer to V3 than to V2, thus negating the first property, and moved V1 closer to EBA and FFA than to the other early- and mid-level visual areas, thus negating the third property (Fig. 7d, red condition; the representational relationship configurations for other RNC pairwise area comparisons and control conditions are shown in Supplementary Figs. 15–18).

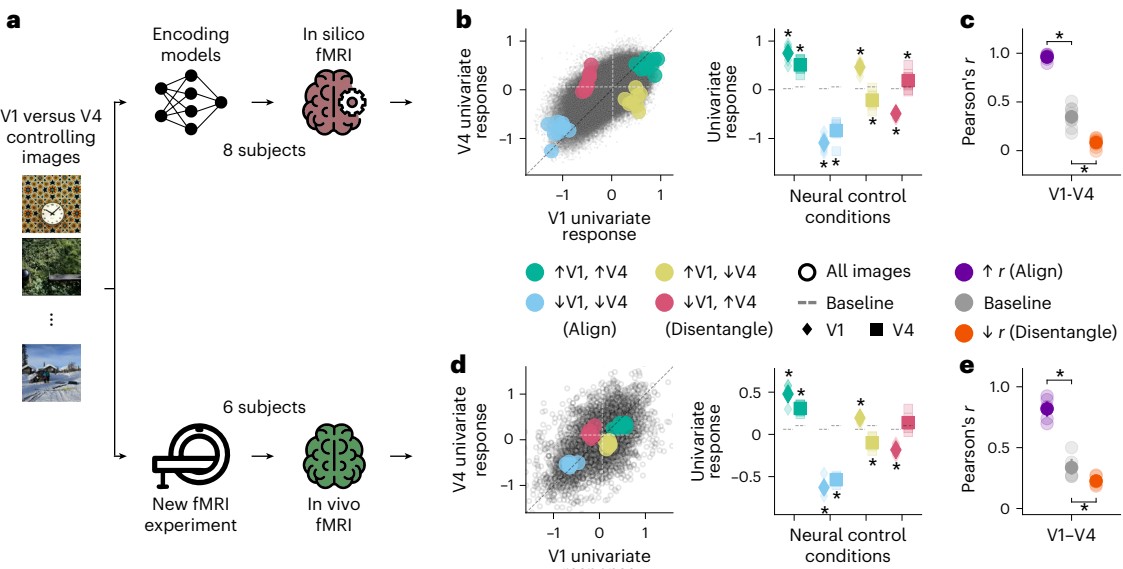

**Fig. 8 | In-silico-discovered controlling images control in vivo fMRI responses of independent participants. a,** We tested whether the images controlling univariate and multivariate in silico fMRI responses for the V1 versus V4 comparison generalized their control effect to in vivo fMRI responses of six new, independent participants. Photos from the COCO image dataset/Flickr[85]. **b,** Univariate RNC in silico results. **c,** Multivariate RNC in silico results. **d,** Univariate RNC in vivo results. Small black points indicate in vivo fMRI responses of all participants for all V1 versus V4 univariate RNC controlling images. **e,** Multivariate RNC in vivo results. In **b**–**e**, asterisks indicate a significant effect of the controlling images with respect to baseline (within-participant permutation test, one-sided, $P < 0.05$, Benjamini–Hochberg corrected over eight or two tests for univariate or multivariate RNC respectively; population prevalence test, one-sided, $P < 1 \times 10^{-4}$, indicating within-participant significance in at least 4 out of 8 participants for the in silico results or 4 out of 6 participants for the in vivo results). Opaque and transparent dots, diamonds and squares represent participant-average and single-participant results, respectively. Error bars reflect 95% confidence intervals.

Together, these results provide a unified picture of how visual areas jointly represent the world as an interconnected network. We showed that representational relationships between visual areas adaptively vary around a typical network-level configuration and that RNC enables the exploration of the state space of possible network configurations.

### In-silico-discovered controlling images control in vivo fMRI responses of independent participants

In silico discoveries empower and accelerate empirical research but do not replace it; these discoveries need to be validated empirically. Thus, we complemented the above results for areas V1 and V4—which were cross-participant validated on in silico fMRI responses—with empirical validation on in vivo fMRI responses.

We conducted an fMRI experiment where we presented an independent set of participants ($n = 6$) with the univariate and multivariate RNC controlling images for the V1 versus V4 comparison (Fig. 8a; for the experimental design, see Extended Data Fig. 8a). We defined V1 and V4 in the new participants using population receptive field (pRF) mapping[49] (an illustration of the pRF experiment and the V1/V4 delineations are shown in Extended Data Fig. 8b,c). We found that the controlling images aligned or disentangled both univariate (Fig. 8d) and multivariate (Fig. 8e) responses of V1 and V4 in these new participants, (within-participant permutation test, one-sided, $P < 0.05$, Benjamini–Hochberg corrected over eight or two tests for univariate or multivariate RNC, respectively; population prevalence test, one-sided, $P < 1 \times 10^{-4}$, indicating within-participant significance in at least 4 out of 6 participants), except for V4's univariate response in the univariate neural control condition suppressing V1 while driving V4.

The successful generalization to in vivo fMRI responses closed the empirical cycle, confirming the in silico discoveries and validating RNC as a new exploratory neural control method for investigating representational relationships.

## Discussion

We investigated representational relationships between early-, mid- and high-level visual areas of the human cortex using RNC. Through RNC, we extensively explored in silico fMRI responses for a vast collection of naturalistic images, finding controlling images that aligned or disentangled univariate and multivariate in silico fMRI responses across areas, thus indicating shared or unique representational content. Closing the empirical cycle, we validated the in silico discoveries on in vivo fMRI responses by presenting the controlling images to independent participants.

Representations are the key concept in theories of information processing in visual cortex[9,11–13,50,51], and because visual processing is supported by the concerted effort of multiple areas[1–8], understanding how visual cortex works requires a joint investigation of the representational relationships between such areas. Thus, RNC invites a perspective shift from asking 'What does each area represent?' to asking 'What is the relationship between representations in different areas?'. RNC answers the latter question by applying neural control[29,38,39,52–59] jointly to multiple cortical areas, thus determining the causal role of specific visual input to their representational relationships[53,60]. Hence, RNC extends existing anatomical[2] and functional[17] connectivity research assessing the brain as a complex interconnected network with the concept of representation. Representations being the material of information processing and transfer[9,13,61], our results promote the understanding of how perception and cognition emerge from the joint interaction of the representational content of multiple cortical areas.

Through RNC, we successfully controlled univariate and multivariate in silico fMRI responses jointly for areas across the visual cortical network. This confirmed representational properties known from investigating these areas in isolation, for example, that V1 is tuned to high spatial frequencies[34], that V4 is tuned to object-like shapes[36] and that both areas are tuned to topological image properties[44]. Jointly controlling multiple areas additionally revealed network-level properties

of how these areas represent the visual world that are not available from investigating them in isolation. First, that unique representational content of early- and mid-level visual areas increases as a function of cortical distance. This representational pattern probably reflects the decrease in anatomical connectivity with increasing distance between areas[5], as well as other gradual changes along the visual hierarchy such as increasing receptive field sizes[62] and increasingly complex functional specialization[1]. Second, that shared representational content is higher and unique representational content is lower for high-level visual areas with similar categorical selectivity. The successful disentanglement of high-level visual areas with similar categorical selectivity (for example, PPA and RSC[63]) demonstrates that RNC is sensitive to fine-grained representational differences. Furthermore, the successful control of RSC—an area that, beyond visual navigation, is also involved in memory and planning[64]—suggests that RNC could also reveal representational relationships properties of more anterior areas that contribute to visual processing, such as ventrolateral prefrontal cortex[65]. Third, that early- and mid-level visual areas are more similar to each other than they are to high-level visual areas in terms of representational content. Finally, that representational relationships between visual areas adaptively vary around a typical configuration defined by the previous three properties. Using RNC to jointly control all areas within the visual cortical network—as opposed to pairwise comparisons of areas, as done here—might reveal new representational relationship configurations and properties. For instance, it might uncover configurations that maximize the representational alignment of two areas and their disentanglement from all other areas, or representational relationships that diverge considerably from the typical configuration.

RNC builds on recent innovations in neural control, a paradigm used to find controlling stimuli that elicit a neural response state of interest. To increase the solution space and allow rapid exploration, the controlling stimuli are found using large amounts of in silico neural responses generated by encoding models[29,52–54,57]. Next, to ensure that the controlling stimuli truly elicit the neural response state of interest, these stimuli are empirically validated on in vivo neural responses[53,54,57]. Finally, to elicit more complex neural response states of interest, multiple neurons or areas are jointly controlled[29,53]. Building on these innovations, RNC extends neural control research in two ways. First, RNC uses neural control to enable a network-level characterization of the representational relationships between visual cortical areas. Second, to investigate representational relationships at multiple brain response levels, RNC jointly applies neural control on univariate and multivariate neural responses. We found that the visual features aligning or disentangling the univariate responses of V1 and V4 (that is, spatial frequency and object-like shapes) are different from the ones aligning and disentangling their multivariate responses (that is, topological image properties). Thus, the univariate and multivariate response levels captured complementary aspects of representational relationships between areas, suggesting that visual cortex multiplexes diverse neural codes for visual information processing[9,61,66] and, in turn, encouraging the integrated analysis of diverse neural response levels.

RNC embodies a research paradigm combining the advantages of in silico neural response exploration with empirical validation of findings on in vivo neural responses[53,54,57,67,68]. In silico exploration takes the power of recently emerging large-scale in vivo neural datasets[27,33,69–71] to the next level. In vivo neural responses are available in limited numbers and are expensive and slow to acquire. By contrast, after initial training on an in vivo dataset, encoding models cheaply and quickly generate in silico neural responses to any amount and type of stimuli, thus allowing large upscaling of the solution spaces on which to explore and test scientific hypotheses. Moreover, encoding models generate in silico neural responses that are less affected by noise compared with in vivo responses, thus reducing the effect of noise on results[19,29]. Together, this allows exploration of a much larger amount of stimuli and corresponding neural responses, effectively reducing the risk of suboptimal

or biased findings deriving from smaller and noisier samples or from experimenters' hand-picked stimuli.

The key limitation of RNC lies in the component that empowers it: the encoding models generating the in silico neural responses do not predict all explainable neural signals, and their predictions generalize imperfectly beyond the distribution of the visual data they were trained on. Thus, to ascertain the validity of our findings, we showed that our encoding models achieved high prediction accuracies when tested both in and out of distribution, and also that RNC's solutions are replicated when using alternative distributions of images and of encoding model training data. Furthermore, the current push in the development of more accurate and robust visual encoding models[72–75] using large in vivo datasets[27,33,69–71] that also include out-of-distribution components[28,33,75] promises increasingly accurate in silico neural responses, in turn increasing the reliability of findings from experimentation on computer-generated brain data.

The assumption of aligned or disentangled neural responses indicating shared or unique representational content is not a given. For example, for early- and mid-level visual areas, we found that the assumption was correct for all multivariate RNC control conditions and for the univariate RNC disentanglement conditions, but not for the univariate RNC alignment conditions. There, the controlling images aligning both V1 and V4 univariate fMRI responses consisted of the logical combination of the visual features disentangling them (that is, high spatial frequency and objects), rather than of features for which the two areas are not disentangled, as would be the case if the assumption held. This might be why the absolute distance between the univariate responses in the alignment control condition driving both areas and the baseline did not decrease with cortical distance (Fig. 2d, green dots). Together, this highlights a way to rigorously test the assumptions of RNC. Furthermore, finding the assumption to be negated is in itself scientifically interesting, as it reveals how either shared or unique representational content can lead to aligned univariate responses.

When one or few of the multiple stimulus features dominate the fMRI response magnitude, RNC primarily exploits these few features to align or disentangle the fMRI responses of two areas, resulting in images with few salient controlling visual features that are easier to interpret. As an example, the V1 versus V4 controlling images from the multivariate RNC alignment condition contained easily interpretable visual patterns, which indicated that the shared representational content of these two areas' multivariate fMRI responses is dominated by image topological properties. When no stimulus feature dominates the fMRI response magnitude, RNC instead exploits multiple features, resulting in images with multiple controlling visual features that complicate interpretation. As an example, the multivariate RNC disentangling images for the V1 versus V4 comparison did not reveal common patterns upon visual inspection, which suggests that the unique representational content of these two areas' multivariate fMRI responses is similarly determined by multiple non-dominating visual features. Determining the non-dominating visual features might be possible through RNC variants that isolate each of them, for example, by using parameterized artificial stimuli for targeted hypothesis testing.

Feedforward and feedback connections both shape the representational content of visual areas within the cortical visual hierarchy[5,76–78]. Therefore, beyond feedforward processing[12,50], shared and unique representational content might also be influenced by visual stimulation that either promotes (for example, with challenging stimuli[79,80]) or suppresses (for example, with visual masks[81–83]) feedback from higher areas to lower areas. Applying RNC on time-resolved magneto/electroencephalography (M/EEG), electrocorticography (ECoG) or electrophysiology data, or on cortical-layer resolved data[84], is a promising avenue to isolate the respective influence of feedforward and feedback signalling on representational relationships.

In summary, using RNC, we uncovered representational relationships between areas across the human visual cortex. This demonstrates

the power of in silico exploration combined with in vivo validation to reveal how human cortical areas, at the level of networks, collectively represent the visual world.

## Methods

### Encoding models

The trained fMRI encoding models used to generate the in silico fMRI responses are available as part of the Brain Encoding Response Generator (BERG) (https://github.com/gifale95/BERG). Below we describe how these models were trained and tested.

**Data.** The encoding models were trained on the NSD[27], a large-scale dataset of 7 T fMRI responses from 8 participants who each viewed up to 10,000 distinct colour images of natural scenes from the Common Objects in Context (COCO) dataset[85]. Out of these 10,000 images, 9,000 were participant unique (that is, seen only by individual participants) and 1,000 were shared (that is, seen by all participants). Each image was presented up to three times, for a maximum of 30,000 trials per participant.

The fMRI data consisted of NSD's prepared fMRI responses in participant-native volume space ('func1pt8mm') from betas version 3 ('betas_fithrf_GLMdenoise_RR'). To reduce session-specific noise, the responses of each voxel were $z$-scored across all trials of every data acquisition session. For model building, only voxels falling within areas V1, V2, V3, V4, EBA, FFA, PPA and RSC were selected (using the area definitions provided by NSD).

To prevent results being biased towards noisy voxels (that is, voxels with low stimulus-related signal), for all subsequent analyses we used only in silico fMRI responses for voxels with a noise ceiling signal-to-noise ratio (NCSNR) above 0.5. The NCSNR is a measure of stimulus-related signal in the fMRI responses, and its calculation is detailed below. We based the voxel selection on the NCSNR scores provided by the NSD data release. For the amount of retained voxels, see Supplementary Tables 1 and 2.

For each participant, the images and corresponding fMRI responses were split into training, validation and testing partitions. The training partition consisted of the 9,000 participant-unique images. The testing partition consisted of the 515/1,000 shared images that were presented to all participants three times (to maximize the reliability of the data on which the models were tested). The validation partition consisted of the remaining 485/1,000 shared images that were not presented to all participants three times.

**Model architecture.** Each encoding model predicted fMRI responses for multiple voxels (that is, all voxels of a specific participant and area) and consisted of two components: a feature extractor shared across all voxels, and one projection head for each voxel (Extended Data Fig. 1a).

The feature extractor is a multilayer feedforward convolutional neural network called GNet[27] (Extended Data Fig. 1b). Giving an image as input to GNet activates its layers, resulting in multiple features maps (that is, GNet's representations of this image; Extended Data Fig. 1a). The feature extractor's weights are fully learned during model training, based on the joint loss of all voxels.

The projection head of each voxel is a feature-weighted receptive field that combines a spatial pooling field and feature weights[27,86]. The spatial pooling field determines the region of visual space (that is, the GNet feature space that, due to the convolutional operations, preserves the topology of visual space of the input images) that drives voxel activity. After GNet's feature maps are spatially pooled, they are linearly combined by the feature weights, resulting in the voxel response prediction (Extended Data Fig. 1a). Both the spatial pooling field and feature weights are learned during model training, independently for each voxel. This allows the empirical determination of the optimal contribution of each model feature to each voxel based on the training data—for example, by learning the optimal hierarchical

correspondence configuration between model layers and visual cortical areas[87,88].

**Model training.** Separate encoding models were trained for each NSD participant and area (V1, V2, V3, V4, EBA, FFA, PPA and RSC), resulting in 8 participants × 8 areas = 64 encoding models. To reduce spurious statistical dependencies between the in silico (that is, model-generated) fMRI responses from models of different participants and areas, each model was trained starting from a different random initialization[89].

Given an input image, the encoding model's objective was to minimize the mean squared error between the predicted fMRI responses (for all voxels of a given participant and area) and the corresponding target fMRI responses. During training, the mean squared error loss was backpropagated and the model weights updated (Extended Data Fig. 2a). At each backpropagation step, the projection head weights were only optimized on the basis of the loss of their corresponding voxel, whereas the feature extractor weights were optimized on the basis of the loss combined over all voxels. The models were trained using single (that is, not averaged) NSD trials.

The encoding models' weights were optimized on 75 epochs of the training data partition using batch sizes of 128 images and Pytorch's[90] Adam optimizer with a learning rate of 0.001, a weight decay term of 0, betas coefficients of (0.9, 0.999) and an eps term of $1 \times 10^{-8}$. The final model weights were taken from the epoch that achieved the lowest loss between the predicted and target fMRI responses, using the validation data partition.

**Model testing.** We used the trained models to generate in silico fMRI responses for the test data partition images and compared them with the corresponding target responses, averaged across the three trials, using Pearson's correlation. We computed the correlation independently for each voxel across the 515 test images, set negative correlation coefficients to zero and squared the resulting scores to obtain $r^2$, the total variance explained by the models. We then divided the $r^2$ score of each voxel with that voxel's noise ceiling, resulting in the noise-ceiling-normalized explained variance, a measure that quantifies the portion of the explainable variance that had been accounted for by the models, given the noise in the data. We tested the models independently for each voxel and participant, and then averaged the results across voxels belonging to the same participant and area (Extended Data Figs. 2b and 3a).

Following the same procedure, we additionally tested the encoding models' out-of-distribution generalization performance on NSD-synthetic[28]. NSD-synthetic consists of an additional scan session from the same eight NSD participants, during which fMRI responses were measured to 284 carefully controlled synthetic (non-naturalistic) stimuli. We used NSD-synthetic's prepared fMRI responses in participant-native volume space ('func1pt8mm') from betas version 3 ('nsdsyntheticbetas_fithrf_GLMdenoise_RR').

**Noise analysis.** Because neural noise is not predictable from the stimulus images, encoding models model the signal- and not noise-related variability of the neural response, thus resulting in silico fMRI responses less affected by noise compared to in vivo responses[19,29]. To establish this empirically, we compared the noise of the in silico fMRI responses with the noise of the in vivo fMRI responses from the NSD, by comparing how much variance these two data types explained for a third, independent split of the in vivo NSD responses. Because the in silico fMRI responses did not capture all signal variance in the NSD responses (Extended Data Figs. 2b and 3a), the in silico fMRI responses explaining more variance than the in vivo NSD responses would be indicative of the former being less affected by noise. We carried out the comparison through three sets of predictions, using the in silico and the in vivo NSD fMRI responses for the 515 test images (Extended Data Fig. 2c) and the same noise-ceiling-normalized explained variance metric described

in the 'Model testing' section. Each prediction involved explaining in vivo single response trials from the NSD experiment with a different predictor. In the first set of predictions, the predictor consisted of one of the two remaining in vivo NSD experiment response trials. We conducted six such predictions, such that each of the three in vivo NSD trials was used as the target to be explained and each of the two remaining in vivo NSD trials was used as a predictor. We then averaged the noise-ceiling-normalized explained variance scores from the six different predictions. In the second set of predictions, the predictor consisted of the average of the two remaining in vivo NSD experiment response trials. We conducted three such predictions, each time using one of the three in vivo NSD trials as the target to be explained and the average of the remaining two in vivo NSD trials as the predictor. We then averaged the noise-ceiling-normalized explained variance scores from the three different predictions. In the third set of predictions, the predictor consisted of the in silico responses from the trained encoding models. We repeated the prediction three times, each time using one of the three in vivo NSD trials as the target to be explained and the corresponding in silico responses as the predictor. We averaged the noise-ceiling-normalized explained variance scores from the three different predictions. We carried out these comparisons independently for each voxel and participant, and averaged the results across voxels belonging to the same participant and area.

**Noise ceiling derivation.** We derived the noise ceiling of each voxel from its NCSNR score, provided by the NSD and computed on $z$-scored fMRI betas. For each voxel, the NCSNR quantified the ratio of the signal standard deviation over the noise standard deviation

$$\text{NCSNR} = \frac{\hat{\sigma}_{\text{signal}}}{\hat{\sigma}_{\text{noise}}}.$$

The noise standard deviation was obtained by calculating the variance of the betas across the three presentations of each image (using the unbiased estimator that normalizes by $n - 1$ where $n$ is the sample size), averaging this variance across images and then computing the square root of the result as follows:

$$\hat{\sigma}_{\text{noise}} = \sqrt{\text{mean}\left(\beta_\sigma^2\right)},$$

where $\beta_\sigma^2$ indicates the variance across the betas obtained for a given image. Given that the variance of the $z$-scored betas is 1, the signal standard deviation was estimated as

$$\hat{\sigma}_{\text{signal}} = \sqrt{\left|1 - \hat{\sigma}_{\text{noise}}^2\right|_+},$$

where $||_+$ indicates positive half-wave rectification. Finally, we used the NCSNR scores to derive the noise ceiling (NC) of each voxel as

$$\text{NC} = 100 \times \frac{\text{NCSNR}^2}{\text{NCSNR}^2 + \frac{1}{n}},$$

where $n$ indicates the number of trials that are averaged together. We used $n = 3$ when evaluating the encoding models against the NSD test responses averaged across all three trials (Extended Data Figs. 2b and 3a), and $n = 1$ during the noise analysis, because there we considered single trials as the target to be explained (Extended Data Fig. 2c,d).

**VIR dataset.** To ascertain that RNC's solutions reflect properties of early- and mid-level visual areas, rather than biases of specific datasets, we additionally trained encoding models on the VIR dataset[33], fMRI responses of 7 participants for 3,200 naturalistic images divided into 1,200 images of objects on natural backgrounds, 1,000 images of natural scenes and 1,000 images of materials. Each of these images

was presented 5 times for 4 of the 7 participants (for a total of 16,000 trials per participant) and 2 times for the remaining 3 participants (for a total of 6,400 trials per participant). The fMRI data consisted of VIR's preprocessed fMRI responses in participant-native volume space, as provided in the data release. For model building, we selected voxels falling within areas V1, V2, V3 and V4 (using the area definitions provided by VIR). We computed the NCSNR and noise ceiling of each voxel following the procedure described above and selected for further analyses only voxels with NCSNR above 0.5. Because area V4 of VIR participant 4 did not have voxels with NCSNR above 0.5, for this participant and area we instead lowered the NCSNR threshold to 0.4 (for the amount of retained voxels, see Supplementary Table 3).

For each participant, we split the images and corresponding fMRI responses into training, validation and testing partitions. The training partition consisted of 2,500/3,200 images (900 objects, 800 scenes and 800 materials). The validation partition consisted of 350/3,200 images (150 objects, 100 scenes and 100 materials). The testing partition consisted of 350/3,200 images (150 objects, 100 scenes and 100 materials). Following the procedure described above, we used this data to train and test separate encoding models for each VIR participant and area, resulting in 7 participants × 4 areas = 28 encoding models. We additionally tested the encoding models' out-of-distribution generalization performance on fMRI responses from the same 7 VIR participants for 38 artificially created visual illusion images (the encoding accuracies are shown in Extended Data Fig. 3b).

### In silico fMRI response generation

We used the trained encoding models of each NSD participant and area to generate in silico fMRI responses for the 73,000 images from NSD (depicting complex natural scenes consisting of several or no objects appearing at different locations), as well as for the 50,000 images from the ImageNet 2012 challenge validation split[31] and for the 26,107 images from the THINGS database[32] (depicting single objects presented centrally on natural backgrounds).

We used the trained encoding models of each VIR participant and area to generate in silico fMRI responses for the 73,000 images from NSD. Because VIR does not include EBA and RSC area definitions, we did not use VIR for the RNC analyses on high-level visual areas.

### Univariate RNC

The goal of univariate RNC was to search the 73,000 NSD images for images that controlled (that is, aligned or disentangled) the in silico univariate fMRI responses—that is, averaged responses across all voxels of a given given area—of each pairwise comparison of areas.

**Univariate RNC baseline.** For each area, we randomly selected a batch of 25 images (out of all 73,000 NSD images), fed them into the given area's encoding model and averaged the corresponding in silico univariate fMRI responses across the 25 images, resulting in a single score corresponding to the mean fMRI response for that image batch. By repeating this step 1 million times, we created the univariate RNC null distribution and selected the 25 images from the batch with scores closest to the null distribution mean. The mean univariate response score across these 25 images provided the area-wise univariate response baseline against which we tested the neural control scores from the controlling images selected through univariate RNC.

**Univariate RNC algorithm.** We fed the 73,000 NSD images to the trained encoding models of two areas and averaged the resulting in silico fMRI responses across voxels, obtaining a one-dimensional univariate response vector of length 73,000 for each of the two areas. We then either summed (alignment) or subtracted (disentanglement) the univariate response vectors of the two areas and ranked the resulting sum or difference scores. Finally, we kept the 25 controlling images that yielded the highest or lowest (depending on the neural control

condition) scores and, at the same time, resulted in in silico univariate fMRI responses higher or lower than the areas' univariate response baselines by a margin of at least 0.04.

This resulted in 4 sets of 25 controlling images, each set corresponding to a different neural control condition. The controlling images from the sum vector led to two neural control conditions in which the two areas have aligned univariate responses (that is, images that either drive or suppress the responses of both areas), whereas the controlling images from the difference vector led to two neural control conditions in which the two areas have disentangled univariate responses (that is, images that drive the responses of one area while suppressing the responses of the other area, or vice versa) (the univariate RNC algorithm is visualized in Extended Data Fig. 4).

To ascertain that the resulting representational relationships reflect properties of visual processing, rather than biases of specific datasets, we additionally performed two tests. First, we applied univariate RNC on the 50,000 ImageNet or 26,107 THINGS images and corresponding in silico fMRI responses, generated through encoding models trained on NSD. Second, we applied univariate RNC on the 73,000 NSD images and corresponding in silico fMRI responses, generated through encoding models trained on VIR.

**Participant-wise cross-validation.** We used participant-wise leave-one-out cross-validation to evaluate the univariate RNC solutions (as well as the baseline) by selecting the controlling images based on the in silico univariate fMRI responses averaged across seven participants and evaluating them on the in silico univariate fMRI responses of the left-out participant. We repeated cross-validation for each unique set of seven participants, resulting in eight cross-validated solutions.

**Univariate RNC MDS analysis.** We applied MDS[48] on the in silico univariate fMRI responses of eight early-, mid- and high-level visual areas (V1, V2, V3, V4, EBA, FFA, PPA and RSC). We started by computing the in silico fMRI univariate responses of each participant and area for $N$ images (where $N$ corresponds to all 73,000 NDS images, or to the 25 controlling images from a chosen univariate RNC control condition). Next, we averaged these univariate responses across participants, resulting in an array of shape (8 areas × $N$ images). Finally, we reduced the dimensionality of this array with MDS, resulting in a reduced array of shape (8 areas × 2 dimensions). We used scikit-learn's[91] MDS implementation with n_components = 2, metric = True, n_init = 10, max_iter = 1,000, verbose = 0, eps = 0.001, n_jobs = None, dissimilarity = 'euclidean'.

**Generative univariate RNC.** Generative univariate RNC used an image generator and genetic optimization to generate stimulus images leading to aligned or disentangled in silico univariate fMRI responses for V1 and V4, while at the same time being as simple as possible, thus isolating the controlling visual features of interest.

We began by creating 1,000 random latent vectors from a standard normal distribution (each vector being 4,096-dimensional). We gave the latent vectors as input to DeePSiM[37], a pretrained generative adversarial network, which used them to generate 1,000 images, and clamped the output image pixel values to the valid red, green and blue (RGB) range [0 255]. We stored the PNG compression file sizes of these images, as well as their latent vectors, for later use during the genetic optimization.

We then fed the generated images to the V1 and V4 trained encoding models of all participants and averaged the resulting in silico fMRI responses across both voxels and participants, obtaining a one-dimensional univariate response vector of length 1,000, for both V1 and V4. We stored the univariate responses of both areas for later use during the genetic optimization. Depending on the univariate RNC neural control condition being optimized, we then either summed or subtracted the univariate response vectors of the two areas and stored these sum or difference scores.

Next, we fed the latent vectors, the PNG compression file sizes, the V1 and V4 univariate responses and the sum or difference scores to a genetic optimization algorithm[38,39], which used these inputs to create a new generation of latent vectors. Optimization consisted of two phases. At first, the objective of the genetic optimization was to create new latent vectors leading to images more likely to result in univariate responses closer to threshold level. Once the univariate response threshold was reached, the objective switched to creating new latent vectors leading to images more likely to have lower PNG compression file sizes while at the same time keeping the univariate responses above threshold.

This resulted in a new batch of 1,000 latent vectors, which we fed to the generative adversarial network for the second optimization generation, repeating the same steps. After 500 genetic optimization generations, we obtained a single image (that is, the best-performing image from the last genetic optimization generation) that optimally controlled univariate neural responses following one of four univariate RNC neural control conditions, while at the same time being as simple as possible (that is, having a low PNG compression file size). We optimized the images for the four neural control conditions independently of each other (the generative univariate RNC algorithm is visualized in Extended Data Fig. 5).

For each neural control condition we ran ten independent evolutions, each based on a different random seed. The random seed determined the initial latent vectors, as well as the new latent vectors produced by the genetic optimization, resulting in ten genetically optimized images for each control condition.

For each area, the univariate response threshold consisted in the area's univariate response baseline plus a margin of 0.6 (for control conditions driving the area's response) or −0.6 (for control conditions suppressing the area's response).

*Genetic optimization algorithm.* The genetic optimization assigned a global score to each latent vector. If a latent vector led to univariate responses below threshold level for at least one of the two areas, its global score consisted in the corresponding sum or difference score, plus a large penalty ($10^{10}$). If a latent vector led to univariate responses above threshold level for both V1 and V4, its global score consisted in the PNG compression file size of the corresponding image. Because the penalty value was constant, the global scores of several latent vectors leading to below-threshold univariate responses were ranked on the basis of the corresponding sum or difference scores of these vectors. Thus, until the threshold was reached, the latent vectors were optimized to result in better sum or difference scores. Because the sum or difference scores were based on the univariate responses, this in turn led to univariate responses progressively closer to threshold level. Furthermore, because the penalty was always larger than the PNG file sizes, the global scores of latent vectors leading to above-threshold univariate responses always ranked better than the global scores of latent vectors leading to below-threshold univariate responses. This ensured that the optimization would favour latent vectors leading to univariate responses above threshold.

We transformed the global scores of all latent vectors into probabilities through z-scoring, scaling by a factor of 0.5 and passing the resulting values through a softmax function. The genetic optimization algorithm used these probabilities to create a new generation of latent vectors, while balancing exploitation and exploration. Exploitation involved keeping (untouched) the 250 latent vectors with the highest probability scores (that is, the latent vectors leading to either univariate responses closest to threshold or lowest PNG compression file sizes). Exploration involved creating 750 new children latent vectors from recombinations between two parent latent vectors from the current generation, where the likelihood of each latent vector being a parent was determined by its probability score. The two parents contributed unevenly to any one child: 75% and 25% of the child latent vector came

from the parent latent vector with the highest and lowest probability scores, respectively. Finally, during recombination, each of the 4,096 components of a child latent vector had a 0.25 probability of being mutated, with mutations drawn from a 0-centred Gaussian with a standard deviation of 0.75.

## Multivariate RNC

The goal of multivariate RNC was to search the 73,000 NSD images for images that controlled (that is, aligned or disentangled) the in silico multivariate fMRI responses—that is, the population response pattern of all voxels of a given area—of each pairwise comparison of areas.

**Multivariate RNC baseline.** For each pairwise comparison of areas, we randomly selected a batch of 50 images (out of all 73,000 NSD images), used the encoding models to generate the corresponding in silico fMRI responses, transformed these in silico responses into RSMs and used RSA[25] to compare the RSMs of the two areas (using Pearson's correlation), resulting in one score for the image batch. By repeating this step 1 million times, we created the multivariate RNC null distribution and selected the 50 images from the batch with scores closest to the null distribution mean. The RSA score of these 50 images provided the baseline against which we tested the neural control scores from multivariate RNC.

**Multivariate RNC algorithm.** The multivariate RNC was based on a genetic optimization[38,39,42,43] that, through 2,000 generations, selected images that best aligned or disentangled the in silico multivariate fMRI responses.

We started by creating 2,400 random batches of 50 images from the 73,000 NSD images, with no repeating image within each batch. We fed these image batches to the trained encoding models of two given areas and transformed the resulting in silico fMRI responses into RSMs[25], resulting in one 50 × 50 image RSM for each of the 2,400 image batches and each of the two areas. We then compared the RSMs of each image batch between the two areas using Pearson's correlation, obtained one correlation score (r) for each image batch and ranked these correlation scores. To align the two areas, we kept the 200 image batches with the highest correlation scores (that is, images most similarly represented by the two areas), whereas to disentangle them, we kept the 200 image batches with lowest absolute correlation scores (that is, images most differently represented by the two areas). Finally, we used these 200 highest- and lowest-ranked image batches as input to a genetic optimization algorithm, which used them to create 2,400 image batches, while balancing exploitation and exploration. Exploitation involved creating five mutated versions for each of the 200 image batches. In each version, a different number of images (1, 5, 12, 25 and 38) was randomly replaced with other images out of the 73,000 NSD images, while ensuring that no image repeated within the same batch. This increased the image batches to 1,200 (200 best batches + 200 best batches × 5 mutated versions = 1,200 batches). Exploration involved creating another 1,200 new random batches that, together with the 1,200 batches from the exploitation step, amounted to 2,400 batches of 50 images. During the second optimization generation, we once again fed these 2,400 image batches to the encoding models and repeated the same steps.

We ran 2,000 genetic optimization generations and selected the best-performing image batch from the last generation. This resulted in one of two sets of 50 controlling images, each set corresponding to a different neural control condition (the image batches from the two neural control conditions were optimized independently of each other). The controlling images from the ranked correlation vector led to an alignment of multivariate responses in the two areas (that is, images leading to high Pearson's r scores for the two areas), whereas the controlling images from the absolute ranked correlation vector led to a disentanglement of multivariate responses in the two areas

(that is, images leading to low absolute Pearson's r scores for the two areas) (the multivariate RNC algorithm is visualized in Extended Data Fig. 6).

To ascertain that the resulting representational relationships reflect properties of visual processing, rather than biases of specific datasets, we additionally performed two tests. First, we applied multivariate RNC on the 50,000 ImageNet or 26,107 THINGS images and corresponding in silico fMRI responses, generated through encoding models trained on NSD. Second, we applied multivariate RNC on the 73,000 NSD images and corresponding in silico fMRI responses, generated through encoding models trained on VIR.

**Participant-wise cross-validation.** We used participant-wise leave-one-out cross-validation to evaluate the multivariate RNC solutions (as well as the baseline) by selecting the controlling images based on the in silico fMRI RSMs averaged across seven participants and evaluating them on the in silico fMRI RSM of the left-out participant. We repeated cross-validation for each unique set of seven participants, resulting in eight cross-validated solutions.

**Multivariate RNC MDS analysis.** We applied MDS[48] on the in silico multivariate fMRI responses of eight early-, mid- and high-level visual areas (V1, V2, V3, V4, EBA, FFA, PPA and RSC). We started by computing the in silico fMRI RSMs of each participant and area for N images (where N corresponds to all 73,000 NDS images, or to the 50 controlling images from a chosen multivariate RNC control condition). Next, we averaged these RSMs across participants and vectorized the lower triangle entries of the averaged RSMs, resulting in an array of shape (8 areas × M RSM lower triangle entries). Finally, we reduced the dimensionality of this array with MDS, resulting in a reduced array of shape (8 areas × 2 dimensions). We used scikit-learn's[91] MDS implementation with n_components = 2, metric = True, n_init = 10, max_iter = 1,000, verbose = 0, eps = 0.001, n_jobs = None, dissimilarity = 'euclidean'.

**Definition of lower and upper visual field voxels.** For area V1, we selected voxels tuned to the lower and upper portions of the visual field based on the V1d (that is, V1 dorsal) and V1v (that is, V1 ventral) NSD delineations, respectively. For area V4, we used the polar angle maps from the NSD pRF experiment to manually divide the area into voxels tuned to the lower and upper portions of the visual field.

## fMRI experiments

**Participants.** Six healthy adults (mean age 25.83 years, standard deviation 4.67 years; 4 female, 2 male) participated; all had normal or corrected-to-normal vision. No statistical methods were used to predetermine sample sizes, but our sample sizes are similar to those reported in previous publications[27,57]. All participants provided written informed consent and received monetary reimbursement (at the university rate of 12 euros per hour). Procedures were approved by the ethical committee of the Department of Education and Psychology at Freie Universität Berlin and were in accordance with the Declaration of Helsinki.

**Stimuli.** During the fMRI experiments, we presented participants with 150 images from the V1 versus V4 univariate RNC solutions (25 images from each of the two aligning conditions, 25 images from each of the two disentangling conditions, 25 images from V1's baseline and 25 images from V4's baseline) (Fig. 3a) and 150 images from the V1 versus V4 multivariate RNC solutions (50 images from aligning condition, 50 images from disentangling condition and 50 images from the baseline) (Fig. 5a). All images were sized 425 pixels × 425 pixels × 3 RGB channels.

To prevent confounds driven by luminance, we matched each image's mean luminance (that is, its luminance across all pixels) to

the luminance of the stimulus presentation screen background (a uniform grey screen with an RGB value of [127 127 127]), using the 'ImageEnhance' function from the Pillow Python package (https://python-pillow.org/).

**Experimental paradigm.** *Main experiment.* Each participant underwent two fMRI data collection sessions. Each session consisted of multiple 4-s trials, where an image was presented for 2 s, followed by 2 s of grey screen interstimulus interval (Extended Data Fig. 8a). To ensure that participants paid attention, we presented the RNC controlling images within an orthogonal target detection task where we asked participants to report, through a button press, whenever a catch image containing the fictional character Buzz Lightyear appeared on the screen.

During the first session, we presented the 150 controlling images from univariate RNC, across 10 runs. Each run consisted of 109 4-s trials: it started with 3 blank trials (that is, a grey screen where no image was presented), continued with a pseudo-randomized order of 90 univariate RNC image trials, 8 blank trials and 4 catch trials (that is, images containing Buzz Lightyear) and ended with 4 blank trials. Across all 10 runs, this resulted in 6 presentation repeats for each of the 150 univariate RNC controlling images.

During the second session, we presented the 150 controlling images from multivariate RNC, across 12 runs. Each run consisted of 121 4-s trials: it started with 3 blank trials, continued with a pseudo-randomized order of 100 multivariate RNC image trials, 9 blank trials and 5 catch trials (that is, images containing Buzz Lightyear) and ended with 4 blank trials. Across all 12 runs, this resulted in 8 presentation repeats for each of the 150 multivariate RNC controlling images.

All images were presented centrally, with a horizontal and vertical visual angle of 8.4°, against a grey background with an RGB value of [127, 127, 127]. A small semitransparent red fixation dot with a black border (0.2° × 0.2°, 50% opacity) was present at the centre of the images throughout the entirety of both sessions, and we asked participants to maintain central fixation throughout the experiment. We controlled stimulus presentation using the Psychtoolbox[92] and recorded fMRI responses during both experimental sessions.

*pRF experiment.* We ran the 'multibar' pRF experiment used in the NSD[27], which is an adaptation of the pRF experiment used in the Human Connectome Project 7 T Retinotopy Dataset[49]. Stimuli consisted of slowly moving apertures filled with a dynamic colourful texture and involved bars sweeping in multiple directions (same as RETBAR in the Human Connectome Project 7 T Retinotopy Dataset) (Extended Data Fig. 8b). Apertures and textures were updated at a rate of 16 Hz. Stimuli filled a circular region with diameter 12°. Each run lasted 300 s and included blank periods. Throughout stimulus presentation, a small semitransparent dot (with diameter 0.2°) was present at the centre of the stimuli. The colour of the central dot switched randomly to one of three colours (black, white or red) every 1–5 s. Participants were instructed to maintain fixation on the dot and to press a button whenever the dot changed colour. To further aid fixation, a semitransparent fixation grid was superimposed on the stimuli and was present throughout the experiment[93]. For each participant, we collected three runs of the pRF experiment, at the beginning of the first fMRI session.

**fMRI.** *Acquisition.* We collected MRI data using a Siemens Magnetom Prisma Fit 3 T system (Siemens Medical Solutions) with a 64-channel head coil.

Anatomical scans were acquired during each recording session using a standard T1-weighted sequence (repetition time (TR) 1.9 s, echo time (TE) 3.22 ms, number of slices 176, field of view (FOV) 225 mm, voxel size 1.0 mm isotropic, flip angle 8°).

Functional images were acquired using gradient-echo echo planar imaging (EPI) at 2.5-mm isotropic resolution with partial brain coverage

(TR 1 s, TE 33 ms, number of axial slices 39, matrix size 82 × 82, FOV 205 mm, flip angle 70°, acquisition order interleaved, interslice gap 0.25 mm, multiband slice acceleration factor 3). The acquisition volume fully covered the occipital lobe.

Dual-echo field maps were acquired during each recording session (TR 0.4 s, $TE_1$ 4.92 ms, $TE_2$ 7.38 ms, number of slices 38, voxel size 3 mm isotropic, matrix size 66 × 66, FOV 198 mm, flip angle 60°).

*Preprocessing.* We preprocessed the fMRI data using SPM12 (https://www.fil.ion.ucl.ac.uk/spm/software/spm12/). Preprocessing steps included realigning all functional images to the first image of each run, slice-time correction, field map correction and co-registration of the functional images to the anatomical image of the first recording session.

*pRF mapping.* The pRF mapping analysis was run using the prf-workflow package (https://github.com/mayajas/prf-workflow), with the model fitting done with the pRFpy package (v0.1.0; https://github.com/VU-Cog-Sci/prfpy). The preprocessed functional data of the three pRF runs were projected to the Freesurfer reconstruction of the white matter cortical surface of the given participant. The surface-projected signals at each surface mesh vertex were detrended to account for linear drifts, bandpass filtered (0.01–0.1 Hz) and z-scored over time. The signals from the three pRF runs were then averaged together. We fit an isotropic two-dimensional Gaussian pRF model to the data at each cortical surface vertex, with an initial coarse grid fit followed by a fine iterative fit, to optimize the parameters that define pRF size and the location (*x*, *y*) in Cartesian coordinates in visual space that the underlying population of neurons responds to.

The optimized location parameters were transformed to eccentricities and polar angle maps, which we then used to manually delineate visual regions of interest V1 and V4. Delineations were constrained to the maximum stimulus eccentricity of the controlling images (that is, 8.4° of visual angle) on the basis of the eccentricity map, while the visual areas were identified on the basis of reversals in the polar angle map. To ensure specificity, the visual area delineations were drawn conservatively, with the dorsal and ventral boundaries drawn just dorsally and ventrally of the corresponding polar angle reversal, respectively (Extended Data Fig. 8c).

*GLM.* We used GLMsingle[94] to estimate single-trial beta responses (that is, blood oxygenation level dependent (BOLD) response amplitudes evoked by each image trial) of the preprocessed fMRI data from the main experiment. GLMsingle provides single-trial beta estimates following three steps. First, for each voxel, a custom haemodynamic response function is identified from a library of candidate functions. Second, cross-validation is used to derive a set of noise regressors from voxels that have negligible amounts of BOLD variance related to the experiment (using an $R^2$ threshold). Third, to improve the stability of beta estimates for closely spaced trials, betas are regularized on a voxel-wise basis using ridge regression. The resulting betas indicate the percentage of BOLD signal change evoked by single-image trials, with respect to a baseline corresponding to the absence of a stimulus (that is, a grey screen with no image presented). We applied GLMsingle with default parameters, independently to the preprocessed fMRI responses of each participant, session and area (that is, independently for the voxels of V1 and V4).

*z-Scoring and voxel selection.* For consistency with the in silico fMRI data, here too we z-scored the beta responses (from GLMsingle) of each voxel across all trials of each session and computed the NCSNR of each voxel. For further analyses, we retained only those voxels with NCSNR scores above 0.4. The more liberal NCSNR threshold (compared with the in silico fMRI data analyses) comes from the fact that not all

recorded participants and areas consisted in voxels with NCSNR scores above 0.5. We computed the NCSNR independently for the data of the two recording sessions, that is, independently for the fMRI responses for the univariate and multivariate RNC images. This resulted in a different amount of retained voxels between the two experimental sessions, which can be seen in Supplementary Tables 4 and 5.

### Statistical testing

We assessed statistical significance using population prevalence testing[95,96], which is well suited to determine the significance of an effect at the level of the population when analysing small samples of intensely scanned participants. Population prevalence testing is a two-level procedure. At the first level, significance is established independently within each participant. At the second level, the binary results from the first level (that is, the counts of significant participants) are used to estimate the probability of this happening by chance, under the null hypothesis of no effect in any member of the population, thus providing a population-level inference. In the following, we describe these two levels in detail.

At the first level, we established significance independently within each participant using non-parametric permutation tests. Each test consisted in the following: computing the statistic of interest (that is, the observed statistic); creating the null distribution of the observed statistic by recomputing it using 100,000 different random permutations of the data; quantifying the $P$ value as the proportion of permutations where the randomized statistic is as extreme or more extreme than the observed statistic; controlling the family-wise error rate by applying (non-negative) Benjamini–Hochberg correction[97] to the resulting $P$ values; and assigning significance to participants with corrected $P$ values below 0.05. In the encoding model noise analysis tests, the null hypothesis was that the noise-ceiling-normalized explained variance scores of the different predictors were equal; we permuted the encoding accuracy scores over fMRI voxels and predictors, and we corrected the $P$ values over two tests for each area (one test comparing single and average NSD trials as predictors, and one test comparing average NSD trials and in silico responses as predictors). In the univariate RNC analysis tests, the null hypothesis was that the univariate responses for the controlling and baseline images were equal; we permuted the univariate responses across image conditions, and we corrected the $P$ values over eight tests for each pairwise comparison of areas (one test for each combination of two areas and four neural control conditions). In the univariate RNC cortical distance analysis tests, the null hypothesis was that the absolute differences between the baseline univariate responses and the univariate responses for controlling images from different stepwise area distances were equal; we permuted the univariate responses across areas, and we corrected the $P$ values over four tests (one test for each neural control condition). In the multivariate RNC analysis tests, the null hypothesis was that the RSA scores for the controlling and baseline images were equal; we permuted the multivariate responses across image conditions, and we corrected the $P$ values over two tests for each pairwise comparison of areas (one test for each neural control condition). In the multivariate RNC cortical distance analysis tests, the null hypothesis was that the absolute differences between the baseline RSA scores and the RSA scores for controlling images from different stepwise area distances were equal; we permuted the multivariate responses across areas and image conditions, and we corrected the $P$ values over three tests (one test for the aligning, disentangling and baseline images, respectively). In the multivariate RNC retinotopy analysis tests, the null hypothesis was that the univariate responses of voxels tuned to the lower and upper portions of the visual field were equal; we permuted the data across ventral and dorsal voxels, and we corrected the $P$ values over two tests for each area (one test for sky and no sky images, respectively). In the univariate RNC categorical selectivity analysis tests, the null

hypothesis was that the absolute differences between the baseline univariate responses and the univariate responses for controlling images from areas within or between categorical selectivity groups were equal; we permuted the univariate responses across areas, and we corrected the $P$ values over four tests (one test for each neural control condition). In the multivariate RNC categorical selectivity analysis tests, the null hypothesis was that the absolute differences between the baseline RSA scores and the RSA scores for controlling images from areas within or between categorical selectivity groups were equal; we permuted the univariate responses across areas and image conditions, and we corrected the $P$ values over two tests (one test for each neural control condition).

At the second level, we used the cumulative density function of the binomial distribution of within-participant significance to estimate the probability of observing significant participants by chance, if there was no effect in any member of the population, as follows:

$$P = 1 - \text{CDF}(k, n, a = 0.05),$$

where CDF is the cumulative density function of the binomial distribution, $k$ is the number of significant participants, $n$ is the total number of participants (eight or seven participants for tests on the in silico fMRI responses from encoding models trained on NSD and VIR, respectively; six participants for tests on the in vivo fMRI responses from the fMRI experiments) and $a$ is the probability of success in each trial under the null hypothesis. Finally, we assigned statistical significance at the population level for probabilities of $P < 0.05$.

To calculate the confidence intervals of each statistic, we created 100,000 bootstrapped samples by sampling the participant-specific results with replacement. This yielded empirical distributions of the results, from which we derived the 95% confidence intervals.

### Software

Data analyses were carried out in Python 3, using the following libraries (version in parenthesis): argparse (v1.4.0); GLMsingle (v1.2); h5py (v3.1.0); Matplotlib (v3.9.4); NEST (v1.0.0); NiBabel (v4.0.2); Nilearn (v0.9.2); Numpy (v1.26.4); nsdcode (v1.0.0); Pandas (v1.5.1); Pillow (v9.2.0); prf-workflow (v0.1); pRFpy (v0.1.0); PyTorch (v1.13.0); scikit-learn (v1.1.1); SciPy (v1.12.0); Statsmodels (v0.13.5); Torchvision (v0.14.0); tqdm (v4.64.1).

fMRI data collection and preprocessing were carried out in MATLAB 2020a, using Psychtoolbox-3 and SPM12, respectively.

### Reporting summary

Further information on research design is available in the Nature Portfolio Reporting Summary linked to this article.

## Data availability

The univariate and multivariate RNC algorithms were applied on in silico fMRI responses generated using the Brain Encoding Response Generator, available via GitHub at https://github.com/gifale95/BERG. The NSD is available at https://naturalscenesdataset.org/. The VIR dataset is available via Figshare at https://doi.org/10.6084/m9.figshare.23590302.v2 (ref. 98). The controlling images for all pairwise comparisons of areas, along with the in vivo fMRI responses for the V1 versus V4 comparison controlling images, are available via OpenNeuro at https://openneuro.org/datasets/ds005503.

## Code availability

The code to reproduce all the results is available via GitHub at https://github.com/gifale95/RNC. To facilitate RNC adoption, the GitHub repository also includes Colab tutorials where users can interactively implement univariate and multivariate RNC on in silico fMRI responses of areas spanning the entire visual cortex for -150,000 naturalistic images.

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

## Acknowledgements

A.T.G. is supported by a PhD fellowship of the Einstein Center for Neurosciences. M.A.J. is supported by the Horizon Europe Framework Programme (HORIZON-MSCA-2021-PF-01, grant number 101064539). R.M.C. is supported by German Research Council (DFG) grants (CI 241/1-1, CI 241/1-3, CI 241/1-7 and INST 272/297-1), the European Research Council (ERC) starting grant (ERC-StG-2018-803370) and the ERC Consolidator grant (ERC-CoG-2024101123101). The funders had no role in the study design, data collection and analysis, decision to publish or preparation of the manuscript. We thank the HPC Service of FUB-IT, Freie Universität Berlin, for computing time (https://doi.org/10.17169/refubium-26754). We thank K. Kay for helpful feedback and I. Charest for help with the Natural Scenes Dataset.

## Author contributions

A.T.G. and R.M.C. designed research. A.T.G., J.J.D.S. and M.A.J. acquired fMRI data. A.T.G. and J.J.D.S. preprocessed fMRI data. A.T.G. and M.A.J. performed pRF analyses. A.T.G. modelled and analysed data. A.T.G. and R.M.C. interpreted results. A.T.G. prepared figures. A.T.G. drafted the manuscript. A.T.G., J.J.D.S., M.A.J. and R.M.C. edited and revised the manuscript. All authors approved the final version of the manuscript.

## Funding

## Competing interests

The authors declare no competing interests.

## Additional information

**Extended data** is available for this paper at https://doi.org/10.1038/s41562-025-02252-z.

**Correspondence and requests for materials** should be addressed to Alessandro T. Gifford.

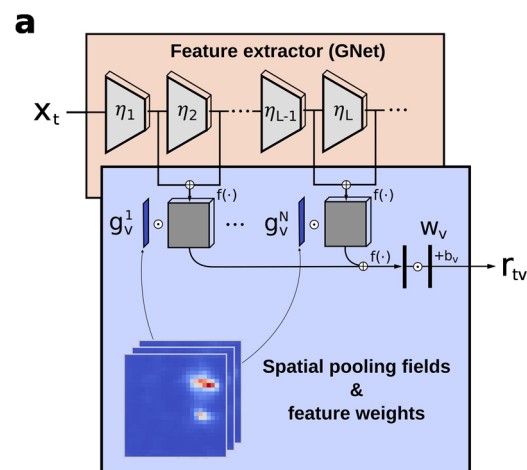

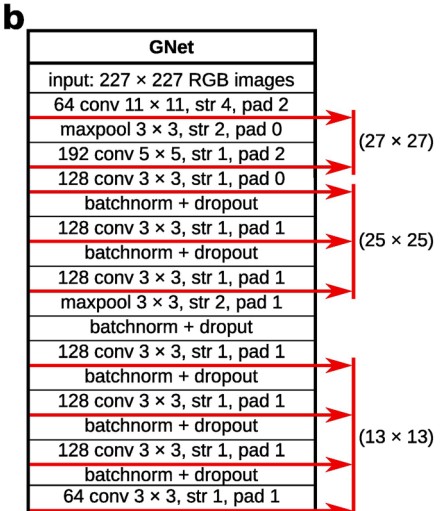

**Extended Data Fig. 1 | Encoding model architecture. a**, Illustration of an encoding model that predicts brain activity in a given voxel ($r_{tv}$) in response to images ($x_t$). Images are passed to nonlinear feature extractors (that is, GNet), $\eta_l$ (trapezoids), that output feature maps (grey cuboids). Feature maps are grouped, passed through an element-wise nonlinearity, $f(\cdot)$, and then multiplied pixel-wise by a spatial pooling field ($g^1,\dots,g^N$ where superscripts index distinct groups of feature maps) that determines the region of visual space that drives voxel activity. The weighted pixel values in each feature map are then summed, reducing each feature map to a scalar value. These scalar values are concatenated across all feature maps, forming a single feature vector that is passed through another element-wise nonlinearity (left black rectangle) and then weighted by a

set of feature weights, $w$ (right black rectangle), to yield predicted voxel activity. The feature extractors $\eta_l$ (that is, GNet), the spatial pooling fields $g^1,\dots,g^N$, and the feature weights $w$ are all optimized while training the encoding model to predict brain responses. **b**, GNet's architecture. GNet is a deep convolutional neural network consisting of convolutional layers (rows labeled 'conv'; values indicate feature depth and convolutional filter resolution; 'str' = filter stride, 'pad' = convolutional padding), max-pooling layers ('maxpool'), batch-normalization and weight-dropout layers ('batchnorm + dropout'). Feature maps in the convolutional layers (indicated by red arrows; resolution of the feature maps in parentheses) are used as predictors of brain activity in the context of an encoding model. **a-b**, (adapted from Allen et al.[27]).

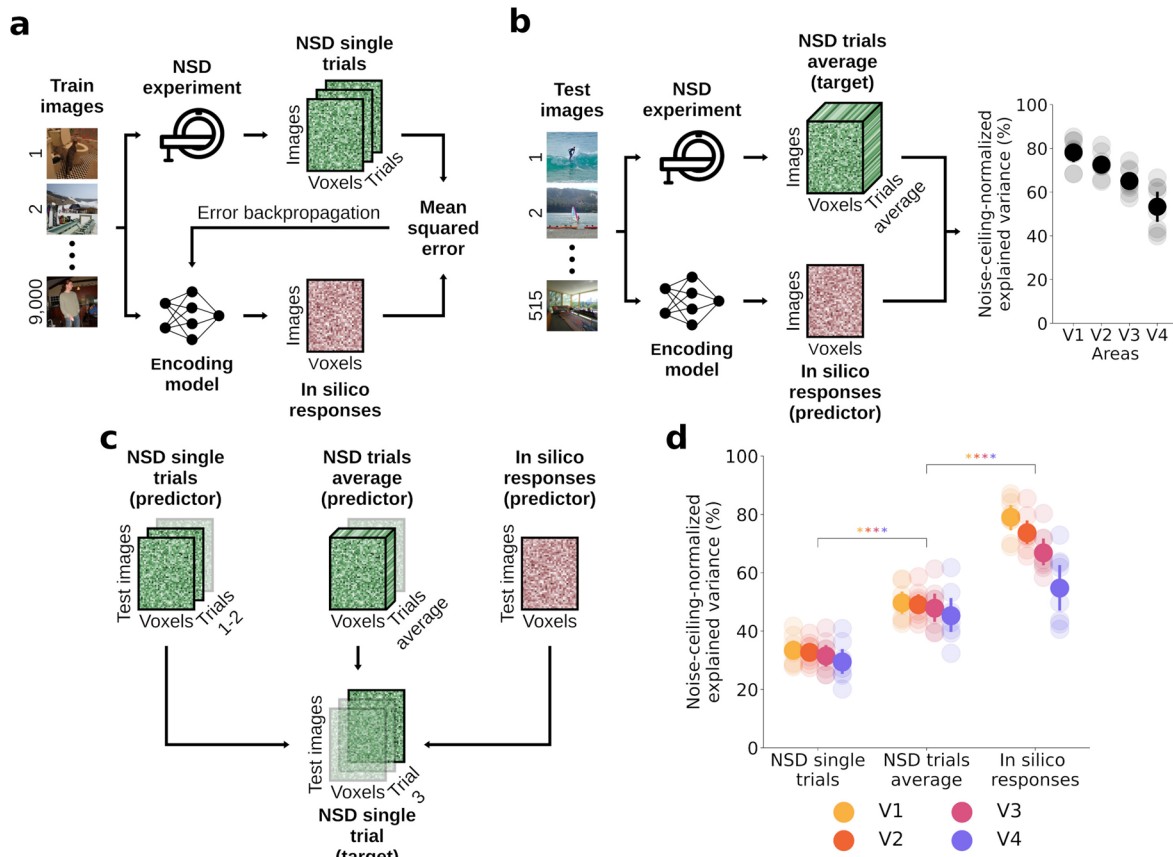

**Extended Data Fig. 2 | Encoding models training and testing. a**, Encoding models training. For each participant and area, we trained end-to-end encoding models that take images as input and predict the corresponding fMRI responses, using the single-trial NSD responses for the 9,000 participant-unique images. During training, the model predictions were compared to the single-trial target fMRI responses, and the resulting error was backpropagated to update the encoding model weights. **b**, We tested the encoding models on an independent portion of the NSD data not used for training, consisting of fMRI responses for 515 images seen three times by all participants, averaged across the three trials. Photos from the COCO image dataset/Flickr[85]. **c**, We compared the noise of the in silico fMRI responses with the noise of the in vivo fMRI responses from the NSD, by comparing how much variance these two data types explained for a third, independent split of the in vivo NSD responses. Because the in silico fMRI responses did not capture all signal variance in the NSD responses, the in silico fMRI responses explaining more variance than the in vivo NSD responses would be indicative of the former being less affected by noise[19,29]. We carried out the comparison through three sets of predictions, using the in silico and the in vivo

NSD fMRI responses for the 515 test images. Each prediction involved explaining in vivo single response trials from the NSD experiment with a different predictor. In the first set of predictions, the predictor consisted of one of the two remaining in vivo NSD experiment response trials. In the second set of predictions, the predictor consisted of the average of the two remaining in vivo NSD experiment response trials. In the third set of predictions, the predictor consisted of the in silico responses from the trained encoding models. **d**, Single NSD response trials noise-ceiling-normalized explained variance, for the three predictors of the noise analysis. The variance explained by the in silico responses is higher than the variance explained by both single and averaged NSD trials, indicating that the in silico fMRI responses are less affected by noise compared to the NSD responses. Colored asterisks indicate significant difference between the noise-ceiling-normalized explained variance scores of two predictors (within-participant permutation test, one-sided, $P < 0.05$, Benjamini/Hochberg corrected over 2 tests for each area; population prevalence test, one-sided, $P < 10^{-10}$, indicating within-participant significance in all 8 participants), for each area. Error bars reflect 95% confidence intervals.

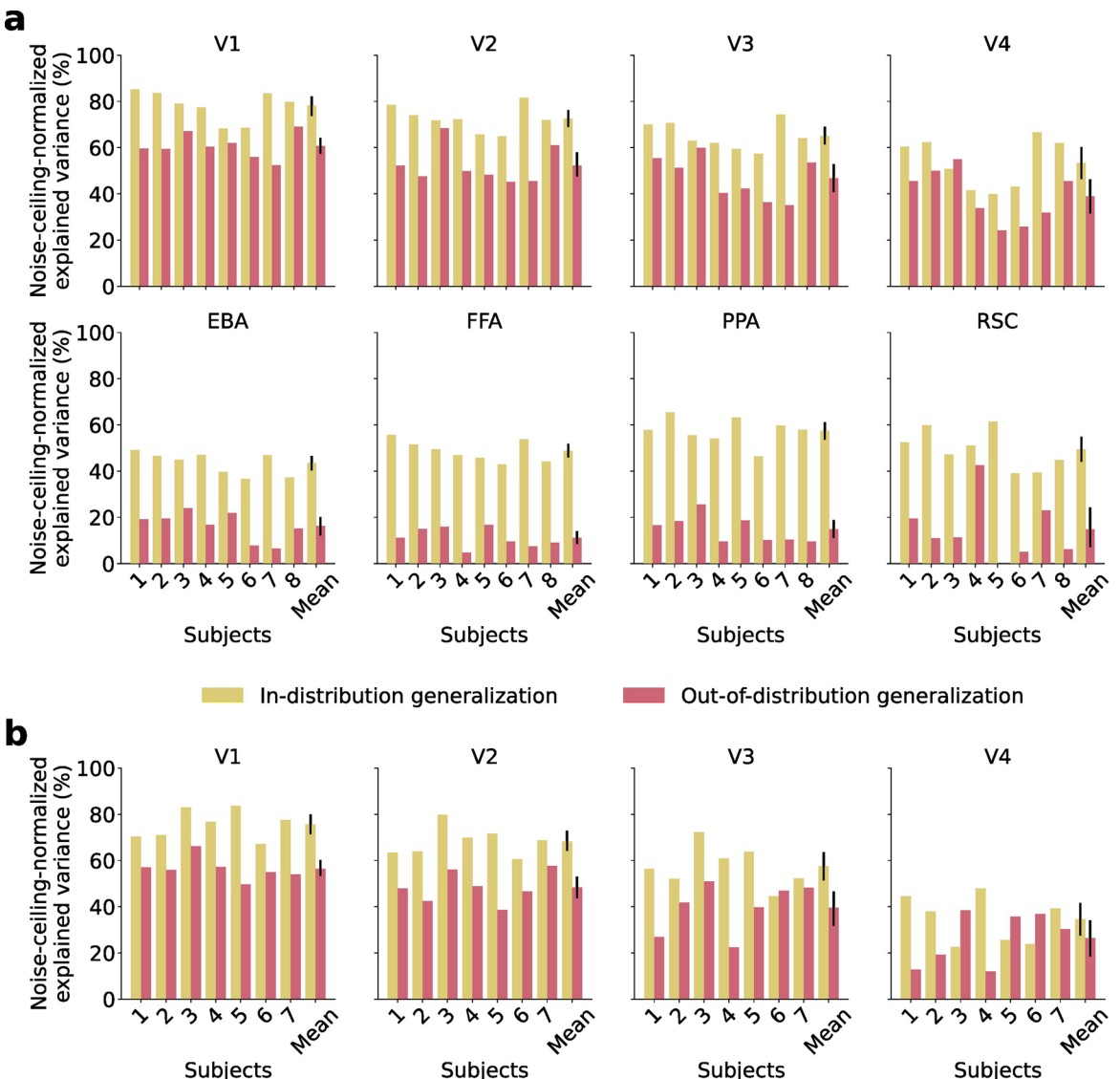

**Extended Data Fig. 3 | Encoding models in-distribution and out-of-distribution noise-ceiling-normalized explained variance scores. a**, Noise-ceiling-normalized explained variance for encoding models trained on NSD[27]. The in-distribution generalization scores reflect tests on a set of 515 images and fMRI responses not used for training. The out-of-distribution generalization scores reflect tests on the 284 NSD-synthetic[28] images and fMRI responses. **b**, Noise-ceiling-normalized explained variance for encoding models trained on the Visual Illusion Reconstruction Dataset[33]. The in-distribution generalization scores reflect tests on a set of 350 images and fMRI responses not used for training. The out-of-distribution generalization scores reflect tests on the 38 visual illusion images from the same dataset. **a-b**, The noise-ceiling-normalized explained variance scores are averaged across all voxels within each area.

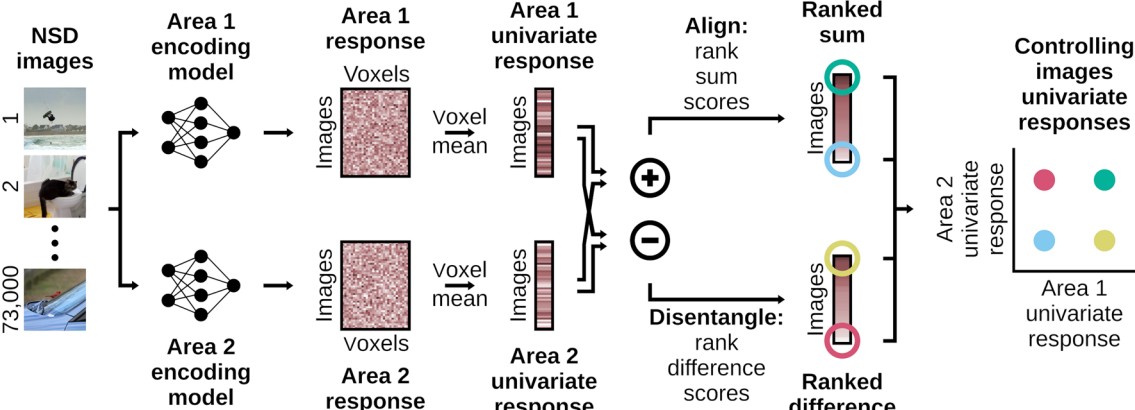

**Extended Data Fig. 4 | Univariate RNC algorithm.** Univariate RNC searches for images leading to aligned or disentangled in silico univariate fMRI responses of two visual areas. The 73,000 NSD images are fed to the trained encoding models of two areas, and the resulting in silico fMRI responses averaged across voxels, obtaining a one-dimensional univariate response vector of length 73,000, for each area. The univariate response vectors of the two areas are either summed (alignment) or subtracted (disentanglement), the sum or difference scores ranked, and the controlling images leading to highest and lowest scores are kept. This results in four sets of controlling images, each set corresponding to a different neural control condition. The controlling images from the sum vector lead to two neural control conditions in which both areas have aligned univariate responses (that is, images that either drive or suppress the responses of both areas), whereas the controlling images from the difference vector lead to two neural control conditions in which both areas have disentangled univariate responses (that is images that drive the responses of one area while suppressing the responses of the other area, and vice versa). Photos from the COCO image dataset/Flickr[85].

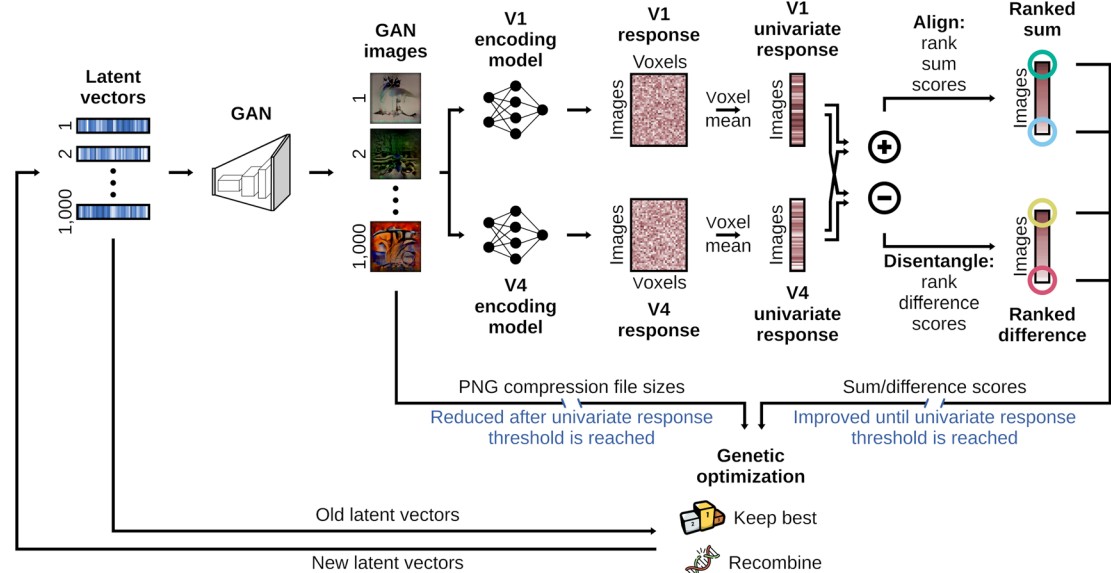

**Extended Data Fig. 5 | Generative univariate RNC algorithm.** Generative univariate RNC generates stimulus images leading to aligned or disentangled in silico univariate fMRI responses for V1 and V4, while at the same time being as simple as possible. A batch of 1,000 random latent vectors is given as input to a GAN[37], which uses them to generate 1,000 images, and the PNG compression file size of these images is calculated. Next, these images are fed to the trained encoding models of V1 and V4, and the resulting in silico fMRI responses averaged across voxels, obtaining a one-dimensional univariate response vector of length 1,000, for each area. The univariate response vectors of the two areas are either summed or subtracted, based on the neural control condition univariate RNC is optimizing for, thus obtaining sum or difference scores. The latent vectors, PNG compression file sizes, univariate responses, and sum or difference scores are then fed to a genetic optimization algorithm[38,39], which uses them to create a new generation of latent vectors (by keeping the 250 best performing latent vectors, and recombining the remaining 750 latent vectors). At first the latent vectors are optimized using the sum or difference scores, so to result in images

leading to in silico univariate fMRI responses for V1 and V4 closer to a threshold level. After this threshold is reached, the latent vectors are optimized using the PNG compression file sizes, so to result in images that are as simple as possible (while keeping the in silico univariate fMRI responses over the threshold). Finally, the new latent vectors are once again fed to the GAN, and the same steps are repeated over a new generation. After several genetic algorithm optimizations, this results in an image (that is, the best performing image from the last genetic optimization generation) that well controls neural responses following one of the four univariate RNC neural control conditions (that is, two alignment conditions where the in silico univariate fMRI responses of both areas are either driven or suppressed, and two disentanglement conditions where the in silico univariate fMRI response of one area is driven while the response of the other area is suppressed, and vice versa), while at the same time being as simple as possible. The images for the four neural control conditions are optimized independently of each other.

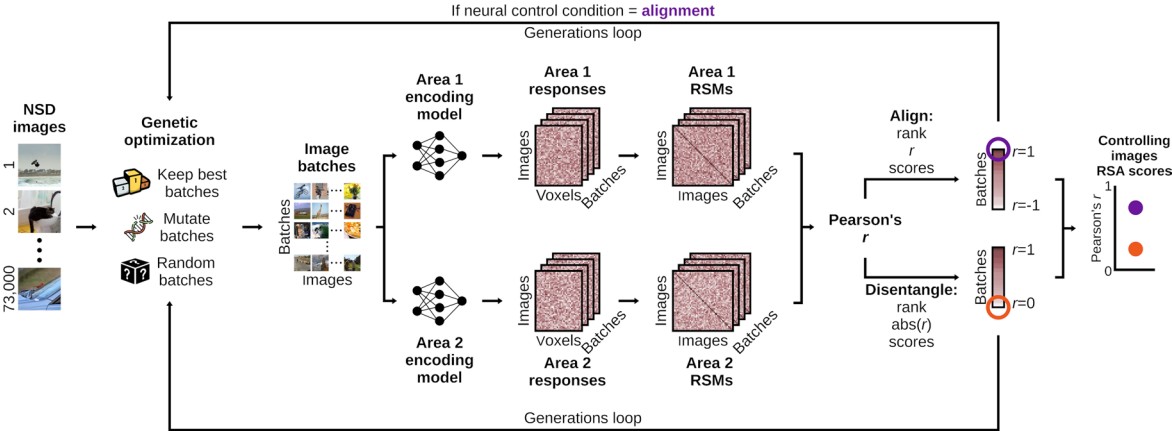

**Extended Data Fig. 6 | Multivariate RNC algorithm.** Multivariate RNC searches for images leading to aligned or disentangled in silico multivariate fMRI responses of two visual brain areas. Random images batches from the 73,000 NSD images are fed to the trained encoding models of two areas, and the resulting in silico fMRI responses are transformed into representational similarity matrices (RSMs)[25], yielding one RSM for each image batch and area. The RSMs of the two areas are then compared through RSA (that is, Pearson's correlation), obtaining one RSA correlation score ($r$) for each image batch, and the correlation scores ranked. To align the two areas, the image batches with highest correlation scores (that is, containing images most similarly represented by the two areas) undergo a genetic optimization (which involves keeping these image batches, creating mutated versions of them, and adding random image batches), resulting in new image batches likely to better align the two areas[38,39,42,43]. Finally, these new image batches are once again fed to the encoding models, and the same steps are repeated over a new generation. To disentangle the two areas the image batches with lowest absolute correlation scores (that is, containing images most differently represented by the two areas) are instead genetically optimized, resulting in new image batches likely to better disentangle the two areas. After several genetic optimization generations, this results in an image batch that well controls neural responses following one of the two multivariate RNC neural control conditions. The controlling images from the ranked correlation vector lead both areas to have aligned multivariate responses (that is, images leading to high RSA correlation scores for the two areas), whereas the controlling images from the absolute ranked correlation vector lead both areas to have disentangled multivariate responses (that is, images leading to low absolute RSA correlation scores for the two areas). The image batches from the two neural control conditions are optimized independently of each other. Photos from the COCO image dataset/Flickr[85].

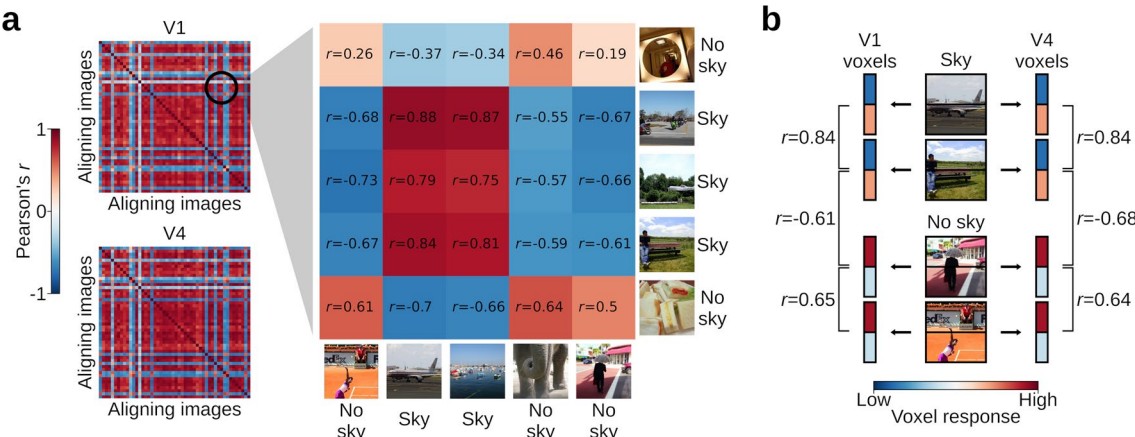

**Extended Data Fig. 7 | Relationship between multivariate RNC controlling images, fMRI responses, and RSM entries. a**, V1 and V4 participant-average RSMs for the multivariate RNC aligning images. **b**, Relationship between images with and without the sky on their upper half, and fMRI responses for voxels tuned to the higher and lower portion of the visual field. Due to retinotopy, uniform regions on a spatially constrained portion of the image led to suppressed responses for voxels tuned to the corresponding portion of the visual field. Thus, the response of voxels tuned to the upper portion of the visual field were consistently suppressed by images including the sky on their upper half, whereas the same images drove the response of voxels tuned to the lower portion of the visual field (since the lower portion of these images includes non-uniform regions such as objects) (Fig. 5d). The opposite pattern was observed for images not including the sky (Fig. 5d). This led to highly positive correlations (and corresponding RSM entries) when correlating the voxel responses for two sky images, or for two no sky images, and to highly negative correlations when correlating the voxel responses for a sky image and a no sky image (Fig. 5c). Photos from the COCO image dataset/Flickr[85].

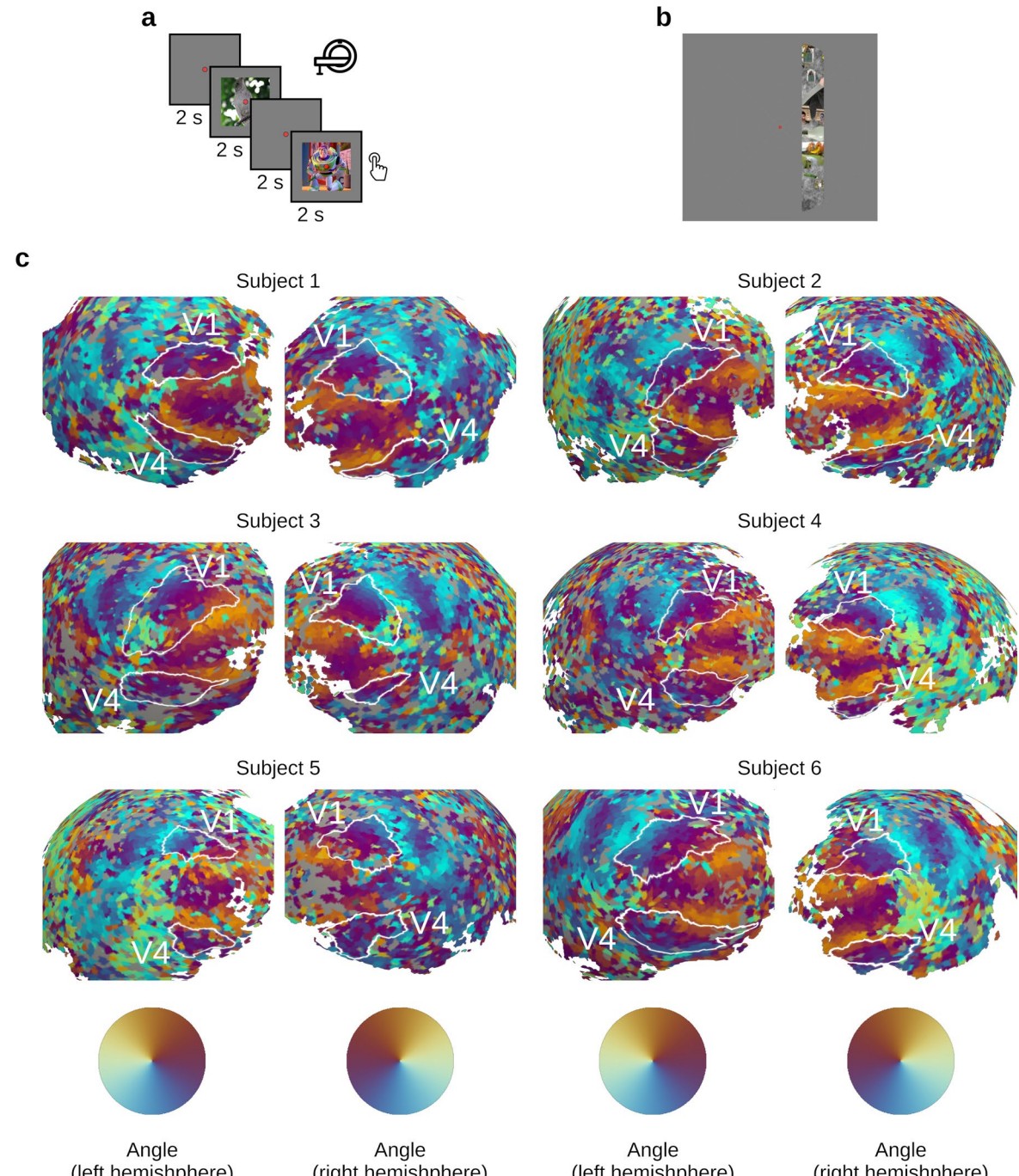

**Extended Data Fig. 8 | In vivo fMRI experiments and polar angle maps.**
**a**, Experimental design. We presented the univariate and multivariate RNC controlling and baseline images during a target detection task, where we asked participants to press a button whenever an image with Buzz Lightyear appeared on the screen. Each image was presented for two seconds, followed by two seconds of inter-stimulus interval. Participants were asked to fixate a central red dot during the entire experiment. fMRI responses were collected during image presentation. Photos from the COCO image dataset/Flickr[85]. **b**, Screenshot of the pRF experiment used to delineate areas V1 and V4. **c**, Polar angle maps and V1/V4 delineations. Results are shown on FreeSurfer's sphere surface.

# Reporting Summary

## Statistics

For all statistical analyses, confirm that the following items are present in the figure legend, table legend, main text, or Methods section.

| n/a | Confirmed | |
|-----|-----------|---|
| ☐ | ☒ | The exact sample size (*n*) for each experimental group/condition, given as a discrete number and unit of measurement |
| ☐ | ☒ | A statement on whether measurements were taken from distinct samples or whether the same sample was measured repeatedly |
| ☐ | ☒ | The statistical test(s) used AND whether they are one- or two-sided *Only common tests should be described solely by name; describe more complex techniques in the Methods section.* |
| ☒ | ☐ | A description of all covariates tested |
| ☐ | ☒ | A description of any assumptions or corrections, such as tests of normality and adjustment for multiple comparisons |
| ☐ | ☒ | A full description of the statistical parameters including central tendency (e.g. means) or other basic estimates (e.g. regression coefficient) AND variation (e.g. standard deviation) or associated estimates of uncertainty (e.g. confidence intervals) |
| ☐ | ☒ | For null hypothesis testing, the test statistic (e.g. *F*, *t*, *r*) with confidence intervals, effect sizes, degrees of freedom and *P* value noted *Give P values as exact values whenever suitable.* |
| ☒ | ☐ | For Bayesian analysis, information on the choice of priors and Markov chain Monte Carlo settings |
| ☒ | ☐ | For hierarchical and complex designs, identification of the appropriate level for tests and full reporting of outcomes |
| ☐ | ☒ | Estimates of effect sizes (e.g. Cohen's *d*, Pearson's *r*), indicating how they were calculated |

*Our web collection on statistics for biologists contains articles on many of the points above.*

## Software and code

Policy information about availability of computer code

| Data collection | MATLAB 2020a<br>Psychtoolbox-3 (https://github.com/Psychtoolbox-3/Psychtoolbox-3) |
|-----------------|------------------------------------------------------------------------------------|
| Data analysis | fMRI preprocessing (MATLAB 2020a):<br>- SPM12 (https://www.fil.ion.ucl.ac.uk/spm/software/spm12/).<br><br>Data analysis (Python 3):<br>- argparse v1.4.0 (https://github.com/python/cpython/blob/main/Lib/argparse.py)<br>- GLMsingle v1.2 (https://github.com/cvnlab/GLMsingle)<br>- h5py v3.1.0 (https://github.com/h5py/h5py)<br>- Matplotlib v3.9.4 (https://github.com/matplotlib/matplotlib)<br>- NEST v1.0.0 (https://github.com/gifale95/NEST)<br>- NiBabel v4.0.2 (https://github.com/nipy/nibabel)<br>- Nilearn v0.9.2 (https://github.com/nilearn/nilearn)<br>- Numpy v1.26.4 (https://github.com/numpy/numpy)<br>- nsdcode v1.0.0 (https://github.com/cvnlab/nsdcode)<br>- Pandas v1.5.1 (https://github.com/pandas-dev/pandas)<br>- Pillow v9.2.0 (https://github.com/python-pillow/Pillow)<br>- prf-workflow v0.1 (https://github.com/mayajas/prf-workflow)<br>- pRFpy v0.1.0 (https://github.com/VU-Cog-Sci/prfpy)<br>- PyTorch v1.13.0 (https://github.com/pytorch/pytorch) |

- scikit-learn v1.1.1 (https://github.com/scikit-learn/scikit-learn)
- SciPy v1.12.0 (https://github.com/scipy/scipy)
- Statsmodels v0.13.5 (https://github.com/statsmodels/statsmodels)
- Torchvision v0.14.0 (https://github.com/pytorch/vision)
- tqdm v4.64.1 (https://github.com/tqdm/tqdm)

The code to reproduce all the paper results is available on GitHub (https://github.com/gifale95/RNC).

For manuscripts utilizing custom algorithms or software that are central to the research but not yet described in published literature, software must be made available to editors and reviewers. We strongly encourage code deposition in a community repository (e.g. GitHub). See the Nature Portfolio guidelines for submitting code & software for further information.

## Data

Policy information about availability of data

All manuscripts must include a data availability statement. This statement should provide the following information, where applicable:
- Accession codes, unique identifiers, or web links for publicly available datasets
- A description of any restrictions on data availability
- For clinical datasets or third party data, please ensure that the statement adheres to our policy

- The univariate and multivariate RNC algorithms were applied on in silico fMRI responses generated using the Neural Encoding Simulation Toolkit (https://github.com/gifale95/NEST).
- The Natural Scenes Dataset (NSD) is available at (https://naturalscenesdataset.org/).
- The Visual Illusion Reconstruction (VIR) dataset is available at (https://figshare.com/articles/dataset/Reconstructing_visual_illusory_experiences_from_human_brain_activity/23590302).
- The controlling images for all pairwise comparison of areas, along with the in vivo fMRI responses for the V1 vs. V4 comparison controlling images, are available on OpenNeuro (https://openneuro.org/datasets/ds005503).

## Research involving human participants, their data, or biological material

Policy information about studies with human participants or human data. See also policy information about sex, gender (identity/presentation), and sexual orientation and race, ethnicity and racism.

| | |
|---|---|
| Reporting on sex and gender | While we recorded sex (4 self-reported female, and 2 male), we did not use this information in our analyses. We studied the visual perception of naturalistic images, and have no reason to believe sex or gender plays a substantial role. |
| Reporting on race, ethnicity, or other socially relevant groupings | We did not perform any groupings of participants using race, ethnicity, socially relevant groupings, or otherwise. |
| Population characteristics | Participants self-reported their age (25.83 +/- 2.67 years mean+/- STD) and confirmed normal or corrected-to-normal vision. |
| Recruitment | Participants were recruited locally around the university through email messaging. This recruitment may bias participants towards a certain age group and social status. However, such biases are not thought to have any significant impact on visual processing. |
| Ethics oversight | The experiment was conducted in accordance with the Declaration of Helsinki and approved by the Ethics Committee of the Department of Education and Psychology of the Freie Universität Berlin. |

Note that full information on the approval of the study protocol must also be provided in the manuscript.

# Field-specific reporting

Please select the one below that is the best fit for your research. If you are not sure, read the appropriate sections before making your selection.

☒ Life sciences  ☐ Behavioural & social sciences  ☐ Ecological, evolutionary & environmental sciences

For a reference copy of the document with all sections, see nature.com/documents/nr-reporting-summary-flat.pdf

# Life sciences study design

All studies must disclose on these points even when the disclosure is negative.

| | |
|---|---|
| Sample size | For the in silico experiments we used n = 8 participants, as this is the number of participants available in the natural scenes dataset (on which the encoding models generating the in silico data were based). <br> For the in vivo experiments we used n = 6 participants. No prior sample size calculation was performed. We chose this sample size based on previous related work (see paper reference: Ratan Murty et al., 2021; Gu et al., 2023; Tuckute et al., 2024) to strike a good balance between within-subject brain measurements (two recording sessions per subject to obtain multiple stimulus repetitions for increased signal), and between-subject brain measurements. |

| Data exclusions | For one participant the pRF mapping analysis resulted in retinotopic maps of poor quality, from which we could not identify and delineate area V4. Since delineation of area V4 was a necessary precondition for the main analyses, we excluded this participant. |
| --- | --- |
| Replication | We provide the data and code to reproduce our analysis results, including plots and statistics. We have internally run the code multiple times to verify the same results. |
| Randomization | Experimental groups were not relevant in this study. Our study aims did not include differences between groups. |
| Blinding | Blinding was not relevant to this study as this study did not incorporate multiple experimental groups. |

# Behavioural & social sciences study design

All studies must disclose on these points even when the disclosure is negative.

| Study description | *Briefly describe the study type including whether data are quantitative, qualitative, or mixed-methods (e.g. qualitative cross-sectional, quantitative experimental, mixed-methods case study).* |
| --- | --- |
| Research sample | *State the research sample (e.g. Harvard university undergraduates, villagers in rural India) and provide relevant demographic information (e.g. age, sex) and indicate whether the sample is representative. Provide a rationale for the study sample chosen. For studies involving existing datasets, please describe the dataset and source.* |
| Sampling strategy | *Describe the sampling procedure (e.g. random, snowball, stratified, convenience). Describe the statistical methods that were used to predetermine sample size OR if no sample-size calculation was performed, describe how sample sizes were chosen and provide a rationale for why these sample sizes are sufficient. For qualitative data, please indicate whether data saturation was considered, and what criteria were used to decide that no further sampling was needed.* |
| Data collection | *Provide details about the data collection procedure, including the instruments or devices used to record the data (e.g. pen and paper, computer, eye tracker, video or audio equipment) whether anyone was present besides the participant(s) and the researcher, and whether the researcher was blind to experimental condition and/or the study hypothesis during data collection.* |
| Timing | *Indicate the start and stop dates of data collection. If there is a gap between collection periods, state the dates for each sample cohort.* |
| Data exclusions | *If no data were excluded from the analyses, state so OR if data were excluded, provide the exact number of exclusions and the rationale behind them, indicating whether exclusion criteria were pre-established.* |
| Non-participation | *State how many participants dropped out/declined participation and the reason(s) given OR provide response rate OR state that no participants dropped out/declined participation.* |
| Randomization | *If participants were not allocated into experimental groups, state so OR describe how participants were allocated to groups, and if allocation was not random, describe how covariates were controlled.* |

# Ecological, evolutionary & environmental sciences study design

All studies must disclose on these points even when the disclosure is negative.

| Study description | *Briefly describe the study. For quantitative data include treatment factors and interactions, design structure (e.g. factorial, nested, hierarchical), nature and number of experimental units and replicates.* |
| --- | --- |
| Research sample | *Describe the research sample (e.g. a group of tagged Passer domesticus, all Stenocereus thurberi within Organ Pipe Cactus National Monument), and provide a rationale for the sample choice. When relevant, describe the organism taxa, source, sex, age range and any manipulations. State what population the sample is meant to represent when applicable. For studies involving existing datasets, describe the data and its source.* |
| Sampling strategy | *Note the sampling procedure. Describe the statistical methods that were used to predetermine sample size OR if no sample-size calculation was performed, describe how sample sizes were chosen and provide a rationale for why these sample sizes are sufficient.* |
| Data collection | *Describe the data collection procedure, including who recorded the data and how.* |
| Timing and spatial scale | *Indicate the start and stop dates of data collection, noting the frequency and periodicity of sampling and providing a rationale for these choices. If there is a gap between collection periods, state the dates for each sample cohort. Specify the spatial scale from which the data are taken* |
| Data exclusions | *If no data were excluded from the analyses, state so OR if data were excluded, describe the exclusions and the rationale behind them, indicating whether exclusion criteria were pre-established.* |
| Reproducibility | *Describe the measures taken to verify the reproducibility of experimental findings. For each experiment, note whether any attempts to repeat the experiment failed OR state that all attempts to repeat the experiment were successful.* |

| Randomization | Describe how samples/organisms/participants were allocated into groups. If allocation was not random, describe how covariates were controlled. If this is not relevant to your study, explain why. |
| Blinding | Describe the extent of blinding used during data acquisition and analysis. If blinding was not possible, describe why OR explain why blinding was not relevant to your study. |

Did the study involve field work? ☐ Yes ☐ No

## Field work, collection and transport

| Field conditions | Describe the study conditions for field work, providing relevant parameters (e.g. temperature, rainfall). |
| Location | State the location of the sampling or experiment, providing relevant parameters (e.g. latitude and longitude, elevation, water depth). |
| Access & import/export | Describe the efforts you have made to access habitats and to collect and import/export your samples in a responsible manner and in compliance with local, national and international laws, noting any permits that were obtained (give the name of the issuing authority, the date of issue, and any identifying information). |
| Disturbance | Describe any disturbance caused by the study and how it was minimized. |

# Reporting for specific materials, systems and methods

We require information from authors about some types of materials, experimental systems and methods used in many studies. Here, indicate whether each material, system or method listed is relevant to your study. If you are not sure if a list item applies to your research, read the appropriate section before selecting a response.

### Materials & experimental systems

| n/a | Involved in the study |
|---|---|
| ☒ | Antibodies |
| ☒ | Eukaryotic cell lines |
| ☒ | Palaeontology and archaeology |
| ☒ | Animals and other organisms |
| ☒ | Clinical data |
| ☒ | Dual use research of concern |
| ☒ | Plants |

### Methods

| n/a | Involved in the study |
|---|---|
| ☒ | ChIP-seq |
| ☒ | Flow cytometry |
| ☐ | ☒ MRI-based neuroimaging |

## Antibodies

| Antibodies used | Describe all antibodies used in the study; as applicable, provide supplier name, catalog number, clone name, and lot number. |
| Validation | Describe the validation of each primary antibody for the species and application, noting any validation statements on the manufacturer's website, relevant citations, antibody profiles in online databases, or data provided in the manuscript. |

## Eukaryotic cell lines

Policy information about cell lines and Sex and Gender in Research

| Cell line source(s) | State the source of each cell line used and the sex of all primary cell lines and cells derived from human participants or vertebrate models. |
| Authentication | Describe the authentication procedures for each cell line used OR declare that none of the cell lines used were authenticated. |
| Mycoplasma contamination | Confirm that all cell lines tested negative for mycoplasma contamination OR describe the results of the testing for mycoplasma contamination OR declare that the cell lines were not tested for mycoplasma contamination. |
| Commonly misidentified lines (See ICLAC register) | Name any commonly misidentified cell lines used in the study and provide a rationale for their use. |

## Palaeontology and Archaeology

| Specimen provenance | Provide provenance information for specimens and describe permits that were obtained for the work (including the name of the |

| Specimen provenance | *issuing authority, the date of issue, and any identifying information). Permits should encompass collection and, where applicable, export.* |
|---|---|
| Specimen deposition | *Indicate where the specimens have been deposited to permit free access by other researchers.* |
| Dating methods | *If new dates are provided, describe how they were obtained (e.g. collection, storage, sample pretreatment and measurement), where they were obtained (i.e. lab name), the calibration program and the protocol for quality assurance OR state that no new dates are provided.* |

☐ Tick this box to confirm that the raw and calibrated dates are available in the paper or in Supplementary Information.

| Ethics oversight | *Identify the organization(s) that approved or provided guidance on the study protocol, OR state that no ethical approval or guidance was required and explain why not.* |
|---|---|

Note that full information on the approval of the study protocol must also be provided in the manuscript.

# Animals and other research organisms

Policy information about studies involving animals; ARRIVE guidelines recommended for reporting animal research, and Sex and Gender in Research

| Laboratory animals | *For laboratory animals, report species, strain and age OR state that the study did not involve laboratory animals.* |
|---|---|
| Wild animals | *Provide details on animals observed in or captured in the field; report species and age where possible. Describe how animals were caught and transported and what happened to captive animals after the study (if killed, explain why and describe method; if released, say where and when) OR state that the study did not involve wild animals.* |
| Reporting on sex | *Indicate if findings apply to only one sex; describe whether sex was considered in study design, methods used for assigning sex. Provide data disaggregated for sex where this information has been collected in the source data as appropriate; provide overall numbers in this Reporting Summary. Please state if this information has not been collected. Report sex-based analyses where performed, justify reasons for lack of sex-based analysis.* |
| Field-collected samples | *For laboratory work with field-collected samples, describe all relevant parameters such as housing, maintenance, temperature, photoperiod and end-of-experiment protocol OR state that the study did not involve samples collected from the field.* |
| Ethics oversight | *Identify the organization(s) that approved or provided guidance on the study protocol, OR state that no ethical approval or guidance was required and explain why not.* |

Note that full information on the approval of the study protocol must also be provided in the manuscript.

# Clinical data

Policy information about clinical studies
All manuscripts should comply with the ICMJE guidelines for publication of clinical research and a completed CONSORT checklist must be included with all submissions.

| Clinical trial registration | *Provide the trial registration number from ClinicalTrials.gov or an equivalent agency.* |
|---|---|
| Study protocol | *Note where the full trial protocol can be accessed OR if not available, explain why.* |
| Data collection | *Describe the settings and locales of data collection, noting the time periods of recruitment and data collection.* |
| Outcomes | *Describe how you pre-defined primary and secondary outcome measures and how you assessed these measures.* |

# Dual use research of concern

Policy information about dual use research of concern

## Hazards

Could the accidental, deliberate or reckless misuse of agents or technologies generated in the work, or the application of information presented in the manuscript, pose a threat to:

No | Yes

☐ ☐ Public health

☐ ☐ National security

☐ ☐ Crops and/or livestock

☐ ☐ Ecosystems

☐ ☐ Any other significant area

## Experiments of concern

Does the work involve any of these experiments of concern:

No | Yes
☐ | ☐ Demonstrate how to render a vaccine ineffective
☐ | ☐ Confer resistance to therapeutically useful antibiotics or antiviral agents
☐ | ☐ Enhance the virulence of a pathogen or render a nonpathogen virulent
☐ | ☐ Increase transmissibility of a pathogen
☐ | ☐ Alter the host range of a pathogen
☐ | ☐ Enable evasion of diagnostic/detection modalities
☐ | ☐ Enable the weaponization of a biological agent or toxin
☐ | ☐ Any other potentially harmful combination of experiments and agents

# Plants

| | |
|---|---|
| Seed stocks | *Report on the source of all seed stocks or other plant material used. If applicable, state the seed stock centre and catalogue number. If plant specimens were collected from the field, describe the collection location, date and sampling procedures.* |
| Novel plant genotypes | *Describe the methods by which all novel plant genotypes were produced. This includes those generated by transgenic approaches, gene editing, chemical/radiation-based mutagenesis and hybridization. For transgenic lines, describe the transformation method, the number of independent lines analyzed and the generation upon which experiments were performed. For gene-edited lines, describe the editor used, the endogenous sequence targeted for editing, the targeting guide RNA sequence (if applicable) and how the editor was applied.* |
| Authentication | *Describe any authentication procedures for each seed stock used or novel genotype generated. Describe any experiments used to assess the effect of a mutation and, where applicable, how potential secondary effects (e.g. second site T-DNA insertions, mosiacism, off-target gene editing) were examined.* |

# ChIP-seq

## Data deposition

☐ Confirm that both raw and final processed data have been deposited in a public database such as GEO.

☐ Confirm that you have deposited or provided access to graph files (e.g. BED files) for the called peaks.

| | |
|---|---|
| Data access links<br>*May remain private before publication.* | *For "Initial submission" or "Revised version" documents, provide reviewer access links.  For your "Final submission" document, provide a link to the deposited data.* |
| Files in database submission | *Provide a list of all files available in the database submission.* |
| Genome browser session<br>(e.g. UCSC) | *Provide a link to an anonymized genome browser session for "Initial submission" and "Revised version" documents only, to enable peer review.  Write "no longer applicable" for "Final submission" documents.* |

## Methodology

| | |
|---|---|
| Replicates | *Describe the experimental replicates, specifying number, type and replicate agreement.* |
| Sequencing depth | *Describe the sequencing depth for each experiment, providing the total number of reads, uniquely mapped reads, length of reads and whether they were paired- or single-end.* |
| Antibodies | *Describe the antibodies used for the ChIP-seq experiments; as applicable, provide supplier name, catalog number, clone name, and lot number.* |
| Peak calling parameters | *Specify the command line program and parameters used for read mapping and peak calling, including the ChIP, control and index files used.* |
| Data quality | *Describe the methods used to ensure data quality in full detail, including how many peaks are at FDR 5% and above 5-fold enrichment.* |
| Software | *Describe the software used to collect and analyze the ChIP-seq data. For custom code that has been deposited into a community repository, provide accession details.* |

# Flow Cytometry

## Plots

Confirm that:

☐ The axis labels state the marker and fluorochrome used (e.g. CD4-FITC).

☐ The axis scales are clearly visible. Include numbers along axes only for bottom left plot of group (a 'group' is an analysis of identical markers).

☐ All plots are contour plots with outliers or pseudocolor plots.

☐ A numerical value for number of cells or percentage (with statistics) is provided.

## Methodology

| | |
|---|---|
| Sample preparation | *Describe the sample preparation, detailing the biological source of the cells and any tissue processing steps used.* |
| Instrument | *Identify the instrument used for data collection, specifying make and model number.* |
| Software | *Describe the software used to collect and analyze the flow cytometry data. For custom code that has been deposited into a community repository, provide accession details.* |
| Cell population abundance | *Describe the abundance of the relevant cell populations within post-sort fractions, providing details on the purity of the samples and how it was determined.* |
| Gating strategy | *Describe the gating strategy used for all relevant experiments, specifying the preliminary FSC/SSC gates of the starting cell population, indicating where boundaries between "positive" and "negative" staining cell populations are defined.* |

☐ Tick this box to confirm that a figure exemplifying the gating strategy is provided in the Supplementary Information.

# Magnetic resonance imaging

## Experimental design

| | |
|---|---|
| Design type | The pRF experiment was task-based and had a continuous design. The core experiment was task-based and had an event-related design. We also collected structural data. |
| Design specifications | Each participant underwent two fMRI data collection sessions. Each session consisted of multiple four-second trials, where an image was presented for two seconds, followed by two seconds of gray screen inter-stimulus interval. During the first session we presented the 150 controlling images from univariate RNC across 10 runs, resulting in 6 presentation repeats for each image. During the second session we presented the 150 controlling images from multivariate RNC across 10 runs, resulting in 8 presentation repeats for each image. The pRF experiment consisted of three five-minute runs at the beginning of the first session. |
| Behavioral performance measures | Button presses were recorded. To ensure data quality, response accuracy was measured. |

## Acquisition

| | |
|---|---|
| Imaging type(s) | Structural, functional. |
| Field strength | 3T. |
| Sequence & imaging parameters | Anatomical scans were acquired during each recording session using a standard T1-weighted sequence (TR = 1.9 s, TE = 3.22 ms, number of slices 176, FOV = 225 mm, voxel size 1.0 mm isotropic, flip angle 8°).<br>Functional images were acquired using a multiband 3 sequence with partial brain coverage (TR = 1 s, TE = 33 ms, number of slices 39, voxel size 2.5 mm isotropic, matrix size 82 × 82, FOV = 205 mm, flip angle 70°, acquisition order interleaved, inter-slice gap 0.25 mm). The acquisition volume fully covered the occipital lobe. |
| Area of acquisition | Partial brain coverage including visual cortex. |
| Diffusion MRI | ☐ Used   ☒ Not used |

## Preprocessing

| | |
|---|---|
| Preprocessing software | SPM12 (https://github.com/spm/spm12)<br>FreeSurfer v7 (https://github.com/freesurfer/freesurfer) |
| Normalization | Data were not normalized as the main analyses did not assume a common brain space. |
| Normalization template | Data were not normalized as the main analyses did not assume a common brain space. |

| | |
|---|---|
| Noise and artifact removal | For the GLM preparation of the data, the data-driven analysis method GLMdenoise was used. This method accounted for a variety of sources of noise (e.g., physiological, motion, scanner artifacts, effects of collinearity). |
| Volume censoring | No censoring was performed. |

## Statistical modeling & inference

| | |
|---|---|
| Model type and settings | Trial-wise fMRI response amplitudes were estimated for individual voxels in individual participants. Two types of analyses were then performed on these response amplitudes: mass univariate and RSA. |
| Effect(s) tested | For the mass univariate analysis, we tested whether the univariate fMRI responses for the controlling images were significantly higher/lower than the univariate fMRI responses for the baseline images.<br>For the RSA analysis, we tested whether the fMRI responses for the controlling images led to RSA scores significantly higher/lower than the fMRI responses for the baseline images. |

Specify type of analysis: ☐ Whole brain ☒ ROI-based ☐ Both

| | |
|---|---|
| Statistic type for inference<br><br>(See Eklund et al. 2016) | Voxel-wise. |
| Correction | FDR correction to adjust the p-values for multiple comparisons. |

## Models & analysis

| n/a | Involved in the study |
|---|---|
| ☒ | ☐ Functional and/or effective connectivity |
| ☒ | ☐ Graph analysis |
| ☐ | ☒ Multivariate modeling or predictive analysis |

| | |
|---|---|
| Functional and/or effective connectivity | No connectivity analysis was performed. |
| Graph analysis | No graph analysis was performed. |
| Multivariate modeling and predictive analysis | For the mass univariate analysis, we averaged the activity patterns over all voxels within a region of interest.<br>For the RSA analysis, we constructed representational similarity matrices by correlating multivariate brain activity patterns of all voxels within a region of interest. |

