## [Peer Review File · Nature Human Behaviour]

In silico discovery of representational relationships across visual cortex

Corresponding Author: Mx Alessandro Gifford

Version 0:

Decision Letter:

10th December 2024

Dear Mx Gifford,

Thank you once again for your manuscript, entitled "In silico discovery of representational relationships across visual cortex", and for your patience during the peer review process.

Your Article has now been evaluated by 3 referees. You will see from their comments copied below that, although they find your work of considerable potential interest, they have raised quite substantial concerns. In light of these comments, we cannot accept the manuscript for publication, but would be interested in considering a revised version if you are willing and able to fully address reviewer and editorial concerns.

We hope you will find the referees' comments useful as you decide how to proceed. If you wish to submit a substantially revised manuscript, please bear in mind that we will be reluctant to approach the referees again in the absence of major revisions. We are committed to providing a fair and constructive peer-review process. Do not hesitate to contact us if there are specific requests from the reviewers that you believe are technically impossible or unlikely to yield a meaningful outcome.

In particular, Reviewers #1 and #2 remark on the lack of engagement on how the present findings relate to prior work. Given the existence of similar methods, we ask that you address the issues related to the advance in the context of the earlier literature. Reviewer #2 also raises important concerns relating to statistical testing and requests the extension of the analyses to higher visual areas, which we ask you to perform. Finally, Reviewer #3 questions the generalizability of the results. We ask that you follow their recommendations to demonstrate robustness and generalization.

If you wish to submit a suitably revised manuscript, we would hope to receive it within 4 months. I would be grateful if you could contact us as soon as possible if you foresee difficulties with meeting this target resubmission date.

- Include a "Response to the editors and reviewers" document detailing, point-by-point, how you addressed each editor and referee comment. If no action was taken to address a point, you must provide a compelling argument. When formatting this document, please respond to each reviewer comment individually, including the full text of the reviewer comment verbatim followed by your response to the individual point. This response will be used by the editors to evaluate your revision and sent back to the reviewers along with the revised manuscript.
- Highlight all changes made to your manuscript or provide us with a version that tracks changes.

Link Redacted

Thank you for the opportunity to review your work. Please do not hesitate to contact me if you have any questions or would like to discuss the required revisions further.

Sincerely,

Nature Human Behaviour

Reviewer expertise:

Reviewer #1: Human vision, neuroscience, fMRI, DNNs

Reviewer #2: Human vision, neuroscience, fMRI, DNNs

Reviewer #3: Human vision, neuroscience, fMRI, DNNs

REVIEWER COMMENTS:

Reviewer #1 (Remarks to the Author):

This paper uses an in silico brain model of visual responses to identify images that maximally stimulate and maximally disentangle early visual regions. They use this model to identify what features may be common and different in these 4 visual regions. Overall the paper appears to be rigorous and the results interesting and novel. However there are a few items that need to be modified in terms of the framing and placement of the paper and its findings to the larger field.

Overall, there lacks critical discussion of what has been done before in this area and what this paper does to extend it. No discussion of previous work that has done similar things is presented. Works are cited but no detail is provided on the works that are closest to this one. For example, there has been other work that 1) used encoding model predictions and input image generation to create/find images that were maximally co-activating or disentangling the activity of one, two and three brain regions and 2) show that encoding model predictions increase SNR in brain responses to images. See figures 8 and 3, respectively, in the cited Gu et al paper (neurogen). The bashivan paper also did something similar. Just from reading the text there is no detail provided on what previous work has done or found; this is critical for putting the current results in context.

This paper also provided a closed loop experiment using in silico brain response predictions and optimization of brain responses to create a set of images that was then tested for validity: Zijin Gu, Keith Jamison, Mert R. Sabuncu & Amy Kuceyeski. Human brain responses are modulated when exposed to optimized natural images or synthetically generated images. Communications Biology volume 6, Article number: 1076 (2023). Again please provide some discussion of how these results contribute to the larger field and how they relate to what has been done before.

I would also say that many of the joint activation images of these early visual regions have bright colors (oranges and reds), compared to the ones that are disentangling the activations (in both natural and generated images). In addition, the images that stimulate V1 (and V2) and suppress V4 (and V3) are almost always outside scenes with green trees/brush - this is even evident in the generated images. Perhaps add to discussion.

The motivation for adding the extra penalty term to the generated images (to reduce complexity?) is not provided. Complexity may indeed be a feature important to these early visual regions that respond to high frequency texture. More motivation for this is needed, and a comparison of what images are produced when it is not added should be made.

I also think there should be more than one image created, perhaps by relaxing the complexity constraint this would be possible. Even with the constraint it should be possible.

It may be worth making a figure in the main text summarizing the activating and disentangling images for each pair of regions. There could be information gleaned by looking at them together that cannot be by looking at them apart (for example that V1 and V2 respond specifically to what appears to be green vegetation that V3 and V4 do not respond to as much)

Figure 5c could be explained a bit more clearly - they are comparing what two things in the correlation that is being represented on the y axis? I think it is the elements of the RSA for each region for sky/nosky but it could be labeled more clearly.

Reviewer #1 (Remarks on code availability):

I took a cursory look and the code appear to be well documented and complete, with links to the original data when needed.

Reviewer #2 (Remarks to the Author):

This paper introduces relational neural control (RNC), demonstrating three key advances: (1) a novel method to identify images that align or disentangle neural responses between visual areas, (2) evidence that shared representational content decreases with cortical distance, and (3) identification of specific visual features driving shared versus unique representations. The validation of in silico predictions through independent in vivo fMRI experiments strengthens these findings. The thorough documentation and comprehensive supplementary materials support reproducibility.

The significance of this work extends beyond visual neuroscience. The RNC framework provides a novel approach for studying representational relationships that could generalize to other neural systems. The study demonstrates how combining in silico exploration with targeted empirical validation can accelerate neuroscientific discovery.

Considerations for revision:

1. Scope and title alignment

The title "across visual cortex" suggests a comprehensive analysis of the visual hierarchy, yet the study focuses on early-to-mid areas (V1-V4). Given that fMRI data for higher visual areas are publicly available and their representations are of current interest, extending analyses to higher visual areas would better reflect the title's scope and increase the paper's impact. If there are limitations regarding which areas RNC can analyze, these should be discussed explicitly.

2. Statistical Testing Clarification

The statistical methods section (lines 929-938) states "The sample sizes of these tests were either N = 8 subjects for tests of the in silico fMRI responses, or N = 6 for tests on the in vivo fMRI responses from the fMRI experiments..." This description requires clarification in two aspects:

First, the small subject numbers (N = 6 or 8) combined with Benjamini-Hochberg correction could limit statistical power for t-tests, warranting careful interpretation of the results. Second, there appears to be potential confusion between "sample size" and "number of comparisons." For improved clarity and reproducibility, the authors should:

- Clearly specify the actual number of samples used in each statistical test
- Detail the number of comparisons being corrected for in each analysis
- Justify the statistical power given these parameters
- Consider reporting effect sizes alongside p-values to better convey the strength of the findings

Additional suggestions for enhancement:

3. Choice of generator in "Generative univariate RNC" analysis

The study uses a GAN pre-trained on ImageNet validation images to generate stimulus images that align or disentangle in silico univariate fMRI responses for areas. Given recent advances in generative AI, particularly diffusion models which can produce more diverse and naturalistic samples, exploring RNC with such models could be valuable. Specifically, using diffusion models might generate more interpretable controlling images that better reveal the visual features driving neural responses and test the robustness of the current findings across different image generation approaches. This extension would strengthen the generalizability of the findings while leveraging state-of-the-art generative models to improve our understanding of visual representations.

4. Multivariate analysis interpretation

While multivariate fMRI responses should theoretically reveal more detailed representational differences between areas than univariate responses, the images that disentangle RSMs for V1 and V4 did not reveal common visual patterns (lines 326-328). The paper should clarify whether this conclusion is based on qualitative image inspection and discuss whether this suggests limitations of the RNC method or reflects genuine properties of neural representations.

5. Context within existing literature

To better highlight RNC's strengths and novelty, the discussion should relate the findings (high spatial frequencies and object-like shapes driving unique V1-V4 univariate responses, shared multivariate representations arising from common retinotopic properties) to existing literature studying V1 and V4 in isolation. This would help readers understand how RNC advances our understanding beyond traditional approaches.

6. Network-level visualization

While results are presented for each pair of visual areas, a network-level visualization would better support the paper's claims about "how visual areas jointly represent the world as an interconnected network" (abstract) and how "RNC extends existing anatomical and functional connectivity research assessing the brain as a complex interconnected network with the concept of representation" (discussion lines 398-400). Such visualization could help readers grasp the holistic nature of the findings.

Reviewer #2 (Remarks on code availability):

The code for reproducing the paper's results has been organized according to the order of main results presented in the paper. The authors also provide the tutorials for the RNC method that can be run with Jupyter Notebook/Google Colab via their GitHub repository.

Reviewer #3 (Remarks to the Author):

The paper describes RNC, an encoding model for generating fMRI responses to natural images that is used to investigate the relationships between neural responses in the early and intermediate visual areas. The model investigates a novel angle to approach the problem of understanding neural representations in visual processing. The paper itself is theoretically sound and the author have done their due diligence to ensure their work stands on a rigorous base. Additionally, I would like to commend the authors on a very clear explanation of their process especially with illustrations in every figure making the paper very clear. That being said, there are multiple areas of improvement that would make the paper's contribution more valuable to the scientific community.

Major points:

- The biggest issue in this work is that the use of NSD as a dataset for the model might not be the best choice for generalization, which the authors have acknowledged in the limitations. While it offers a deep stimulus set, it fails in the avenue of stimulus diversity, which makes the image set limited to a certain distribution of images. This problem has plagued many of the papers recently and was discussed in detail in the following preprint: <https://arxiv.org/abs/2405.10078>. Alternate datasets that offer more diversity would include the deep image reconstruction dataset which includes both natural and artificial image dataset (reference: <https://doi.org/10.1371/journal.pcbi.1006633>).
- One additional limitation is, given the model is feedforward, it might not work very well on stimuli that require top-down modulation. This can be tested on blurred images (ref: <https://doi.org/10.1523/ENEURO.0443-17.2018>) and illusory images (ref: <https://doi.org/10.1126/sciadv.adj3906>). This is an extensive list that might not all fit within the context of this work so you do not need to do all of them but additional testing of limitations would make the paper much richer at least in the in-silico model.
- Given models are independent across ROIs, it is expected that deeper areas end up having lower explained variance as model complexity remains the same but processing depth of voxels coming from V4 is larger than those in V1. Is there a reasoning for the choice of model depth? Previous studies have shown that that voxels coming from deeper regions within the visual processing hierarchy were more similar to deeper layer representations in DNN (refs: <https://doi.org/10.1523/JNEUROSCI.5023-14.2015> ; <https://www.nature.com/articles/ncomms15037>).
- The alignment of align image group does not appear to decrease with cortical processing distance whereas disentanglement increases (as shown in figure 2d and 4c). Given how the align images with both V1 increase and V4 increase show to have elements from of both the disentangle images, I think it is important to discuss that point and give a reasoning on why it alignment does not decrease.
- Figure 4d is hard to understand. The authors introduce the concepts of observed vs. target RSA scores for the first time here without proper explanation making the reader puzzled as to why the difference in r values decreases with area distance. It looks to me on first sight that multivariate patterns align more with cortical distance based on this figure which negates figure 4c.
- From looking at the multivariate align and disentangle analysis, I can see why there would not be a certain pattern since the algorithm can choose a set for visually similar images vs. a random set. Perhaps it is not the best way to conduct this analysis and it requires more constraints on the set of images put together in a set rather than letting the algorithm select randomly. One possible way is to make sets of images that are visually or contextually similar and testing them for the most align and disentangle.
- In figure 6d: what do the black circles in the background of the left figure represent?
- The authors should discuss the possible role of temporal processing. While fMRI loses most temporal information, it is possible that the disentanglement seen between V4 and V1 is due to a top-down suppression of signals as a result of object recognition (the aha moment) which was shown in multiple previous studies.

Minor points:

- I would appreciate having the model architecture in detail in the supplementary information for reproducibility.
- Page 18 line 537: Avoid mentioning that you followed the default values and explicitly mention them as these values could be different in newer versions of the same library.
- Page 14 line 357: c written but should be e

Reviewer #3 (Remarks on code availability):

python version not included. Advisable to create a requirements.txt file with libraries and their versions for a pip install or alternatively a yml file for anaconda environment creation.

Version 1:

Decision Letter:

Our ref: NATHUMBEHAV-24114428A

9th May 2025

Dear Dr Gifford,

Thank you for submitting your revised manuscript "In silico discovery of representational relationships across visual cortex" (NATHUMBEHAV-24114428A). It has now been seen by the original referees and their comments are below. As you can see, the reviewers find that the paper has improved in revision. We will therefore be happy in principle to publish it in Nature Human Behaviour, pending minor revisions to satisfy the referees' final requests and to comply with our editorial and formatting guidelines.

We are now performing detailed checks on your paper and will send you a checklist detailing our editorial and formatting requirements within two weeks. Please do not upload the final materials and make any revisions until you receive this additional information from us.

Sincerely,

Nature Human Behaviour

Reviewer #1 (Remarks to the Author):

The authors have done a good job of responding to the comments with new analyses and edits to the manuscript. I think the overall manuscript is much improved, and has wider impact.

Reviewer #1 (Remarks on code availability):

I do not have time to review the code in detail but a cursory review revealed no glaring issues - the code base appears well documented and complete.

Reviewer #2 (Remarks to the Author):

I thank the authors for addressing all my comments in a thorough and thoughtful manner. The manuscript has been significantly improved and the new results from additional analyses add valuable insights to the manuscript. I have only two minor suggestions for further clarification:

1. There is an inconsistency in the number of corrected tests (3) mentioned in the figure caption of Figure 4d compared to the methods section. For clarity, it is recommended to specify which tests were corrected. This has been done for some analyses (e.g., lines 1145-1146: "we corrected the p-values over 4 tests (one for each neural control condition).") but not for all.
2. Please verify the results presented in the leftmost plot in Fig. 7d (images driving both FFA and RSC, green condition). The depicted proximity between FFA and RSC appears weaker (shown by lower opacity connecting lines) compared to the all-image condition in Fig. 7a, which seems counterintuitive.

Reviewer #2 (Remarks on code availability):

The code repository has been updated to incorporate new code for the additional analyses presented in this revised manuscript.

Reviewer #3 (Remarks to the Author):

Thank you very much for sending the revised manuscript. The paper is much stronger now after the additions. I do not have any new requests but just wanted to add a few points in response:
- In regards to using the Illusory image reconstruction dataset for training, while it is a useful addition, the dataset is based on mostly images selected from ImageNet so it is relatively redundant to the ImageNet. My comment at the time was to use the

test images for non-natural images as OOD data which you already achieved using the synthetic-NSD dataset.

- I agree with your comment that it is not easy to discern whether the drop in performance is due to the training on natural images or because of other components such as top-down component (it is most-likely a mixture of everything). To actually measure the effect of top-down, that would be another analysis (such as one we did in Abdelhack and Kamitani 2018) that is beyond the scope of this work. I thought it was important to discuss these points in the discussion though as a guide to future studies which you did very well.

Thank you very much for this interesting read.
Mohamed Abdelhack

Reviewer #3 (Remarks on code availability):

The authors have addressed my previous comment on the code.

Dear Editor, Dear Reviewers,

Please find the detailed point-by-point response below. Editor and Reviewer comments are highlighted in gray, and the corresponding responses indented. Within the responses, extracts from the manuscript are “*quoted in italics*”, and extracts reflecting changes to the manuscript are “*quoted in italics and red font*”. Manuscript line/page/figure numbers are indicated in **bold font**. Blue underscored text denotes hyperlinks.

Editor

Thank you once again for your manuscript, entitled "In silico discovery of representational relationships across visual cortex", and for your patience during the peer review process.

Your Article has now been evaluated by 3 referees. You will see from their comments copied below that, although they find your work of considerable potential interest, they have raised quite substantial concerns. In light of these comments, we cannot accept the manuscript for publication, but would be interested in considering a revised version if you are willing and able to fully address reviewer and editorial concerns.

We hope you will find the referees' comments useful as you decide how to proceed. If you wish to submit a substantially revised manuscript, please bear in mind that we will be reluctant to approach the referees again in the absence of major revisions. We are committed to providing a fair and constructive peer-review process. Do not hesitate to contact us if there are specific requests from the reviewers that you believe are technically impossible or unlikely to yield a meaningful outcome.

In particular, Reviewers #1 and #2 remark on the lack of engagement on how the present findings relate to prior work. Given the existence of similar methods, we ask that you address the issues related to the advance in the context of the earlier literature. Reviewer #2 also raises important concerns relating to statistical testing and requests the extension of the analyses to higher visual areas, which we ask you to perform. Finally, Reviewer #3 questions the generalizability of the results. We ask that you follow their recommendations to demonstrate robustness and generalization.

We thank the Editor for considering a revised version of the manuscript. We found the Reviewers' comments to be extremely useful. Addressing these comments led to the introduction of two new Results sections, to the addition of several new control analyses including the discussion of their implications. Following we summarize the major changes of the revision.

(1) Relation to previous work | We contextualized RNC within previous related work, explicitly stating which aspects of the RNC approach build on recent innovations in neural control, or which aspects are instead new additions to neural control research [Reviewer #1 points 1-2]. Furthermore, we also elaborated on the advantages of jointly investigating areas with RNC, as opposed to investigating them in isolation following more traditional approaches [Reviewer #2 point 5].

(2) Extension to high-level visual areas | We added a new Results section where we extended RNC to high-level cortical visual areas. Through RNC we discovered controlling images that align or disentangle the in silico univariate and multivariate fMRI responses of high-level areas, thus demonstrating RNC's applicability across the visual cortical network [Reviewer #2 point 1].

(3) Statistical testing | We replaced the significance test of the population mean effect with population prevalence tests, which are especially well suited for small sample sizes of densely sampled subjects as used here. Furthermore, we improved the clarity and level of detail of the statistical significance reportings [Reviewer #2 point 2].

(4) Robustness and generalization of RNC | We applied RNC on the in silico neural responses generated from encoding models trained on the Visual Illusion Reconstruction dataset. This replicated our original results, thus strengthening the robustness of RNC and the generalization of its solutions [Reviewer #3 point 1]. Furthermore, we tested the out-of-distribution generalization performance of the encoding models used to generate the in silico responses, obtaining prediction accuracies well above chance, which suggest the generalizability of RNC's solutions across very diverse visual distributions [Reviewer #3 point 2].

(5) Network-level visualizations | We added a new Results section where we provide a network-level analysis and visualization of the representational relationships discovered for individual pairwise comparisons of areas, thus revealing a unified picture of how visual areas jointly represent the world as an interconnected network. [Reviewer #2 point 6].

Besides these minor changes, we added several clarifications, control analyses, and discussions throughout the manuscript that improved the clarity, validity, and reproducibility of RNC.

Reviewer #1

Remarks to the Author

This paper uses an in silico brain model of visual responses to identify images that maximally stimulate and maximally disentangle early visual regions. They use this model to identify what features may be common and different in these 4 visual regions. Overall the paper appears to be rigorous and the results interesting and novel. However there are a few items that need to be modified in terms of the framing and placement of the paper and its findings to the larger field.

We thank Reviewer #1 for the positive evaluation and helpful comments that resulted in valuable additional analyses and discussions. In brief, we:

(1) Added a paragraph to the Discussion section where we contextualize RNC within previous related work in neural control.

(2) Implemented variants of the generative univariate RNC analysis (i.e., generation of multiple images, relaxation of the complexity reduction objective), which replicated our original results, thus corroborating our findings.

(3) Further discussed what are the image visual features leading to aligned or disentangled responses.

(4) Added the controlling images for all pairwise comparisons of areas in the same figures, to facilitate comparisons of similarities and differences between the representational content of different areas.

(5) Improved the clarity of text and figures across the manuscript.

1. Overall, there lacks critical discussion of what has been done before in this area and what this paper does to extend it. No discussion of previous work that has done similar things is presented. Works are cited but no detail is provided on the works that are closest to this one. For example, there has been other work that 1) used encoding model predictions and input image generation to create/find images that were maximally co-activating or disentangling the activity of one, two and three brain regions and 2) show that encoding model predictions increase SNR in brain responses to images. See figures 8 and 3, respectively, in the cited Gu et al paper (neurogen). The bashivan paper also did something similar. Just from reading the text there is no detail provided on what previous work has done or found; this is critical for putting the current results in context.

We agree with the Reviewer that putting RNC in context with related work is important for readers to understand what has been done before in this area and what RNC does to extend it. We therefore added the following paragraph to the Discussion section (**Discussion, page 22, lines 554-571**):

“RNC builds on recent innovations in neural control, a paradigm used to find controlling stimuli that elicit a neural response state of interest. To increase the solution space and allow for rapid exploration, the controlling stimuli are found using large amounts of in silico neural responses generated by encoding models (Lehky et al., 1992; Bashivan et al., 2019; Walker et al., 2019; Gu et al., 2022; Gu et al., 2023). Next, to ensure that the controlling stimuli truly elicit the neural response state of interest, these stimuli are empirically validated on in vivo neural responses (Bashivan et al., 2019; Walker et al., 2019; Gu et al., 2022). Finally, to elicit more complex neural response states of interest, multiple neurons or areas are jointly controlled (Bashivan et al., 2019; Gu et al., 2022). Building on these innovations, RNC extends neural control research in two ways. First, RNC uses neural control to enable a network-level characterization of the representational relationships between visual cortical areas. Second, to investigate representational relationships at multiple brain response levels, RNC jointly applies neural control on univariate and multivariate neural responses. We found that the visual features aligning or disentangling V1 and V4’s univariate responses (i.e., spatial frequency and object-like shapes) are different from the ones aligning and

disentangling their multivariate responses (i.e., topological image properties). Thus, the univariate and multivariate response levels captured complementary aspects of representational relationships between areas, suggesting that visual cortex multiplexes diverse neural codes for visual information processing (deCharms & Zador, 2000; Perkel & Bullock, 1968; Dumoulin & Wandell, 2008) and, in turn, encouraging the integrated analysis of diverse neural response levels.”

We additionally thank the Reviewer for pointing out that the NeuroGen paper also shows that encoding model predictions increase SNR in brain responses to images. In light of this, we cited the NeuroGen paper in the following relevant passages.

(Results, page 4, lines 93-95): *“Finally, since neural noise is not predictable from the stimulus images, encoding models modeled the signal- and not noise-related variability of the neural response (Wu et al., 2006; Gu et al., 2022), thus resulting in silico fMRI responses less affected by noise compared to the NSD responses.”*

(Discussion, page 23, lines 579-581): *“Moreover, encoding models generate in silico neural responses which are less affected by noise compared to in vivo responses, thus reducing the effect of noise on results (Wu et al., 2006; Gu et al., 2022).”*

(Methods, page 27, lines 718-720): *“Because neural noise is not predictable from the stimulus images, encoding models model the signal- and not noise-related variability of the neural response¹⁹, thus resulting in silico fMRI responses less affected by noise compared to in vivo responses (Wu et al., 2006; Gu et al., 2022).”*

(Supplementary Figure 2): *“Because the in silico neural responses did not capture all signal variance in the NSD responses, the in silico neural responses explaining more variance than the in vivo NSD responses would be indicative of the former being less affected by noise (Wu et al., 2006; Gu et al., 2022).”*

2. This paper also provided a closed loop experiment using in silico brain response predictions and optimization of brain responses to create a set of images that was then tested for validity: Zijin Gu, Keith Jamison, Mert R. Sabuncu & Amy Kuceyeski. Human brain responses are modulated when exposed to optimized natural images or synthetically generated images. Communications Biology volume 6, Article number: 1076 (2023). Again please provide some discussion of how these results contribute to the larger field and how they relate to what has been done before.

In the new Discussion paragraph referred to in the previous review point (review point 1) we also mention and cite previous work where in silico experimentation is followed by in vivo validation (Bashivan et al., 2019; Walker et al., 2019; Gu et al., 2023). There we also discuss how, building on this seminal work, RNC extends neural control research.

3. I would also say that many of the joint activation images of these early visual regions have bright colors (oranges and reds), compared to the ones that are disentangling the activations (in both natural and generated images). In addition, the images that stimulate V1 (and V2)

and suppress V4 (and V3) are almost always outside scenes with green trees/brush - this is even evident in the generated images. Perhaps add to discussion.

We thank the Reviewer for bringing this up. We believe that the visual features actually controlling univariate fMRI responses are high spatial frequency content and object-like shape, and that green vegetation and warm-color cluttered food items are just visual features that, in the NSD images, co-occur with the controlling visual features. For example, in NSD highly cluttered object images with high spatial frequencies tend to be images with warm-color food items, and images with high spatial frequencies but little or no objects tend to be images with green vegetation. Supporting this is the finding that, when applying RNC on in silico univariate fMRI responses for the 50,000 ImageNet 2012 challenge validation split images, or for the 26,107 THINGS or images, the controlling images still consist of combinations of high spatial frequencies and objects, but not necessarily of warm-color cluttered food items, or green vegetation. In light of these observations, we edited the displayed example controlling images in (**Supplementary Figures 5-6**) to highlight that the controlling images always consist of combinations of high spatial frequencies and objects, but not always of warm-color cluttered food items, or green vegetation. Furthermore, we edited the corresponding Results paragraph as follows (**Results, page 9, lines 202-218**):

“The controlling images driving V1 while suppressing V4 responses contained high spatial frequency backgrounds (e.g., green vegetation), whereas the controlling images driving V4 while suppressing V1 responses contained one or multiple objects on a low spatial frequency background (e.g., a plane on a sky background). Controlling images driving or suppressing both areas simultaneously were the logical combination thereof: high spatial frequency and objects were present in controlling images driving the response of both areas (e.g., warm-color cluttered food items), whereas they were lacking in controlling images suppressing the response of both areas (e.g., empty skies) (Fig. 3a). As expected, we discerned no consistent visual patterns in the baseline images (Fig. 3a). When using alternative distributions of images and of encoding model training data, the resulting controlling images also consisted of combinations of high spatial frequencies and objects (Supplementary Figures 5-7). However, they did not always contain green vegetation, planes on a sky background, or warm-color cluttered food items (as was the case with the NSD images in Fig. 3a), suggesting that these visual categories correlate with, but are not, the visual features controlling univariate responses. This showed, through large-scale exploratory analysis using naturalistic images from diverse image sets, that V1 is uniquely tuned to high spatial frequency content (Foster et al., 1985; Kay et al., 2008), whereas V4 is uniquely tuned to object-like shapes (Kobatake & Tanaka, 1994).”

“Supplementary Figure 5 | Results of univariate RNC applied on the *in silico* fMRI responses for the 50,000 ImageNet images, generated through encoding models trained on NSD. a, Univariate RNC quantitative results (univariate response magnitudes), embedded in a four-by-four matrix. b, Stepwise distance between areas. c, Absolute difference between controlling and baseline image univariate responses, averaged across all pairwise comparisons of areas with same stepwise distance. d, Correlation between the univariate responses of two areas, averaged across pairwise comparisons of areas with same stepwise distance. e, Univariate RNC controlling and baseline images for the V1 vs. V4 comparison.”

“Supplementary Figure 6 | Results of univariate RNC applied on the in silico fMRI responses for the 26,107 THINGS images, generated through encoding models trained on NSD. a, Univariate RNC quantitative results (univariate response magnitudes), embedded in a four-by-four matrix. b, Stepwise distance between areas. c, Absolute difference between controlling and baseline image univariate responses, averaged across all pairwise comparisons of areas with same stepwise distance. d, Correlation between the univariate responses of two areas, averaged across pairwise comparisons of areas with same stepwise distance. e, Univariate RNC controlling and baseline images for the V1 vs. V4 comparison.”

4. The motivation for adding the extra penalty term to the generated images (to reduce complexity?) is not provided. Complexity may indeed be a feature important to these early visual regions that respond to high frequency texture. More motivation for this is needed, and a comparison of what images are produced when it is not added should be made.

We thank the Reviewer for highlighting that our goals with the generative univariate RNC analysis can be better clarified. If our goal with generative univariate RNC was to align or disentangle V1 and V4 as much as possible, adding the extra complexity penalty would not be needed. However, our goal with generative univariate RNC is to *isolate* the visual features that well align or disentangle the two areas, which is in practice achieved through two serial objectives (each with its own penalty). The first objective is to generate images that align and disentangle V1 and V4 up to a threshold. Once the threshold is reached, the second objective activates, which is to reduce the complexity of the generated images, *while keeping the neural control scores above threshold*. Thus, the resulting images will both well align or disentangle V1 and V4, and also be as simple as possible, therefore isolating the controlling visual features of interest.

Furthermore, to address the following related review point that also concerns generative univariate RNC (review point 5), for each neural control condition we ran 10 independent evolutions, each based on a different random seed. The random seed determined the initial latent vectors, as well as the new latent vectors produced by the genetic optimization, resulting in 10 genetically optimized images for each control condition.

To better clarify our analysis goal, and to introduce the 10 independent evolutions, we edited the corresponding Results section (**Results, page 9, lines 219-234**):

*“Naturalistic images are complex combinations of multiple visual features making it challenging to isolate, by mere visual inspection, the features leading to aligned or disentangled responses across areas. To further isolate the relevant visual features, we generated de novo controlling images that controlled univariate responses, while being as simple as possible. To this end, we combined RNC with an image generator (Dosovitskiy & Brox, 2016) and genetic optimization (Ponce et al., 2019; Xiao & Kreiman, 2020) to iteratively generate images following two serial objectives. The first objective, active throughout the entire optimization procedure, was to generate images controlling (i.e., driving or suppressing) in silico univariate fMRI responses of V1 and V4 up to a threshold. Once this threshold was reached, the second objective became activated, which was to lower image complexity as measured by the images’ PNG compression file size (Marin & Leder, 2013; Mayer & Landwehr, 2018), while at the same time keeping the univariate responses above threshold. This promoted the generation of **controlling images (first objective)** containing only the visual features strictly necessary to align or disentangle in silico univariate fMRI responses (**second objective**) (**Fig. 3b**; the generative univariate RNC algorithm is visualized in **Supplementary Figure 8**; for a fine-grained progression of images across generations see **Supplementary Figure 9**). **For each neural control condition we ran ten independent evolutions, resulting in ten genetically optimized images for each condition.**”*

“Fig. 3 | Spatial frequency and object-like shapes determine unique representational content for V1 and V4 in silico univariate fMRI responses. b, Results of ten independent generative univariate RNC evolutions using in silico fMRI responses averaged across all 8 subjects. For each neural control condition, the plots show the in silico univariate fMRI responses (represented by colored lines) and the PNG compression file size (represented by black lines) for the best GAN-generated image of each genetic algorithm generation, averaged across the ten evolutions. The vertical dashed lines indicate the generation where the univariate response threshold is reached (also averaged across evolutions), after which PNG compression file size starts decreasing. On top of each plot are the optimized images from the ten evolutions.”

Because during the second objective (complexity reduction) the neural control scores are required to stay above threshold, if complexity is indeed important to align or disentangle V1 and V4, then the image complexity will simply not decrease too much (or at all) during the optimization generations. Importantly, the complexity optimization does not compromise the neural control scores, and therefore does not prevent generating images that successfully align or disentangle the two areas.

Having clarified this, omitting the complexity penalty could lead to informative results. Indeed, running generative univariate RNC without the complexity penalty yielded qualitatively similar controlling images, which we believe corroborated the findings of generative univariate RNC using the complexity term. These findings were further corroborated by obtaining similar results across the 10 independent evolutions, as explained in the edited Results passage below (**Results, page 9, lines 235-246**):

“Inspection of the genetically optimized images converged with the insights previously gained by naturalistic images. The genetically optimized images driving V1 while suppressing V4 consisted of an uniform high spatial frequency pattern, whereas the

images driving V4 while suppressing V1 consisted of multiple small object-like shapes on a uniform background. The images driving or suppressing both areas were again logical combinations of the previous cases: the images driving both areas consisted of many small object-like shapes clustered together, and the images suppressing both areas consisted of a uniform white background. **The fact that all ten generated images within each neural control condition were strikingly similar to each other indicates that these controlling visual features are the ones optimally aligning or disentangling V1 and V4. Further supporting this, generating images without the image complexity constraint led to images that, albeit visually more complex, still contained the same controlling visual features (Supplementary Figure 10).**

“Supplementary Figure 10 | Generative univariate RNC without image complexity reduction. Results of ten independent generative univariate RNC evolutions using in silico fMRI responses averaged across all 8 subjects. Across the 500 genetic algorithm generations, the images are only optimized to improve the neural control scores (i.e., the PNG compression file size is not reduced). For each neural control condition, the plots show the in silico univariate fMRI responses (represented by colored lines) and the PNG compression file size (represented by black lines) for the best GAN-generated image of each genetic algorithm generation, averaged across evolutions. On top of each plot are the optimized images from the ten evolutions.”

5. I also think there should be more than one image created, perhaps by relaxing the complexity constraint this would be possible. Even with the constraint it should be possible.

We addressed this point in the response to the previous review point (review point 4), where we describe running 10 independent evolutions for each control condition, thus resulting in 10 images per control condition.

6. It may be worth making a figure in the main text summarizing the activating and disentangling images for each pair of regions. There could be information gleaned by looking at them together that cannot be by looking at them apart (for example that V1 and V2 respond specifically to what appears to be green vegetation that V3 and V4 do not respond to as much).

We agree that presenting the controlling images from multiple pairwise area comparisons and control conditions together would be informative. Since we already have 8 figures in the main Results text (which is Nature Human Behavior's limit), and since showing the controlling images for all pairwise area comparisons requires multiple figures, we added these images to the Supplementary Information (**Supplementary Figures 22-25**). These supplementary figures also include controlling images for high-level visual areas (for details on the high-level visual areas, see response to review point 1 of Reviewer #2), and the multidimensional scaling (MDS) embeddings of the in silico fMRI responses (for details on the MDS embeddings, see response to review point 6 of Reviewer #2). Given the already extensive supplementary material, we decided not to include controlling images for the interaction between early-, mid-, and high-level visual areas. We reproduce these supplementary figures below.

Finally, because the amount of controlling images is larger than what we could fit in these figures (i.e., 25 images for each univariate RNC control condition, and 50 images for each multivariate RNC control condition), we made all controlling images for all pairwise comparisons of areas available on OpenNeuro (<https://openneuro.org/datasets/ds005503>).

“**Supplementary Figure 22 | Multidimensional scaling (MDS) embeddings of the in silico univariate fMRI responses for controlling images found by applying univariate RNC on early- and mid-level visual areas.** Six exemplar controlling images are shown for each control condition (all images come from the 73,000 NSD images). Each panel reflects MDS results and controlling images for a different pairwise comparison of areas. **a**, V1 vs. V2 comparison. **b**, V1 vs. V3 comparison. **c**, V1 vs. V4 comparison. **d**, V2 vs. V3 comparison. **e**, V2 vs. V4 comparison. **f**, V3 vs. V4 comparison.”

“Supplementary Figure 23 | Multidimensional scaling (MDS) embeddings of the in silico univariate fMRI responses for controlling images found by applying univariate RNC on high-level visual areas. Six exemplar controlling images are shown for each control condition (all images come from the 73,000 NSD images). Each panel reflects MDS results and controlling images for a different pairwise comparison of areas. a, EBA vs. FFA comparison. b, EBA vs. PPA comparison. c, EBA vs. RSC comparison. d, FFA vs. PPA comparison. e, FFA vs. RSC comparison. f, PPA vs. RSC comparison.”

“**Supplementary Figure 24 | Multidimensional scaling (MDS) embeddings of the in silico multivariate fMRI responses for controlling images found by applying multivariate RNC on early- and mid-level visual areas.** Twelve exemplar controlling images are shown for each control condition (all images come from the 73,000 NSD images). Each panel reflects MDS results and controlling images for a different pairwise comparison of areas. **a**, V1 vs. V2 comparison. **b**, V1 vs. V3 comparison. **c**, V1 vs. V4 comparison. **d**, V2 vs. V3 comparison. **e**, V2 vs. V4 comparison. **f**, V3 vs. V4 comparison.”

“Supplementary Figure 25 | Multidimensional scaling (MDS) embeddings of the in silico multivariate fMRI responses for controlling images found by applying multivariate RNC on high-level visual areas. Twelve exemplar controlling images are shown for each control condition (all images come from the 73,000 NSD images). Each panel reflects MDS results and controlling images for a different pairwise comparison of areas. a, EBA vs. FFA comparison. b, EBA vs. PPA comparison. c, EBA vs. RSC comparison. d, FFA vs. PPA comparison. e, FFA vs. RSC comparison. f, PPA vs. RSC comparison.”

7. Figure 5c could be explained a bit more clearly - they are comparing what two things in the correlation that is being represented on the y axis? I think it is the elements of the RSA for each region for sky/nosky but it could be labeled more clearly.

That is correct, **Fig. 5c** is showing the mean Pearson’s r scores across all RSM comparisons of two sky images, two no sky images, or sky and no sky images. To make this clearer, we relabeled the y-axis to **“Mean RSM Pearson’s r ”**.

Remarks on code availability

I took a cursory look and the code appear to be well documented and complete, with links to the original data when needed.

We thank the Reviewer for checking the code.

Reviewer #2

Remarks to the Author

This paper introduces relational neural control (RNC), demonstrating three key advances: (1) a novel method to identify images that align or disentangle neural responses between visual areas, (2) evidence that shared representational content decreases with cortical distance, and (3) identification of specific visual features driving shared versus unique representations. The validation of in silico predictions through independent in vivo fMRI experiments strengthens these findings. The thorough documentation and comprehensive supplementary materials support reproducibility.

The significance of this work extends beyond visual neuroscience. The RNC framework provides a novel approach for studying representational relationships that could generalize to other neural systems. The study demonstrates how combining in silico exploration with targeted empirical validation can accelerate neuroscientific discovery.

We thank Reviewer #2 for the positive evaluation and helpful comments that resulted in valuable additional analyses and discussions. In brief, we:

(1) Added a new Results section where we extended RNC to high-level cortical visual areas, demonstrating RNC's applicability across the visual cortical network.

(2) Replaced the significance test of the population mean effect with population prevalence tests (which are especially well suited for small sample sizes of densely sampled subjects as used here), and improved the clarity and level of detail of the statistical significance reportings.

(3) Added a new Results section where we provide a network-level analysis and visualization of the representational relationships discovered for individual pairwise comparisons of areas.

(4) Added a paragraph to the Discussion section on the advantages of jointly investigating areas with RNC, compared to investigating them in isolation following traditional approaches.

(5) Discussed a possible reason why the images controlling multivariate responses are less interpretable than the ones controlling univariate responses.

Considerations for revision:

1. Scope and title alignment

The title "across visual cortex" suggests a comprehensive analysis of the visual hierarchy, yet the study focuses on early-to-mid areas (V1-V4). Given that fMRI data for higher visual areas are publicly available and their representations are of current interest, extending analyses to higher visual areas would better reflect the title's scope and increase the paper's impact. If there are limitations regarding which areas RNC can analyze, these should be discussed explicitly.

We thank the Reviewer for this excellent suggestion, which led to a whole new results section featuring successful application of RNC on in silico univariate and multivariate fMRI response of high-level visual areas, thus demonstrating RNC's applicability across visual cortex. Following we present the text from this new Results section (**Results, page 15, lines 355-423**):

"RNC controls in silico univariate and multivariate fMRI responses across high-level visual cortical areas

Fig. 6 | RNC controls in silico univariate and multivariate fMRI responses across high-level visual cortical areas. *a*, Univariate RNC results for each pairwise comparison of areas, embedded in a four-by-four matrix. The upper triangle of the results matrix shows the univariate responses for the controlling images against the baseline. Diamonds and squares indicate the univariate responses of the areas indexed by the rows and columns of the results matrix, respectively. Asterisks indicate neural control conditions for which the in silico univariate fMRI responses for the controlling images are significantly different from baseline (within-subject permutation test, $p < 0.05$, Benjamini/Hochberg corrected over 8 tests for each pairwise comparison of areas; population prevalence test, $p < 0.01$, indicating within-subject significance in at least 3/8 subjects). The lower triangle of the results matrix shows the univariate response image manifolds. Colored dots indicate in silico univariate fMRI responses averaged across the controlling images of each neural control condition, and small black points indicate in silico univariate fMRI responses of all

subjects for all 73,000 NSD images. Vertical and horizontal dashed lines indicate subject-average univariate response baseline for each area. **b**, Multivariate RNC results, consisting of RSA scores (Pearson's r) for each pairwise comparison of areas. Asterisks indicate neural control conditions for which the RSA scores from the controlling images are significantly higher (alignment) or lower (disentanglement) than baseline (within-subject permutation test, $p < 0.05$, Benjamini/Hochberg corrected over 2 tests for each pairwise comparison of areas; population prevalence test, $p < 10^{-9}$, indicating within-subject significance in at least 7/8 subjects). **c**, Categorical selectivity groups. Solid and dashed lines represent within- and between-group area comparisons, respectively. **d**, Absolute difference between controlling and baseline image univariate responses, averaged across within- or between-group area comparisons. Connectors indicate significant differences (within-subject permutation test, $p < 0.05$, Benjamini/Hochberg corrected over 4 tests; population prevalence test, $p < 0.001$, indicating within-subject significance in at least 4/8 subjects). **e**, Multivariate RNC RSA scores, averaged across within- or between-group area comparisons. Connectors indicate significant differences (within-subject permutation test, $p < 0.05$, Benjamini/Hochberg corrected over 3 tests; population prevalence test, $p < 0.01$, indicating within-subject significance in at least 3/8 subjects). Opaque and transparent diamonds/squares/dots represent subject-average and single subject results, respectively. Error bars reflect 95% confidence intervals.

Next, we extended RNC from early- and mid- to high-level visual areas. Using NSD, we trained encoding models of high-level visual areas that play a key role in the representation of bodies (EBA (Downing et al., 2021)), faces (FFA (Kanwisher et al., 1997)), scenes (PPA (Epstein & Kanwisher, 1998)), and in visual navigation (RSC (Maguire, 2001)) (encoding accuracies are shown in **Supplementary Figure 3**). Through RNC, we found controlling images that successfully aligned or disentangled both univariate (**Fig. 6a**) and multivariate (**Fig. 6b**) in silico fMRI responses for the 73,000 NSD images generated through these encoding models (within-subject permutation test, $p < 0.05$, Benjamini/Hochberg corrected over 8 or 2 tests for each univariate or multivariate RNC pairwise comparison of areas respectively; population prevalence test, $p < 0.01$, indicating within-subject significance in at least 3/8 subjects) (RNC results for interactions between early-, mid-, and high-level visual areas are shown in **Supplementary Figures 17-19**). Thus, we successfully aligned or disentangled different high-level visual cortical areas at the univariate and multivariate response levels.

EBA, FFA, PPA, and RSC fall within two broader groups of categorical selectivity: animate objects (EBA and FFA), and scenes (PPA and RSC). This suggests that alignment should be higher and disentanglement lower for within-group areas than for between-group areas (**Fig. 6c**). We confirmed this prediction. For univariate RNC, the absolute difference between the in silico univariate fMRI responses in the control conditions and the baseline was larger for within-group areas in the case of alignment (**Fig. 6d**, green and blue dots), and larger for between-group areas in the case of

disentanglement (Fig. 6d, yellow and red dots) (within-subject permutation test, $p < 0.05$, Benjamini/Hochberg corrected over 4 tests; population prevalence test, $p < 0.001$, indicating within-subject significance in at least 4/8 subjects). This indicates that the responses of within-group areas were more aligned and less disentangled, compared to between-group areas. Strengthening this finding, the univariate responses of within-group areas were strongly correlated, whereas the responses of between-group areas were anticorrelated (Fig. 6a, lower triangle of the results matrix). Similarly, for multivariate RNC the RSA scores for the aligning, disentangling, and baseline images were higher for within-group areas (Fig. 6e) (within-subject permutation test, $p < 0.05$, Benjamini/Hochberg corrected over 3 tests; population prevalence test, $p < 0.01$, indicating within-subject significance in at least 3/8 subjects). We observed quantitatively similar results when using alternative distributions of images (Supplementary Figures 20-21).

Together, through RNC we discovered controlling images that align or disentangle the in silico univariate and multivariate fMRI responses of high-level visual areas. This demonstrated RNC's applicability across the visual cortical network, and revealed that shared representational content is higher and unique representational content lower for high-level visual areas with similar categorical selectivity."

Finally, the interactive Google Colab tutorials that we created to facilitate the adoption of RNC allow users to apply RNC on in silico fMRI responses from a selection of 23 areas spanning the entire visual cortex.

2. Statistical Testing Clarification

The statistical methods section (lines 929-938) states "The sample sizes of these tests were either $N = 8$ subjects for tests of the in silico fMRI responses, or $N = 6$ for tests on the in vivo fMRI responses from the fMRI experiments..." This description requires clarification in two aspects:

First, the small subject numbers ($N = 6$ or 8) combined with Benjamini-Hochberg correction could limit statistical power for t-tests, warranting careful interpretation of the results. Second, there appears to be potential confusion between "sample size" and "number of comparisons." For improved clarity and reproducibility, the authors should:

- Clearly specify the actual number of samples used in each statistical test
- Detail the number of comparisons being corrected for in each analysis
- Justify the statistical power given these parameters
- Consider reporting effect sizes alongside p-values to better convey the strength of the findings

These are important points that made us critically re-evaluate our statistics to provide firmer inference. Because our experiments involved small sample sizes of densely sampled subjects, we replaced the significance test of the population mean effect with population prevalence tests (Ince et al., 2021; Ince et al., 2022), a two-level procedure that is especially well suited for small sample sizes of densely sampled subjects as used here. This is because, at the first level, significance is determined within each individual subject (e.g., with non-parametric permutation tests), effectively increasing power due to the high amount of data available for each subject. At the second level, the binary results from the first level (i.e., the counts of significant subjects) are used to

estimate the probability of this happening by chance, under the null hypothesis of no effect in any member of the population, thus providing a population-level inference. Based on this two-step procedure, we rejected the null hypothesis of no effect in any member of the population with p -values ranging from 0.01 to 10^{-9} . Following is the updated text from the “Statistical testing” Methods section (**Methods, page 37, lines 1120-1177**):

“We assessed statistical significance using population prevalence testing (Ince et al., 2021; Ince et al., 2022), which is well suited to determine significance of an effect at the level of the population when analyzing small samples of intensely scanned subjects. Population prevalence testing is a two-level procedure. At the first level, significance is established independently within each subject. At the second level, the binary results from the first level (i.e., the counts of significant subjects) are used to estimate the probability of this happening by chance, under the null hypothesis of no effect in any member of the population, thus providing a population-level inference. Following we describe these two levels in detail.

At the first level, we established significance independently within each subject using non-parametric permutation tests. Each test consisted in: computing the statistic of interest (i.e., the observed statistic); creating the null distribution of the observed statistic by recomputing it using 100,000 different random permutations of the data; quantifying the p -value as the proportion of permutations where the randomized statistic is as extreme or more extreme than the observed statistic; controlling familywise error rate by applying (non-negative) Benjamini/Hochberg correction (Benjamini & Hochberg, 1995) to the resulting p -values; assigning significance to subjects with corrected p -values below 0.05. In the encoding models noise analysis tests the null hypothesis was that the noise-ceiling-normalized explained variance scores of the different predictors were equal, we permuted the encoding accuracy scores over fMRI voxels and predictors, and we corrected the p -values over 2 tests for each area. In the univariate RNC analysis tests the null hypothesis was that the univariate responses for the controlling and baseline images were equal, we permuted the univariate responses across image conditions, and we corrected the p -values over 8 tests for each pairwise comparison of areas. In the univariate RNC cortical distance analysis tests the null hypothesis was that the absolute differences between the baseline univariate responses and the univariate responses for controlling images from different stepwise area distances were equal, we permuted the univariate responses across areas, and we corrected the p -values over 4 tests (one for each neural control condition). In the multivariate RNC analysis tests the null hypothesis was that the RSA scores for the controlling and baseline images were equal, we permuted the multivariate responses across image conditions, and we corrected the p -values over 2 tests for each pairwise comparison of areas. In the multivariate RNC cortical distance analysis tests the null hypothesis was that the absolute differences between the baseline RSA scores and the RSA scores for controlling images from different stepwise area distances were equal, we permuted the multivariate responses across areas and image conditions, and we corrected the p -values over 2 tests (one for each neural control condition). In the multivariate RNC retinotopy analysis tests the null hypothesis was that the univariate responses of voxels tuned to the lower and upper portions of the visual field were equal, we permuted the data across ventral and dorsal

voxels, and we corrected the p -values over 2 tests for each area. In the univariate RNC categorical selectivity analysis tests the null hypothesis was that the absolute differences between the baseline univariate responses and the univariate responses for controlling images from areas within or between categorical selectivity groups were equal, we permuted the univariate responses across areas, and we corrected the p -values over 4 tests (one for each neural control condition). In the multivariate RNC categorical selectivity analysis tests the null hypothesis was that the absolute differences between the baseline RSA scores and the RSA scores for controlling images from areas within or between categorical selectivity groups were equal, we permuted the univariate responses across areas and image conditions, and we corrected the p -values over 2 tests (one for each neural control condition).

At the second level, we used the cumulative density function of the binomial distribution of within-participant significance to estimate the probability of observing significant subjects by chance, if there was no effect in any member of the population:

$$p = 1 - CDF(k, n, a = 0.05)$$

where CDF is the cumulative density function of the binomial distribution, k is the number of significant subjects, n is the total number of subjects (8 or 7 subjects for tests on the *in silico* fMRI responses from encoding models trained on NSD and VIR, respectively; 6 subjects for tests on the *in vivo* fMRI responses from the fMRI experiments), and a is the probability of success in each trial under the null hypothesis. Finally, we assigned statistical significance at the population-level for probabilities of $p < 0.05$.”

To calculate the confidence intervals of each statistic, we created 100,000 bootstrapped samples by sampling the subject-specific results with replacement. This yielded empirical distributions of the results, from which we derived the 95% confidence intervals.”

We also changed the statistical significance statements throughout the entire manuscript to reflect these two levels of population prevalence testing. Furthermore, following the Reviewer’s suggestion, we made explicit the sample size and number of comparisons being corrected for in each test. Following is an example of a new statistical significance statement (**Results, page 12, lines 282-285**):

“(within-subject permutation test, $p < 0.05$, Benjamini/Hochberg corrected over 2 tests for each pairwise comparison of areas; population prevalence test, $p < 10^{-9}$, indicating within-subject significance in at least 7/8 subjects)”

Finally, because most figures include a large number of tests (e.g., 53 tests in Figure 2 alone), we did not report the effect sizes along the p -values. However, we believe this lack is mitigated by the effect sizes being visually available from the result plots.

Additional suggestions for enhancement:

3. Choice of generator in “Generative univariate RNC” analysis

The study uses a GAN pre-trained on ImageNet validation images to generate stimulus images that align or disentangle in silico univariate fMRI responses for areas. Given recent advances in generative AI, particularly diffusion models which can produce more diverse and naturalistic samples, exploring RNC with such models could be valuable. Specifically, using diffusion models might generate more interpretable controlling images that better reveal the visual features driving neural responses and test the robustness of the current findings across different image generation approaches. This extension would strengthen the generalizability of the findings while leveraging state-of-the-art generative models to improve our understanding of visual representations.

This is an excellent idea. We explored two principled venues: using class-unconditioned and class-conditioned diffusion models. In either case, each generative univariate RNC evolution consists in the generation of 375,250 images (i.e., (1 generation × 1,000 images per generation) + (499 generation × 750 images per generation) => since after the first generations the 25% best images from the previous generations are kept). And because there are four univariate RNC control conditions, this brings the total amount of generated images to 1,501,000.

For the class-unconditioned case (i.e., using stable diffusion), this computational load was prohibitive, given our current equipment. The generation of each image using an A5000 GPU takes on average 3.15 seconds, which corresponds to 2 full weeks of computation for all 1,501,000 images using the four A5000 GPUs available in our HPC system (assuming that no other users are using these GPUs, which is in reality a lower estimate since other users use these GPUs too). Thus, we do not have the compute power to undertake this analysis in reasonable amounts of time.

We therefore instead used a class-conditioned diffusion model with a lighter computational load. We performed the generative univariate RNC analysis using a class conditioned diffusion model (cd_imagenet64_l2) trained on the 1,000 image classes from the ImageNet 2012 challenge. This choice was motivated by the fact that cd_imagenet64_l2 offers a good balance between diversity of the generated images (i.e., 1,000 diverse image classes), and compute time. While this diffusion model is trained on the same ImageNet images as the GAN we used in our original analysis, the images generated by the diffusion model are more realistic (as can be seen by the example images presented below).

We implemented six independent evolutions of the generative univariate RNC analysis using this diffusion model, where the images of each evolution were conditioned on one of six classes balanced between animate and inanimate: goldfish, egret, ladybug, dirigible, container ship, golf ball. The results show that the generated images successfully controlled the in silico fMRI responses for the alignment conditions (images driving or suppressing both V1 and V4), but not for the disentanglement conditions (images driving V1 while suppressing V4, or vice versa) (**Rebuttal Figure 1**, see below). Furthermore, the univariate responses for the generated images reached the univariate response threshold only for the control conditions suppressing both V1 and V4, and as a consequence the image-complexity reduction objective did not activate for the other three control conditions. We believe this is because enforcing

more realistic images effectively reduces the space of possible solutions, making it harder to generate the image solutions required to optimally control two areas. Specifically, since the image class has to be decided a priori (i.e., because the genetic algorithm is designed to optimize continuous variables such as the the image latents, and not categorical variables such as image classes), the generated images will be constrained to the visual features typically contained in images from each class, which might not be the features required to optimally control the fMRI responses. We might obtain significant results by conditioning the diffusion model on other classes, but we could not perform a comprehensive search due to the computational costs of this analysis: each evolution takes 45 hour to run on GPU; independent evolutions are required for each of the four control conditions; and the HPC of our university has only 8 GPUs with enough RAM to perform this analysis (which are shared among other users).

Because the success of the diffusion-based procedure was only partial, we decided to not include this analysis in the paper. Despite this, the generated controlling images were qualitatively similar to the controlling images from our original paper analyses (e.g., images driving both areas consist of cluttered scenes with many objects or object-like shapes; images suppressing both areas are mostly empty; images suppressing V1 while driving V4 consist of single objects on uniform backgrounds; images driving V1 while suppressing V4 consist of vegetation or ocean waves with high spatial frequencies). This corroborates our previous findings that high spatial frequencies and object-like shapes are uniquely represented in V1 and V4.

In sum, while the generative univariate RNC analysis using diffusion models did not successfully control the in silico fMRI responses for all control conditions, the resulting images nevertheless contained the same visual features obtained when applying univariate RNC on the in silico fMRI responses for NSD/ImageNet/THINGS images, or for images generated with GANs (i.e., high spatial frequencies and object-like shapes), which strengthens the generalizability of the findings.

Please note that we made alternative efforts to further strengthen the generalizability of our findings. We replicated both our quantitative and qualitative results when applying RNC on the in silico fMRI responses generated from encoding models trained on a dataset different from NSD (for more details, see response to point 1 of Reviewer #3); and when running 10 independent evolutions in the GAN-based generative univariate RNC analysis (for more details, see response to point 5 of Reviewer #1). Finally, the successful validation on in vivo responses demonstrates that high spatial frequencies and object-like shapes are uniquely represented in V1 and V4.

Rebuttal Figure 1 | Results of generative univariate RNC using a class conditioned diffusion model. For each neural control condition, the plots show the in silico univariate fMRI responses (represented by colored lines) and the PNG compression file size (represented by black lines) for the best generated image of each genetic algorithm generation, averaged across the evolutions for the six image classes (goldfish, egret, ladybug, dirigible, container ship, golf ball). The vertical dashed lines indicate the generation where the univariate response threshold is reached (the threshold is never reached except for the control condition suppressing both V1 and V4). On top of each plot are the optimized images from each of the six image classes.

4. Multivariate analysis interpretation

While multivariate fMRI responses should theoretically reveal more detailed representational differences between areas than univariate responses, the images that disentangle RSMs for V1 and V4 did not reveal common visual patterns (lines 326-328). The paper should clarify whether this conclusion is based on qualitative image inspection and discuss whether this suggests limitations of the RNC method or reflects genuine properties of neural representations.

Following the Reviewer's suggestions, we clarified that the observations made on the aligning and disentangling RSMs and controlling images were based on visual inspection (**Results, page 14, lines 326-334**):

*"To understand the effect of image properties on the multivariate RNC scores, we **visually** inspected the V1 and V4 RSMs in conjunction with the controlling images (Fig. 5b). For both areas, RSM entries comparing different images including the sky in their upper half indicated highly positive correlations, while RSM entries comparing images with and without the sky in the upper half indicated highly negative correlations (Fig*

5c; Supplementary Figure 16a). This similar combination of highly positive and negative correlation RSM entries led to a high RSA correlation score for V1 and V4 and thus to alignment. On the other hand, *upon visual inspection* the V1 and V4 RSMs for the disentangling images contained correlation scores of lower absolute magnitude and did not reveal common visual patterns (**Fig. 5b**).

Since the controlling images found by RNC are successful in quantitatively disentangling the multivariate fMRI responses of V1 and V4 (both in silico and in vivo), RNC successfully performed the task it was designed to perform. Therefore, we do not think that the lack of common visual patterns (upon visual inspections) in the multivariate RNC disentangling images is a limitation of the approach, and instead directly reflects a property of neural representations, namely, the extent to which several stimulus features dominate the fMRI response magnitudes. We elaborated on this in the Discussion (**Discussion, page 24, lines 607-621**):

“When one or few of the multiple stimulus features dominate the fMRI response magnitude, RNC primarily exploits these few features to align or disentangle the fMRI responses of two areas, resulting in images with few salient controlling visual features that are easier to interpret. As an example, the V1 vs. V4 controlling images from the multivariate RNC alignment condition contained easily interpretable visual patterns, which indicated that the shared representational content of these two areas’ multivariate fMRI responses is dominated by image topological properties. When no stimulus feature dominates the fMRI response magnitude, RNC instead exploits multiple features, resulting in images with multiple controlling visual features that complicate interpretation. As an example, the multivariate RNC disentangling images for the V1 vs. V4 comparison did not reveal common patterns upon visual inspection, which suggests that the unique representational content of these two areas’ multivariate fMRI responses is similarly determined by multiple non-dominating visual features. Determining the non-dominating visual features might be possible through RNC variants that isolate each of them, for example by using parameterized artificial stimuli for targeted hypothesis testing.”

5. Context within existing literature

To better highlight RNC's strengths and novelty, the discussion should relate the findings (high spatial frequencies and object-like shapes driving unique V1-V4 univariate responses, shared multivariate representations arising from common retinotopic properties) to existing literature studying V1 and V4 in isolation. This would help readers understand how RNC advances our understanding beyond traditional approaches.

We elaborated on RNC's strength and novelty with respect to traditional approaches studying areas in isolation in the Discussion section (the additions also reflect new analyses and findings from the network-level visualization suggested by the Reviewer in the following review point - review point 6) (**Discussion, page 22, lines 532-553**):

“Through RNC, we successfully controlled univariate and multivariate in silico fMRI responses jointly for areas across the visual cortical network. This confirmed representational properties known from investigating these areas in isolation, for example that V1 is tuned to high spatial frequencies (Foster et al., 1985), that V4 is

tuned to object-like shape (Kobatake & Tanaka, 1994), and that both areas are tuned to topological image properties (Wandell & Winawer, 2011). Jointly controlling multiple areas additionally revealed network-level properties of how these areas represent the visual world that are not available from investigating them in isolation. First, that unique representational content of early- and mid- level visual areas increases as a function of cortical distance. This representational pattern likely reflects the decrease in anatomical connectivity with increasing distance between visual areas (Felleman & Van Essen, 1991), as well as other gradual changes along the visual hierarchy such as increasing receptive field sizes (Dumoulin & Wandell, 2008) and increasingly complex functional specialization (Grill-Spector & Malach, 2004). Second, that shared representational content is higher and unique representational content lower for high-level visual areas with similar categorical selectivity. The successful disentanglement of high-level visual areas with similar categorical selectivity (i.e., PPA and RSC (Park & Chun, 2009)) demonstrates that RNC is sensitive to fine-grained representational differences. Furthermore, the successful control of RSC—an area that, beyond visual navigation, is also involved in memory and planning (Vann et al., 2009)—suggests that RNC could also reveal representational relationships properties of more anterior areas that contribute to visual processing, such as ventrolateral prefrontal cortex (Rose & Ponce, 2024). Third, that early- and mid-level visual areas are more similar to each other than they are to high-level visual areas in terms of representational content. Finally, that representational relationships between visual areas adaptively vary around a typical configuration defined by the previous three properties.”

6. Network-level visualization

While results are presented for each pair of visual areas, a network-level visualization would better support the paper's claims about "how visual areas jointly represent the world as an interconnected network" (abstract) and how "RNC extends existing anatomical and functional connectivity research assessing the brain as a complex interconnected network with the concept of representation" (discussion lines 398-400). Such visualization could help readers grasp the holistic nature of the findings.

We really thank the Reviewer for this suggestion, which we developed into a new Results section where we provide network-level visualization of representational relationships across the visual cortical network. Following we present the text from this new Results section (**Results, page 18, lines 424-478**):

“Representational relationships between visual areas adaptively vary around a typical network-level configuration

Fig. 7 | Representational relationships between visual areas adaptively vary around a typical network-level configuration. **a**, Multidimensional scaling (MDS) embeddings of the *in silico* univariate fMRI responses of early-, mid-, and high-level visual areas for all 73,000 NSD images, indicating the typical representational relationship configuration of the visual cortical network. **b**, MDS embeddings of the *in silico* multivariate fMRI responses for all 73,000 NSD images, indicating the typical representational relationship configuration of the visual cortical network. **c**, MDS embeddings of the *in silico* multivariate fMRI responses for the controlling images from two multivariate RNC control conditions: images aligning EBA and RSC (purple condition), or disentangling PPA and RSC (orange condition). **d**, MDS embeddings of the *in silico* univariate fMRI responses for the controlling images from four univariate RNC control conditions: images driving both FFA and RSC (green condition), suppressing both V1 and PPA (blue condition), driving EBA while suppressing FFA (yellow condition), or suppressing V1 while driving V2 (red condition). The opacity of the lines connecting each area reflects the proximity of these areas in two-dimensional embedding space (more opaque lines indicate higher proximity). A higher proximity between two areas indicates a stronger resemblance of their representational content. Six representative images are shown for each multidimensional scaling analysis.

Vision is enabled by a complex interconnected network of cortical areas that jointly represent visual information. Thus, we next moved to the network-level visualization of the representational relationships discovered for individual pairwise comparisons of areas.

We first asked what is the typical representational relationship configuration of areas within the visual cortical network. Using multidimensional scaling (MDS) (Hout et al., 2012), we reduced the dimensionality of the subject-average in silico univariate or multivariate fMRI responses for all 73,000 NSD images of early-, mid-, and high-level visual areas (V1, V2, V3, V4, EBA, FFA, PPA, RSC). This resulted in two-dimensional embeddings where a higher proximity between two areas reflects a stronger resemblance of their representational content. For both univariate (Fig. 7a) and multivariate (Fig. 7b) in silico fMRI responses, these embeddings revealed three network-level properties that together defined a common, typical network-level configuration. First, that early- and mid-level visual areas' proximity in embedding space mirror their cortical distance, further supporting that unique representational content increases as a function of cortical distance. Second, that high-level visual areas cluster based on categorical selectivity for animate objects (EBA and FFA) and scenes (PPA and RSC), further supporting that shared representational content is higher and unique representational content lower for areas with similar categorical selectivity. Third, that early- and mid-level visual areas are closer to each other than they are to high-level visual areas, indicating an analogous relationship for their representational content.

*Visual stimulation continuously alters the representational content of visual areas, leading to reconfigurations of these areas' representational relationships. Are these reconfigurations rigidly preserving the typical representational relationship configuration properties revealed above, or is the visual cortical network flexibly spanning any configuration? To assess this, we applied MDS on the in silico fMRI responses for the aligning or disentangling images selected through RNC. The controlling images led to representational relationship configurations that negated one, two, but not all three properties, indicating that representational relationships adaptively vary around their typical configuration (Fig. 7c-d). As an illustrative example, the controlling images suppressing V1 while driving V2's univariate responses moved V1 closer to V3 than to V2 thus negating the first property, and moved V1 closer to EBA/FFA than to the other early- and mid-level visual areas thus negating the third property (Fig. 7d, red condition; the representational relationship configurations for other RNC's pairwise area comparisons and control conditions are shown in **Supplementary Figures 22-25**).*

Together, these results provide a unified picture of how visual areas jointly represent the world as an interconnected network. We showed that representational relationships between visual areas adaptively vary around a typical network-level configuration, and that RNC enables the exploration of the state space of possible network configurations."

Remarks on code availability

The code for reproducing the paper's results has been organized according to the order of main results presented in the paper. The authors also provide the tutorials for the RNC method that can be run with Jupyter Notebook/Google Colab via their GitHub repository.

We thank the Reviewer for checking the code.

Reviewer #3

Remarks to the Author

The paper describes RNC, an encoding model for generating fMRI responses to natural images that is used to investigate the relationships between neural responses in the early and intermediate visual areas. The model investigates a novel angle to approach the problem of understanding neural representations in visual processing. The paper itself is theoretically sound and the authors have done their due diligence to ensure their work stands on a rigorous basis. Additionally, I would like to commend the authors on a very clear explanation of their process especially with illustrations in every figure making the paper very clear. That being said, there are multiple areas of improvement that would make the paper's contribution more valuable to the scientific community.

We thank Reviewer #3 for the positive evaluation and helpful comments that resulted in valuable additional analyses and discussions. In brief, we:

(1) Carried out additional tests (i.e., using additional image sets and the Visual Illusion Reconstruction dataset) that replicated our original results, thus strengthening the robustness of RNC and the generalization of its solutions.

(2) Tested the out-of-distribution generalization performance of the encoding models used to generate the in silico responses, obtaining prediction accuracies well above chance, which suggest the generalizability of RNC's solutions across very diverse visual distributions.

(3) Further motivated the choice of the encoding models used, including a visualization of the model's architecture.

(4) Discussed why, for univariate RNC, alignment does not decrease with cortical distance.

(5) Discussed a possible reason why the images controlling multivariate responses are less interpretable than the ones controlling univariate responses.

(6) Discussed the possible role of feedback processing on alignment and disentanglement of areas.

(7) Improved the clarity of text and figures across the manuscript.

Major points:

1. The biggest issue in this work is that the use of NSD as a dataset for the model might not be the best choice for generalization, which the authors have acknowledged in the limitations. While it offers a deep stimulus set, it fails in the avenue of stimulus diversity, which makes the image set limited to a certain distribution of images. This problem has plagued many of the papers recently and was discussed in detail in the following preprint: <https://arxiv.org/abs/2405.10078>. Alternate datasets that offer more diversity would include the deep image reconstruction dataset which includes both natural and artificial image dataset (reference: <https://doi.org/10.1371/journal.pcbi.1006633>).

We thank the Reviewer for this suggestion, which led us to carry out additional tests that strengthened the robustness of RNC and the generalization of its solutions. Specifically, we applied RNC on in silico fMRI responses for different image sets (the 50,000 images from the ImageNet 2012 challenge validation split, and the 26,107 images from THINGS), and also on in silico fMRI responses from encoding models trained on images and fMRI responses from a dataset different from NDS. Following the Reviewer's suggestion, we trained these encoding models on the Visual Illusion Reconstruction dataset (Chen et al., 2023). Crucially, these tests confirmed both quantitative and qualitative results that we previously obtained when applying RNC on the in silico fMRI responses for the 73,000 NSD images, generated from encoding models trained on NSD. Together, this demonstrates RNC's robustness and generalizability (i.e., across a variety of model training regimes and hypothesis spaces), and indicates that its solutions truly reflect properties of the brain (which we also confirmed through in vivo validation). Following is the updated text throughout the Results section reflecting these additional tests, along with the results figures of tests involving the VIR dataset.

(Results, page 7, lines 167-179): *“To ascertain that the demonstrated representational relationships reflect properties of visual processing, rather than biases of specific datasets, we performed two tests. First, to ensure that RNC's solutions are not biased by the visual distribution from which the controlling images are selected, we applied univariate RNC on the in silico fMRI responses for the 50,000 images from the ImageNet 2012 challenge validation split (Russakovsky et al., 2014), and for the 26,107 images from THINGS (Hebart et al., 2019) (i.e., single objects presented centrally on natural backgrounds, as opposed to the NSD's complex natural scenes consisting of several or no objects appearing at different locations). Second, to ensure that RNC's solutions are not biased by the visual distribution of the encoding models' training data, we applied univariate RNC on the in silico fMRI responses generated from encoding models trained on the Visual Illusion Reconstruction dataset (Chen et al., 2023) (i.e., fMRI responses for images of diverse objects, natural scenes, and materials). Both tests replicated our previous findings (**Supplementary Figures 5-7**), demonstrating RNCs robustness and generalizability, and indicating that its solutions truly reflect properties of the brain.”*

(Results, page 9, lines 210-212): *“When using alternative distributions of images and of encoding model training data, the resulting controlling images also consisted of combinations of high spatial frequencies and objects (**Supplementary Figures 5-7**).”*

“Supplementary Figure 7 | Results of univariate RNC applied on the in silico fMRI responses for the 73,000 NSD images, generated through encoding models trained on the Visual Illusion Reconstruction dataset. a, Univariate RNC quantitative results (univariate response magnitudes), embedded in a four-by-four matrix. b, Stepwise distance between areas. c, Absolute difference between controlling and baseline image univariate responses, averaged across all pairwise comparisons of areas with same stepwise distance. d, Correlation between the univariate responses of two areas, averaged across pairwise comparisons of areas with same stepwise distance. e, Univariate RNC controlling and baseline images for the V1 vs. V4 comparison.”

(Results, page 12, lines 295-297): “We verified the generalizability of these representational relationships, observing quantitatively similar results when using alternative distributions of images and of encoding model training data (Supplementary Figures 13-15).”

(Results, page 13, lines 322-325): “The aligning images often contained uniform portions (i.e., the sky on their upper half), whereas the disentangling images did not, and the baseline images did but to a lesser extent (Fig. 5a). This was also the case when using alternative distributions of images and of encoding model training data (Supplementary Figures 13-15).”

“Supplementary Figure 15 | Results of multivariate RNC applied on the in silico fMRI responses for the 73,000 NSD images, generated through encoding models trained on the Visual Illusion Reconstruction dataset. a, Multivariate RNC quantitative results (RSA scores). b, Stepwise distance between areas. c, Multivariate RNC RSA scores, averaged across pairwise comparisons of areas with same stepwise distance. d, Controlling and baseline images for the V1 vs. V4 comparison.”

Finally, using encoding models trained on the Visual Illusion Reconstruction dataset, we also obtained quantitatively and qualitatively similar results when applying univariate and multivariate RNC on the in silico fMRI responses for the 50,000 ImageNet 2012 challenge validation split images and for the 26,107 THINGS images. Please find the results for these analyses reproduced below (**Rebuttal Figures 2-5**). However, given the confirmatory nature of the results, and the already extensive supplementary material, we decided not to include these additional results in the manuscript.

Rebuttal Figure 2 | Results of univariate RNC applied on the in silico fMRI responses for the 50,000 ImageNet images, generated through encoding models trained on the Visual Illusion Reconstruction dataset. a, Univariate RNC quantitative results (univariate response magnitudes), embedded in a four-by-four matrix. **b**, Stepwise distance between areas. **c**, Absolute difference between controlling and baseline image univariate responses, averaged across all pairwise comparisons of areas with same stepwise distance. **d**, Correlation between the univariate responses of two areas, averaged across pairwise comparisons of areas with same stepwise distance. **e**, Univariate RNC controlling and baseline images for the V1 vs. V4 comparison.

Rebuttal Figure 3 | Results of univariate RNC applied on the in silico fMRI responses for the 26,107 THINGS images, generated through encoding models trained on the Visual Illusion Reconstruction dataset. **a**, Univariate RNC quantitative results (univariate response magnitudes), embedded in a four-by-four matrix. **b**, Stepwise distance between areas. **c**, Absolute difference between controlling and baseline image univariate responses, averaged across all pairwise comparisons of areas with same stepwise distance. **d**, Correlation between the univariate responses of two areas, averaged across pairwise comparisons of areas with same stepwise distance. **e**, Univariate RNC controlling and baseline images for the V1 vs. V4 comparison.

Rebuttal Figure 4 | Results of multivariate RNC applied on the in silico fMRI responses for the 50,000 ImageNet images, generated through encoding models trained on the Visual Illusion Reconstruction dataset. **a**, Multivariate RNC quantitative results (RSA scores). **b**, Stepwise distance between areas. **c**, Multivariate RNC RSA scores, averaged across pairwise comparisons of areas with same stepwise distance. **d**, Controlling and baseline images for the V1 vs. V4 comparison.

Rebuttal Figure 5 | Results of multivariate RNC applied on the in silico fMRI responses for the 26,107 THINGS images, generated through encoding models trained on the Visual Illusion Reconstruction dataset. **a**, Multivariate RNC quantitative results (RSA scores). **b**, Stepwise distance between areas. **c**, Multivariate RNC RSA scores, averaged across pairwise comparisons of areas with same stepwise distance. **d**, Controlling and baseline images for the V1 vs. V4 comparison.

2. One additional limitation is, given the model is feedforward, it might not work very well on stimuli that require top-down modulation. This can be tested on blurred images (ref: <https://doi.org/10.1523/ENEURO.0443-17.2018>) and illusory images (ref: <https://doi.org/10.1126/sciadv.adj3906>). This is an extensive list that might not all fit within the context of this work so you do not need to do all of them but additional testing of limitations would make the paper much richer at least in the in-silico model.

We agree with the Reviewer that one limitation of our work is that the encoding model's predictions might not generalize well outside of their training distribution (e.g., for images that require top-down modulation such as illusory images), and as a consequence that RNC's results using these models might not be valid.

To address this, we additionally included the out-of-distribution (OOD) generalisation scores of the encoding models, reflecting their reliability in predicting in silico fMRI responses for images from outside the training distribution, thus providing an additional and stronger measure of model robustness. For the encoding models trained on NSD we performed OOD tests using NSD-synthetic (Gifford et al., 2025), the companion dataset of NSD consisting of an additional scan session from the same eight subjects of NSD during which fMRI responses were measured to 284 carefully controlled

synthetic (non-naturalistic) stimuli. For the encoding models trained on the Visual Illusion Reconstruction dataset, as suggested by the Reviewer we performed OOD tests using the 38 illusory stimuli included in the dataset (Chen et al., 2023). Since to reduce the effect of noise on results we applied RNC on the in silico fMRI responses for voxels with noise ceiling signal-to-noise ratio (i.e., a measure of a stimulus-related signal in the fMRI responses) above 0.5, to better reflect these results we also changed the encoding accuracies to reflect voxels with noise ceiling signal-to-noise ratio above 0.5. As expected, in both cases we found that the encoding models resulted in lower OOD generalizations scores compared to their in-distribution scores. However, the OOD generalization scores were still well above chance, indicating that the trained encoding models are robust and well generalize both within and outside their training distribution.

Furthermore, to test the validity of RNC's results—namely that these results reflect properties of visual processing in the brain, rather than biases of the encoding model's training data distribution—we showed that RNC's solutions are replicated when using a different set of encoding models trained on the Visual Illusion Reconstruction dataset (for details on this, see the response to the previous review point—review point 1).

To reflect these new considerations and results, we added the following text in the Results and Discussion sections, as well as **Supplementary Figure 3**.

(Results, page 3, lines 75-85): *“The trained encoding models accurately predicted fMRI responses not used for training, resulting in a subject-average noise-ceiling-normalized explained variance score of 78.14% for V1, 72.54% for V2, 65.07% for V3, and 53.29% (Supplementary Figure 2b, Supplementary Figure 3). We further tested the robustness of the encoding models on NSD-synthetic (Gifford et al. 2025), NSD's out-of-distribution companion dataset of fMRI responses to artificial images, obtaining a subject-average noise-ceiling-normalized explained variance score of 60.72% for V1, 52.22% for V2, 46.75% for V3, and 38.89% for V4 (Supplementary Figure 3). These results indicate that the trained encoding models generate reliable in silico fMRI responses, including for images very different from the ones on which the models were trained, therefore providing a solid foundation for in silico experiments.”*

(Discussion, page 23, lines 584-593): *“The key limitation of RNC lies in the component that empowers it: the encoding models generating the in silico neural responses do not predict all explainable neural signal, and their predictions generalize imperfectly beyond the distribution of the visual data they were trained on. Thus, to ascertain the validity of our findings we showed that our encoding models achieved high prediction accuracies when tested both in- and out-of-distribution, and also that RNC's solutions are replicated when using alternative distributions of images and of encoding model training data. Furthermore, the current push in the development of more accurate and robust visual encoding models (Schrimpf et al., 2018; Willeke et al., 2022; Gifford et al., 2023; Gifford et al., 2025a) using large in vivo data sets (Allen et al., 2022; Gifford et al., 2022; Chen et al., 2023; Hebart et al., 2023; Lahner et al., 2024) that also include out-of-distribution datasets (Gifford et al., 2025b; Chen et al., 2023; Gifford et al., 2025a) promises increasingly accurate in silico neural responses,*

in turn increasing the reliability of findings from experimentation on computer-generated brain data.”

“Supplementary Figure 3 | Encoding models in-distribution and out-of-distribution noise-ceiling-normalized explained variance. a, Noise-ceiling-normalized explained variance for encoding models trained on NSD. The in-distribution generalization scores reflect tests on a set of 515 images and fMRI responses not used for training. The out-of-distribution generalization scores reflect tests on the 284 NSD-synthetic (Gifford et al., 2025) images and fMRI responses. b, Noise-ceiling-normalized explained variance for encoding models trained on the Visual Illusion Reconstruction Dataset (Chen et al. 2023). The in-distribution generalization scores reflect tests on a set of 350 images and fMRI responses not used for training. The out-of-distribution generalization scores reflect tests on the 38 visual illusion images from the same dataset.”

Together, these results strengthen the robustness and generalizability of RNC’s encoding models and results, including for stimuli outside of the encoding model’s training distributions. Our results also showed that our encoding models perform better in-distribution than they do out-of-distribution. However, we do not think that it is

practically easy to isolate what are the model characteristics leading to this lower out-of-distribution generalization performance. For example, it is not easy to test whether the feedforward architecture of our models leads to decreases in generalization accuracy for images that require top-down modulation, and here is why.

In a mental experiment, let us test the prediction accuracy of feedforward encoding models on stimuli that require top-down modulation (e.g., the illusory or blurred images suggested by the Reviewer). Let us further imagine that these tests result in lower prediction accuracies compared to testing the encoding models on images from the same visual distribution on which they were trained (i.e., naturalistic images that are not illusory nor blurred). Now, would this lower prediction accuracy be due to the stimuli requiring top-down modulation? Not necessarily. Top-down modulation is only one of the properties by which the blurred and illusory images differ from the naturalistic images on which the encoding models were trained. For example, given that the illusory images are highly artificial, they would be far away from the model training visual distribution of natural images even if they did not contain visual illusions inducing top-down modulation. And because of this, we would expect encoding models to result in lower prediction accuracy regardless of whether the artificial images include, or do not include, illusions inducing top-down modulation. In conclusion, we could not be sure of whether the decreases in the encoding models' prediction accuracies are due to top-down modulations, or to other stimulus-induced visual processing mechanisms.

Furthermore, let us now imagine that we test the prediction accuracy of feedforward encoding models for a set of test stimuli that differ from the model training stimuli *only* in terms of top-down modulation (i.e., everything else being equal). Again, let us imagine that these tests result in lower prediction accuracies compared to testing the encoding models on images from the same visual distribution on which they were trained (i.e., naturalistic images that are not illusory nor blurred). This would indicate that the encoding models are failing in explaining top-down modulations. However, would this lower prediction accuracy for stimuli that require top-down modulation be due to the encoding models being feedforward? Not necessarily. The feedforward nature of the models is only one of their many model properties (others being model depth, type of non-linearities, learning algorithm, etc.), all of which could have contributed to this lower prediction accuracy. In conclusion, we could not be sure of whether the decreases in the encoding models' prediction accuracies are due to the feedforward nature of the models, or to other model properties.

Put briefly, if it is straightforward to measure *that* a model leads to decreases in performance, it is often very hard to understand *why* this is the case.

3. Given models are independent across ROIs, it is expected that deeper areas end up having lower explained variance as model complexity remains the same but processing depth of voxels coming from V4 is larger than those in V1. Is there a reasoning for the choice of model depth? Previous studies have shown that that voxels coming from deeper regions within the visual processing hierarchy were more similar to deeper layer representations in DNN (refs: <https://doi.org/10.1523/JNEUROSCI.5023-14.2015> ; <https://www.nature.com/articles/ncomms15037>).

We agree that model depth is an important factor when predicting visual areas across the hierarchy. This is why we used a training procedure that flexibly chooses model features across depth. Specifically, we used as encoding models the feature-weighted receptive field (fwRF) proposed in (St-Yves & Naselaris, 2018) which, during training, empirically learns each layer's contribution to each area. As a consequence, the optimal layer contribution does not rely on potentially suboptimal assumptions of the experimenter (since the optimal layer contribution likely changes for each model, area, and training data), but instead is found during training based on the training data. We edited the Methods text to reflect these considerations (**Methods, page 25, lines 662-679**):

*“Each encoding model predicted fMRI responses for multiple voxels (i.e., all voxels of a specific subject and area), and consisted of two components: a feature extractor shared across all voxels, and one projection head for each voxel (**Supplementary Figure 1a**).*

*The **feature extractor** is a multi-layer feedforward convolutional neural network called GNet (Allen et al. 2022) (**Supplementary Figure 1b**). Giving an image as input to GNet activates its layers, resulting in multiple features maps (i.e., GNet's representations of this image) (**Supplementary Figure 1a**). The **feature extractor's** weights are fully learned during model training, based on the joint loss of all voxels.*

*The projection head of each voxel is a **feature-weighted receptive field (fwRF)** that combines a spatial pooling field and feature weights (St-Yves & Naselaris, 2018). The spatial pooling field determines the region of visual space (i.e., the GNet feature space which, due to the convolutional operations, preserves the topology of visual space of the input images) that drives voxel activity. After GNet's feature maps are spatially pooled, they are linearly combined by the feature weights, resulting in the voxel response prediction (**Supplementary Figure 1a**). Both the spatial pooling field and feature weights are learned during model training, independently for each voxel. This allows for the empirical determination of the optimal contribution of each model feature to each voxel based on the training data—for example, by learning the optimal hierarchical correspondence configuration between model layers and visual cortical areas (Güçlü & van Gerven, 2015; Horikawa & Kamitani, 2017).”*

4. The alignment of align image group does not appear to decrease with cortical processing distance whereas disentanglement increases (as shown in figure 2d and 4c). Given how the align images with both V1 increase and V4 increase show to have elements from both the disentangle images, I think it is important to discuss that point and give a reasoning on why it alignment does not decrease.

We also believe that the reason why alignment does not decrease with cortical area distance (**Fig. 2d**, green dots) is that the controlling images aligning two areas consist of the logical combination of the visual features disentangling them, rather than of features for which the two areas are not disentangled. Following the Reviewer's suggestion, we elaborated on this in both the Results and Discussion sections.

(Results, page 6, lines 149-166): “Visual areas V1 to V4 form a processing hierarchy in terms of anatomical connectivity, response latency, and the complexity of stimulus properties maximally driving neural responses. This suggests that disentanglement should increase *and that alignment should decrease* with increasing node distance across this hierarchy. We confirmed this prediction. As the stepwise distance between two areas increased (**Fig. 2c**), the absolute difference between the *in silico* univariate fMRI responses in the disentangling control condition and the baseline increased (**Fig. 2d, yellow and red dots**). Furthermore, the absolute distance between the univariate responses in the alignment control condition suppressing both areas (but not driving them) and the baseline decreased (**Fig. 2d, blue dots**). This indicates that the univariate responses of areas further away from each other were *less aligned and more strongly disentangled*. Strengthening this finding, as the stepwise distance between two areas increased, the correlation between their univariate responses decreased (**Fig. 2e**).”

(Discussion, page 23, lines 594-606): “The assumption of aligned or disentangled neural responses indicating shared or unique representational content is not a given. *For example, for early- and mid-level visual areas we found that the assumption was correct for all multivariate RNC control conditions, for the univariate RNC disentanglement conditions, but not for the univariate RNC alignment conditions. There, the controlling images aligning both V1 and V4 univariate fMRI responses consisted of the logical combination of the visual features disentangling them (i.e., high spatial frequency and objects), rather than of features for which the two areas are not disentangled, as would be the case if the assumption held. This might be the reason why the absolute distance between the univariate responses in the alignment control condition driving both areas and the baseline did not decrease with cortical distance (Fig. 2d, green dots).* Together, this highlights a way to rigorously test the assumptions of RNC. Furthermore, finding the assumption to be negated is in itself scientifically interesting, as it reveals how either shared or unique representational content can lead to aligned univariate responses.”

5. Figure 4d is hard to understand. The authors introduce the concepts of observed vs. target RSA scores for the first time here without proper explanation making the reader puzzled as to why the difference in r values decreases with area distance. It looks to me on first sight that multivariate patterns align more with cortical distance based on this figure which negates figure 4c.

We agree with the Reviewer that the concepts of observed vs. target RSA scores were confusing. Furthermore, they did not bring substantially new information that is not available from the other figure panels. Because of these reasons we decided to remove this figure panel, and instead visualize the change of both alignment and disentanglement as a function of cortical distance in one figure panel (**Figure 4d**). Following are the new text and figure reflecting these changes (**Results, page 12, lines 287-294**):

“Here too we tested whether *alignment* of multivariate responses *decreases, and disentanglement increases*, with increasing node distance across the visual processing hierarchy (**Fig. 4c**). The RSA scores for the *aligning, disentangling, and baseline*

images decreased as the stepwise distance between two areas increased (within-subject significance, permutation test, $p < 0.05$, Benjamini/Hochberg corrected over 3 tests; between-subject significance, frequentist population prevalence test, $p < 10^{-5}$, indicating within-subject significance in at least 5/8 subjects), indicating that the multivariate responses of areas further away from each other were *less aligned and more strongly disentangled (Fig. 4d).*”

“Fig. 4 | RNC controls in silico multivariate fMRI responses across early- and mid-level visual cortical areas. a, Multivariate RNC neural control conditions. b, Multivariate RNC results, consisting of RSA scores (Pearson’s r) for each pairwise comparison of areas. Asterisks indicate neural control conditions for which the RSA scores from the controlling images are significantly higher (alignment) or lower (disentanglement) than baseline (within-subject significance, permutation test, $p < 0.05$, Benjamini/Hochberg corrected over 2 tests for each pairwise comparison of areas; between-subject significance, frequentist population prevalence test, $p < 10^{-9}$, indicating within-subject significance in at least 7/8 subjects). c, Stepwise distances between areas. d, Multivariate RNC RSA scores, averaged across pairwise comparisons of areas with same stepwise distance. Connectors between area distances indicate a significant decreasing trend (within-subject significance, permutation test, $p < 0.05$, Benjamini/Hochberg corrected over 3 tests; between-subject significance, frequentist population prevalence test, $p < 10^{-5}$, indicating within-subject significance in at least 5/8 subjects). Opaque and transparent dots represent subject-average and single subject results, respectively. Error bars reflect 95% confidence intervals.”

6. From looking at the multivariate align and disentangle analysis, I can see why there would not be a certain pattern since the algorithm can choose a set for visually similar images vs. a random set. Perhaps it is not the best way to conduct this analysis and it requires more constraints on the set of images put together in a set rather than letting the algorithm select randomly. One possible way is to make sets of images that are visually or contextually similar and testing them for the most align and disentangle.

We think that the lack of interpretable controlling visual features in the multivariate RNC disentanglement condition is due to RNC exploiting multiple visual features for disentanglement, where none of these features dominates the fMRI response magnitude (for details, also see response to review point 4 of Reviewer #2) (**Discussion, page 24, lines 607-621**):

“When one or few of the multiple stimulus features dominate the fMRI response magnitude, RNC primarily exploits these few features to align or disentangle the fMRI responses of two areas, resulting in images with few salient controlling visual features that are easier to interpret. As an example, the V1 vs. V4 controlling images from the multivariate RNC alignment condition contained easily interpretable visual patterns, which indicated that the shared representational content of these two areas’ multivariate fMRI responses is dominated by image topological properties. When no stimulus feature dominates the fMRI response magnitude, RNC instead exploits multiple features, resulting in images with multiple controlling visual features that complicate interpretation. As an example, the multivariate RNC disentangling images for the V1 vs. V4 comparison did not reveal common patterns upon visual inspection, which suggests that the unique representational content of these two areas’ multivariate fMRI responses is similarly determined by multiple non-dominating visual features. Determining the non-dominating visual features might be possible through RNC variants that isolate each of them, for example by using parameterized artificial stimuli for targeted hypothesis testing.”

Therefore, we think the Reviewer is correct in suggesting that adding constraints on the set of images used by RNC might result in controlling images that well disentangle areas, and that also contain controlling visual features interpretable by visual inspection. We thus applied multivariate RNC on the V1 vs. V4 comparison, using four constrained image spaces. Two constrained image spaces were subsets of the 73,000 NSD images: 9,684 images of food and 5,080 images of indoor scenes. The other two constrained image spaces were subsets of the 50,000 ImageNet 2012 challenge validation split images: 6,500 images of devices, and 500 images of scenes. (We selected these image subspaces since they consisted of conceptually and/or visually more homogeneous images (than the full image space); we selected the images of each subspace using image category labels provided by NSD/ImageNet; we generated the in silico fMRI responses with encoding models trained on NSD.)

Rebuttal Figure 6 | Results of multivariate RNC applied to the in silico fMRI responses for four constrained image spaces. a, Multivariate RNC results for two constrained image spaces consisting of two subsets of the 73,000 NSD images: 9,684 images of food and 5,080 images of indoor scenes. **b**, Multivariate RNC results for two constrained image spaces consisting of two subsets of the 50,000 ImageNet images: 6,500 images of devices, and 500 images of scenes.

For all four constrained image spaces, RNC found controlling images that successfully aligned or disentangled V1 and V4's multivariate fMRI responses. However, here too the disentangling images lacked controlling visual features interpretable by visual inspection (**Rebuttal Figure 6**).

We believe this is because, despite being reduced in visual and conceptual variety, the images from the constrained image spaces are still naturalistic, and thus complex combinations of multiple visual features, many of which contribute to the disentanglement of V1 and V4's multivariate fMRI responses. It is not straightforward how to appropriately constrain the image space so to result in more interpretable controlling images. Appropriately constraining the visual space entails choosing the right restricted set of images, that is, correctly hypothesizing which image features should be retained or excluded. This requires prior knowledge about the representational relationships between areas, which is not available a priori (otherwise there would not be a point in applying RNC in the first place).

Applying RNC on strictly controlled artificial stimuli (i.e., artificial stimuli only varying in one or few visual features) might result in more interpretable controlling visual features. However, this is in conflict with the main reason why we developed RNC: to allow for hypothesis-free exploration of the visual space in search of controlling images. By constraining the visual space with hypotheses of how the solution should look like, we risk missing other, more optimal solutions lying outside our constrained hypothesis space (i.e., image solutions that better disentangle two areas, even if they are too easy to visually interpret).

7. In figure 6d: what do the black circles in the background of the left figure represent?

We clarified what the black circles in the background represent in the caption of figure 6d (**Results, page 20, lines 485-487**):

“Small black points indicate in vivo univariate fMRI responses of all subjects for all V1 vs. V4 univariate RNC controlling images.”

8. The authors should discuss the possible role of temporal processing. While fMRI loses most temporal information, it is possible that the disentanglement seen between V4 and V1 is due to a top-down suppression of signals as a result of object recognition (the aha moment) which was shown in multiple previous studies.

We fully agree with the Reviewer that alignment and disentanglement of neural representational content could be influenced by feedback processing between areas, and that the low temporal resolution of fMRI does not allow us to investigate this. Feedback modulation might be driven by visually challenging stimuli, or stimuli that act as masks of feedback processing. However, we do not think that, within the context of our experiment, feedback activity is specifically modulated by the “aha” moment. To our understanding, the “aha” experience arises specifically when solving problems or when images are re-interpreted in surprising ways (Davidson, 1989). The fMRI responses used in our work did not involve problem solving, and were plain naturalistic images unlikely to trigger re-interpretation (as opposed to e.g. illusions or mooney images). To reflect these observations, we added the following paragraph to the Discussion section (**Discussion, page 24, lines 622-628**):

“Feedforward and feedback connections both shape the representational content of visual areas within the cortical visual hierarchy (Felleman & VanEssen, 1991; Markov et al., 2014; Lamme & Roelfsema, 2000; Gilbert & Li, 2013). Therefore, beyond feedforward processing (DiCarlo et al., 2012; DiCarlo & Cox, 2007), shared and unique representational content might also be influenced by visual stimulation that either promotes (e.g., with challenging stimuli (Groen et al., 2018; Kar et al., 2019)) or suppresses (e.g., with visual masks (Breitmeyer & Ogmen, 2000; Fahrenfort et al., 2007; Maguire & Howe, 2016)) feedback from higher areas to lower areas. Applying RNC on time-resolved M/EEG, ECoG, or electrophysiology data, or on cortical-layer resolved data (Lawrence et al., 2019), is a promising avenue to isolate the respective influence of feedforward and feedback signaling on representational relationships.”

Minor points:

9. I would appreciate having the model architecture in detail in the Supplementary information for reproducibility.

We added the encoding model's architecture in **Supplementary Figure 1**:

“Supplementary Figure 1 | Encoding model architecture (adapted from Allen et al., 2022). a, Illustration of an encoding model that predicts brain activity in a given voxel (r_{tv}) in response to images (x_t). Images are passed to nonlinear feature extractors (i.e., GNet), η_i (trapezoids), that output feature maps (grey cuboids). Feature maps are grouped, passed through an element-wise nonlinearity, $f(\cdot)$, and then multiplied pixel-wise by a spatial pooling field (g^1, \dots, g^N where superscripts index distinct groups of feature maps) that determines the region of visual space that drives voxel activity. The weighted pixel values in each feature map are then summed, reducing each feature map to a scalar value. These scalar values are concatenated across all feature maps, forming a single feature vector that is passed through another element-wise nonlinearity (left black rectangle) and then weighted by a set of feature weights, w (right black rectangle), to yield predicted voxel activity. The feature extractors η_i (i.e., GNet), the spatial pooling fields g^1, \dots, g^N , and the feature weights w are all optimized while training the encoding model to predict brain responses. b, GNet's architecture. GNet is a deep convolutional neural network consisting of convolutional layers (rows labeled 'conv'; values indicate feature depth and convolutional filter resolution; 'str' = filter stride, 'pad' = convolutional padding), max-pooling layers ('maxpool'), batch-normalization and weight-dropout layers ('batchnorm + dropout'). Feature maps in the convolutional layers (indicated by red arrows; resolution of the feature maps in parentheses) are used as predictors of brain activity in the context of an encoding model.”

10. Page 18 line 537: Avoid mentioning that you followed the default values and explicitly mention them as these values could be different in newer versions of the same library.

Following the Reviewer's suggestion, we explicitly mentioned the function parameter values (**Methods, page 26, lines 694-696**):

"The encoding models' weights were optimized on 75 epochs of the training data partition using batch sizes of 128 images and Pytorch's (Paszke et al., 2019) Adam optimizer with a learning rate of 0.001, a weight decay term of 0, betas coefficients of (0.9, 0.999), and an eps term of 1e-08."

11. Page 14 line 357: c written but should be e.

We thank the reviewer for pointing out this typo, which we fixed (**Results, page 20, line 487**).

Remarks on code availability

python version not included. Advisable to create a requirements.txt file with libraries and their versions for a pip install or alternatively a yml file for anaconda environment creation.

We thank the Reviewer for checking the code. Following the Reviewer's advice, we added to the GitHub repository instructions for creating an Anaconda environment with the Python version we used in the project. Furthermore, we added a requirements.txt file including the used libraries and versions, and guide users on how to install these libraries with pip within the Anaconda environment previously created (see screenshot below).

⚙️ Installation

This repository contains code to reproduce all paper's results.

To run the code, you first need to install the libraries in the requirements.txt file within an Anaconda environment. Here, we guide you through the installation steps.

First, create an Anaconda environment with the correct Python version:

```
conda create -n rnc_env python=3.9
```

Next, download the requirements.txt file, navigate with your terminal to the download directory, and activate the Anaconda environment previously created with:

```
source activate rnc_env
```

Now you can install the libraries with:

```
pip install -r requirements.txt
```

Finally, you also need to install the NEST Python package with:

```
pip install -U git+https://github.com/gifale95/NEST.git
```

Dear Editor, Dear Reviewers,

Please find the detailed point-by-point response below. Editor and Reviewer comments are highlighted in gray, and the corresponding responses indented. Within the responses, extracts from the manuscript are “*quoted in italics*”, and extracts reflecting changes to the manuscript are “*quoted in italics and red font*”. Manuscript line/page/figure numbers are indicated in **bold font**. Blue underscored text denotes hyperlinks.

Editor

Thank you for submitting your revised manuscript "In silico discovery of representational relationships across visual cortex" (NATHUMBEHAV-24114428A). It has now been seen by the original referees and their comments are below. As you can see, the reviewers find that the paper has improved in revision. We will therefore be happy in principle to publish it in Nature Human Behaviour, pending minor revisions to satisfy the referees' final requests and to comply with our editorial and formatting guidelines.

We thank the Editor for the interest in publishing the manuscript. In this second revision, we:

- (1) Addressed the two minor revision points from Reviewer #2.
- (2) Adapted the manuscript to Nature Human Behavior's editorial and formatting guidelines.

Reviewer #1

Remarks to the Author

The authors have done a good job of responding to the comments with new analyses and edits to the manuscript. I think the overall manuscript is much improved, and has wider impact.

We thank Reviewer #1 for reading the revised manuscript and for the positive evaluation, as well as for the comments of the first review round that resulted in valuable additional analyses and discussions.

Remarks on code availability

I do not have time to review the code in detail but a cursory review revealed no glaring issues - the code base appears well documented and complete.

We thank Reviewer #1 for once again checking the code.

Reviewer #2

Remarks to the Author

I thank the authors for addressing all my comments in a thorough and thoughtful manner. The manuscript has been significantly improved and the new results from additional analyses add valuable insights to the manuscript. I have only two minor suggestions for further clarification:

We thank Reviewer #2 for reading the revised manuscript and for the positive evaluation, as well as for the comments of the first review round that resulted in valuable additional analyses and discussions.

1. There is an inconsistency in the number of corrected tests (3) mentioned in the figure caption of Figure 4d compared to the methods section. For clarity, it is recommended to specify which tests were corrected. This has been done for some analyses (e.g., lines 1145-1146: "we corrected the p-values over 4 tests (one for each neural control condition).") but not for all.

We thank the Reviewer for pointing out this inconsistency. The correct number of corrected tests is 3, as reported in the caption of **Fig. 4d**. We therefore updated the Methods section accordingly and, based on the Reviewer's suggestion, for all analyses we further specified which tests were corrected (**Methods, page 28, lines 1109-1144**):

"In the encoding models noise analysis tests the null hypothesis was that the noise-ceiling-normalized explained variance scores of the different predictors were equal, we permuted the encoding accuracy scores over fMRI voxels and predictors, and we corrected the p-values over 2 tests for each area (one test comparing single and average NSD trials as predictors, and one test comparing average NSD trials and in silico responses as predictors). In the univariate RNC analysis tests the null hypothesis was that the univariate responses for the controlling and baseline images were equal, we permuted the univariate responses across image conditions, and we corrected the p-values over 8 tests for each pairwise comparison of areas (one test for each combination of 2 areas and 4 neural control conditions). In the univariate RNC cortical distance analysis tests the null hypothesis was that the absolute differences between the baseline univariate responses and the univariate responses for controlling images from different stepwise area distances were equal, we permuted the univariate responses across areas, and we corrected the p-values over 4 tests (one test for each neural control condition). In the multivariate RNC analysis tests the null hypothesis was that the RSA scores for the controlling and baseline images were equal, we permuted the multivariate responses across image conditions, and we corrected the p-values over 2 tests for each pairwise comparison of areas (one test for each neural control condition). In the multivariate RNC cortical distance analysis tests the null hypothesis was that the absolute differences between the baseline RSA scores and the RSA scores for controlling images from different stepwise area distances were equal, we permuted the multivariate responses across areas and image conditions, and we corrected the p-values over 3 tests (one test for the aligning, disentangling,

and baseline images, respectively). In the multivariate RNC retinotopy analysis tests the null hypothesis was that the univariate responses of voxels tuned to the lower and upper portions of the visual field were equal, we permuted the data across ventral and dorsal voxels, and we corrected the p-values over 2 tests for each area (one test for sky and no sky images, respectively). In the univariate RNC categorical selectivity analysis tests the null hypothesis was that the absolute differences between the baseline univariate responses and the univariate responses for controlling images from areas within or between categorical selectivity groups were equal, we permuted the univariate responses across areas, and we corrected the p-values over 4 tests (one test for each neural control condition). In the multivariate RNC categorical selectivity analysis tests the null hypothesis was that the absolute differences between the baseline RSA scores and the RSA scores for controlling images from areas within or between categorical selectivity groups were equal, we permuted the univariate responses across areas and image conditions, and we corrected the p-values over 2 tests (one test for each neural control condition).”

2. Please verify the results presented in the leftmost plot in Fig. 7d (images driving both FFA and RSC, green condition). The depicted proximity between FFA and RSC appears weaker (shown by lower opacity connecting lines) compared to the all-image condition in Fig. 7a, which seems counterintuitive.

We agree that, intuitively, FFA and RSC should fall closer to each other in multidimensional scaling (MDS) space in **Fig. 7d**. However, while the controlling images found through RNC successfully aligned FFA and RSC by significantly driving the univariate responses of both areas (as seen from the plots in **Fig. 6a**), these images were not constrained to control the other areas in any specific way. Thus, even if a batch of controlling images aligns two areas, it does not necessarily alter the overall representational relationship configuration of all areas. We found that this was the case for some pairwise comparisons of areas (e.g., in **Fig. 7d**, red condition, the controlling images disentangling V1 and V2 lead V1 being closer to V3 than to V2 in MDS space), but not for all (e.g., in **Fig. 7d**, green condition, the controlling images driving both FFA and RSC). More in general, strictly controlling the representational relationship configuration of all areas requires jointly controlling all these areas, which we see as a good future application for RNC. We edited the Discussion to reflect this last, general point (**Discussion, page 13, lines 465-470**):

“Using RNC to jointly control all areas within the visual cortical network—as opposed to pairwise comparisons of areas, as done here—might reveal new representational relationship configurations and properties. For instance, it might uncover configurations that maximize the representational alignment of two areas and their disentanglement from all other areas, or representational relationships that diverge significantly from the typical configuration.”

Remarks on code availability

The code repository has been updated to incorporate new code for the additional analyses presented in this revised manuscript.

We thank Reviewer #2 for once again checking the code.

Reviewer #3

Remarks to the Author

Thank you very much for sending the revised manuscript. The paper is much stronger now after the additions. I do not have any new requests but just wanted to add a few points in response:

We thank Reviewer #3 for reading the revised manuscript and for the positive evaluation, as well as for the comments of the first review round that resulted in valuable additional analyses and discussions.

- In regards to using the Illusory image reconstruction dataset for training, while it is a useful addition, the dataset is based on mostly images selected from ImageNet so it is relatively redundant to the ImageNet. My comment at the time was to use the test images for non-natural images as OOD data which you already achieved using the synthetic-NSD dataset.

We are glad that we could provide a measure of model robustness by using NSD-synthetic as an OOD test set.

- I agree with your comment that it is not easy to discern whether the drop in performance is due to the training on natural images or because of other components such as top-down component (it is most-likely a mixture of everything). To actually measure the effect of top-down, that would be another analysis (such as one we did in Abdelhack and Kamitani 2018) that is beyond the scope of this work. I thought it was important to discuss these points in the discussion though as a guide to future studies which you did very well.

We thank the Reviewer for the positive comments regarding the role of top-down processing discussed in the manuscript, and also for the paper recommendation.

Remarks on code availability

The authors have addressed my previous comment on the code.

We thank Reviewer #3 for once again checking the code.